# Stress-dependent *miR-980* regulation of Rbfox1/ A2bp1 promotes ribonucleoprotein granule formation and cell survival

Mariya M. Kucherenko[1] & Halyna R. Shcherbata [1]

Upon stress, profound post-transcriptional adjustments of gene expression occur in spatially restricted, subcellular, membraneless compartments, or ribonucleoprotein (RNP) granules, which are formed by liquid phase separation of RNA-binding proteins with low complexity sequence domains (LCDs). Here, we show that Rbfox1 is an LCD-containing protein that aggregates into liquid droplets and amyloid-like fibers and promiscuously joins different nuclear and cytoplasmic RNP granules. Using *Drosophila* oogenesis as an in vivo system for stress response, we demonstrate a mechanism by which Rbfox1 promotes cell survival. The stress-dependent miRNA *miR-980* acts to buffer Rbfox1 levels, since it targets only those *Rbfox1* transcripts that contain extended 3′UTRs. Reduced *miR-980* expression during stress leads to increased Rbfox1 levels, widespread formation of various RNP granules, and increased cell viability. We show that human RBFOX proteins also contain multiple LCDs and form membraneless compartments, suggesting that the RNP granule-linked control of cellular adaptive responses may contribute to a wide range of RBFOX-associated pathologies in humans.

[1] Max Planck Research Group of Gene Expression and Signaling, Max Planck Institute for Biophysical Chemistry, Am Fassberg, 11, Goettingen 37077, Germany. Correspondence and requests for materials should be addressed to H.R.S. (email: halyna.shcherbata@mpibpc.mpg.de)

S tress requires regulation of gene expression. There is increasing evidence for spatial and temporal regulation of gene expression at the post-transcriptional level[1]. This often involves the formation of specialized membraneless compartments, or ribonucleoprotein (RNP) granules, also called liquid organelles, which are formed by proteins containing RNA-binding and low complexity sequence domains (LCDs)[2–4]. There are different types of non-membranous compartments: stress granules and processing bodies in the cytoplasm; nucleoli and Cajal bodies in the nucleus; and mitochondrial RNA granules[5–8].

All of these structures are highly dynamic and often form or increase in size and number upon specific changes in the cellular or organismal environment.

There must be a simple and robust signaling cascade in place that quickly accommodates cellular metabolism to respond to reversible stress. miRNAs are candidates for such a role, since they can simultaneously regulate multiple targets. Multiple components of the miRNA-induced silencing complex are detected in RNP granules, implying that miRNA-based regulation predominantly happens in these non-membranous subcellular compartments[9–11]. Recently, miRNAs have been implicated as major stress-response factors in many organisms[12–17]. It has been demonstrated that stress-dependent alterations in miRNA expression can affect multiple mRNAs simultaneously via direct targeting. However, the idea that miRNAs can also regulate multiple RNAs indirectly by targeting a LCD-containing protein involved in the formation of various RNP granules, thus regulating RNA metabolism, has not been explored. Moreover, knowledge of the functional role of non-membrane compartments in regulating gene expression, especially in multicellular organisms, is largely missing. In particular, oogenesis is a very well-studied process that is known to be extremely sensitive to stress, and in which miRNAs have been shown to play critical roles[14,16]. Using *Drosophila* oogenesis as a readout model for starvation-induced stress, we describe new characteristics for Rbfox1 protein as a structural component of RNP granules. We show that Rbfox1 levels are adjusted by the stress-sensitive miRNA, *miR-980*, which significantly influences the stress response. Rbfox1 has previously been implicated in ovarian germline differentiation; moreover, recently it has been shown that *Rbfox1* is regulated by *miR-980* in the process of memory formation[18–20].

Rbfox1 is the *Drosophila* homolog of human RBFOX1/ Ataxin2-binding protein 1 (Rbfox1/A2bp1), which is known to be involved in alternative splicing[21–30]. In addition, it has been shown that mammalian RBFOX1 can be found not only in the nucleus, but also in the cytoplasm, where it binds to *3′UTRs* of multiple mRNAs, regulating their stability[26,31]. Dysfunctions of human RBFOX proteins are associated with various medical conditions, including spinocerebellar ataxia type 2, mental retardation and epilepsy, attention-deficit hyperactivity disorder, autism, hand osteoarthritis, congenital heart defects, obesity, and diabetes[26,31,32]. The wide range of RBFOX1-associated diseases suggests that RBFOX1 alterations could have a more general effect on regulation of gene expression and that its expression must be tightly controlled.

Here we find that apart from the highly evolutionarily conserved RNA-binding (RRM) domain, Rbfox1 contains multiple LCDs that can be differentially included by alternative splicing. Based on our in vivo and in vitro analyses, we show that depending on its expression level and specific isoforms, Rbfox1 assembles in various RNP granules, which differ in their content, subcellular localization, and function. RNP granules can range from liquid droplets to amyloid-like fibers, and we detect Rbfox1 in all these states in a living organism. In this study, we also find that Rbfox1 promiscuously associates with the nucleolus and Cajal bodies in the nucleus, as well as stress granules and processing bodies in the cytoplasm, possibly, via its multiple LCDs. We uncovered an elegant mechanism by which Rbfox1 levels are adjusted by a stress-dependent miRNA. *miR-980* buffers Rbfox1 levels, since it can target only the portion of *Rbfox1* transcripts that contain extended 3′UTRs. This prevents the reduction of Rbfox1 levels below a certain threshold, which is also detrimental for cellular homeostasis. Reduced *miR-980* expression during stress leads to increased Rbfox1 levels, followed by widespread formation of RNP granules, promoting cell survival. In addition, our informatics analyses on human RBFOX proteins show that they also contain multiple LCDs, and data from human fibroblasts and neurons suggest that human RBFOX1 can associate with various RNP granules, assembly of which is augmented by stress. This demonstrates that similar functions exist for human and *Drosophila* RBFOX1 and suggests a putative molecular mechanism explaining how multiple human disorders develop due to RBFOX1 deregulation upon stress.

## Results

**Stress-responsive *miR-980* targets Rbfox1 in vitro and in ovaries**. *Drosophila miR-980* was previously identified as stress-linked in a screen for miRNAs that are differentially expressed in response to stress and disease[17]. *miR-980* was found to be downregulated upon various stresses (Supplementary Table 1), suggesting that it could play a role in stress tolerance. To investigate *miR-980* function, we generated loss-of-function mutants by imprecise P-element excision (Supplementary Figure 1A–E, Supplementary Table 1) and analyzed the phenotypes caused by *miR-980* deficiency using the newly generated mutants (*miR-980^Ex1-1^*, *miR-980^Ex1-2^* and *miR-980^Ex1-3^*) and an additional *miR-980^KO^* allele[33].

We found that *miR-980* is expressed in the ovarian germline and soma (Supplementary Figure 1E, F). In addition, the ovaries of *miR-980* mutants seemed to be enlarged, and their size did not

**Fig. 1** Rbfox1 levels depend on *miR-980* and stress. **a** Schematics of ovaries and a single ovariole (lower panel), where the germline (blue) is surrounded by the somatic follicular epithelium cells, in which at st.7, Notch signaling-dependent mitotic-to-endocycle and at st.10b, endocycle-to-amplification transitions occur. These phases are distinguished by Cut (red), expression of which is repressed by activated Notch signaling. Upon stress, egg production is sharply terminated at the nutritional stress checkpoint, resulting in small ovaries. Images of sated and 10-day protein-starved *Control* and *miR-980* ovaries. Starvation response is not as robust in *miR-980* mutants, their ovaries are larger and contain later stage egg chambers. **b** *miR-980* overexpression significantly reduces expression of luciferase reporters (P1 and P2), containing parts of the extended *Rbfox1 3′UTR* with different *miR-980* binding sites. **c** *miR-980* ovaries have significantly increased *Rbfox1* mRNA levels. **d** Nutritional stress (2-day) influences *miR-980* and *Rbfox1* expression in ovaries: *miR-980* is significantly downregulated, while *Rbfox1* is elevated. **e** Rbfox1 levels in wild type (*WT*), *miR-980*, and Rbfox1-overexpressing (*hsFlp; act > CD2 > Gal4 UAS-GFP/UAS-Rbfox1-RE*) ovaries from sated and protein-deprived animals detected by western blotting. Colored arrows point to Rbfox1 isoforms that were used for Rbfox1 expression quantification shown in the bar graph. **f** Rbfox1 expression pattern is temporally and spatially dynamic. At mitotic stages (st.3–6), Rbfox1 is nuclear; at early endocycling (st.6–8), its levels are strongly decreased; at late endocycling (st.9–10a), nuclear Rbfox1 reappears. At amplification stages (st.11–14), Rbfox1 nuclear expression is strongly reduced, but the moderate cytosolic remains. **g** In *miR-980^Ex1-2^*, Rbfox1 is elevated in comparison to *Control* egg chambers of the corresponding stages. **h** Assembly of *Control* and *miR-980* ovarioles under normal and starvation stress conditions. Delayed termination of egg production in *miR-980* mutants is rescued by *Rbfox1* downregulation. **i** Quantification of ovarioles with no response to starvation. Introduction of one copy of *Rbfox1* mutation to the *miR-980* loss-of-function background rescues the phenotype. DAPI (green), Cut (blue), Rbfox1 (red, rabbit anti-Rbfox1). **f, g** maximum intensity projections of multiple Z-sections. Scale bars 5 µm. AVE ± SD are reported from triplicates, two-tailed Student's t-test was applied for statistics in **b–d**, and Chi-square test in **i**. See Supplementary Tables 1–3. ** p ≤ 0.01, *** p ≤ 0.001

decrease as apparently as in controls in response to starvation stress (Fig. 1a), suggesting that the miRNA *miR-980* is involved in fine-tuning of the dietary stress response during oogenesis. Therefore, next, we used *Drosophila* oogenesis as an in vivo model to address *miR-980* stress-related functions. Using the miRNA target prediction database TargetScan[34], we found three predicted conserved target sites for *miR-980* in the *3′UTR* of

*Rbfox1* mRNA. The *Rbfox1* gene encodes different isoforms (Supplementary Figure 2A, B) that also differ by the *3′UTR* choice, and only the extended *3′UTRs* (medium and long) contain *miR-980* binding sites (Supplementary Figure 2C, D). To test if *miR-980* targets *Rbfox1* via the predicted miRNA binding sites, we cloned different *Rbfox1 3′UTR* regions containing *miR-980* binding sites into luciferase reporters (P1 and P2, Supplementary

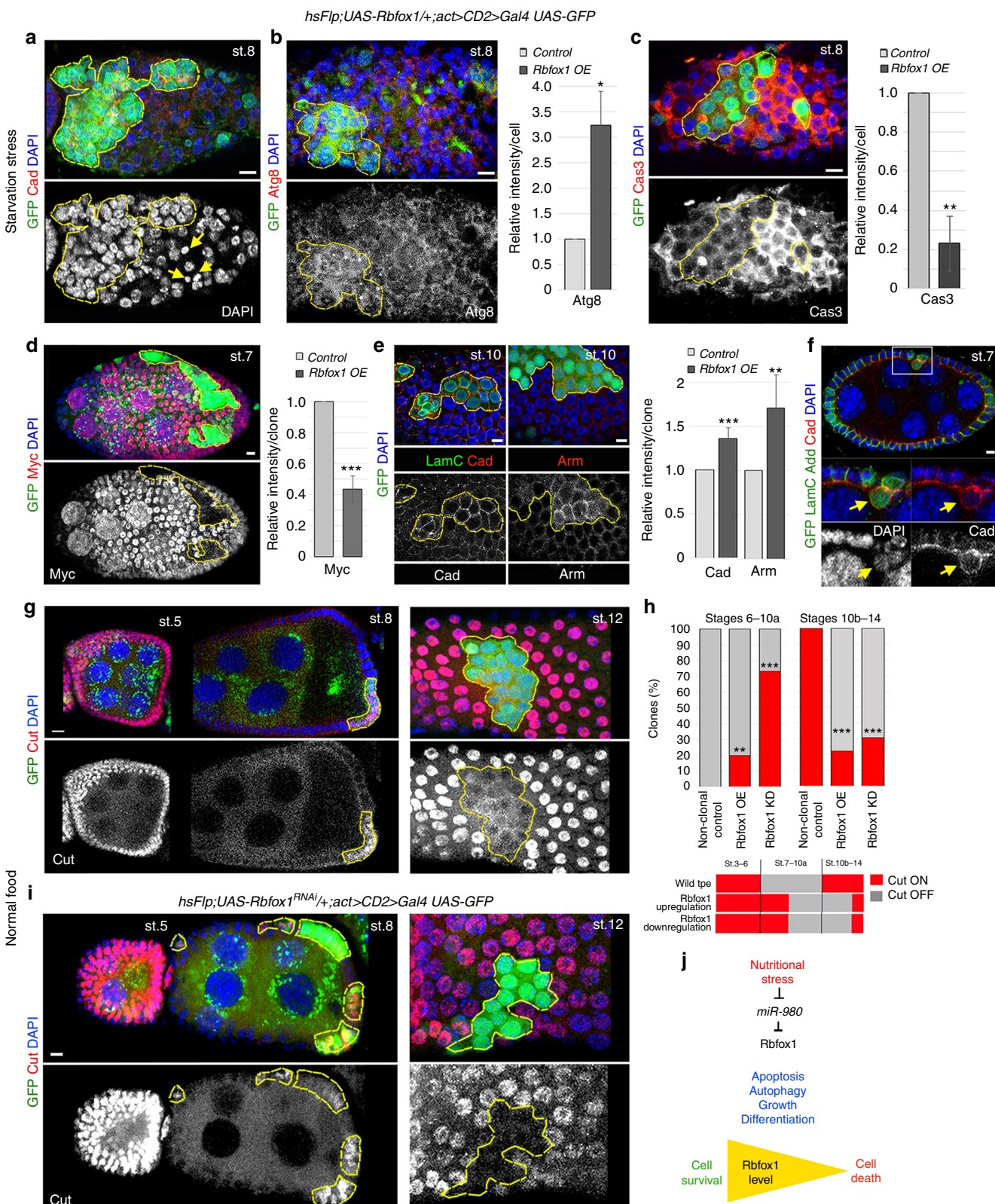

Figure 2D). We found the expression of both reporters to be substantially reduced upon *miR-980* overexpression in S2R+ cells (Fig. 1b, Supplementary Table 2). This miRNA-based regulation was *miR-980*-specific, since overexpression of other miRNAs that are not predicted to target *Rbfox1* (*miR-966* and *miR-278*) did not result in the downregulation of the luciferase reporter containing *Rbfox1* 3′UTR (Fig. 1b, Supplementary Table 2). Similar to in vitro results, in vivo, *Rbfox1* mRNA levels in ovaries were increased upon *miR-980* loss, confirming that *miR-980* regulates *Rbfox1* during oogenesis (Fig. 1c, Supplementary Table 3). *miR-980* and Rbfox1 expression depends on stress: *miR-980* levels are substantially reduced, while *Rbfox1* mRNA levels are increased in response to metabolic stress (Fig. 1d, Supplementary Tables 1 and 3).

To determine whether stress-dependent miRNA-based regulation of Rbfox1 also occurs at the protein level, we analyzed Rbfox1 protein levels and cellular localization, using two different antibodies[19,35]. First, we tested Rbfox1 protein levels in ovarian lysates from sated and starved wild type and *miR-980* mutant animals. We found that Rbfox1 protein levels in general were noticeably increased in response to starvation stress (Fig. 1e). Moreover, higher Rbfox1 levels were detected in both sated and starved *miR-980* mutant ovaries, and this upregulation was isoform-specific (Supplementary Figure 3A, Fig. 1e). While some of the isoforms maintained unchanged expression levels regardless of differences in genotypes or conditions, others responded specifically to either miRNA deficit or starvation. In particular, Rbfox1 isoforms that migrated at ~50 kDa, ~65 kDa, ~75 kDa, and ~120 kDa showed increased expression levels in *miR-980* mutants in comparison to controls at normal conditions (Fig. 1e). Among these, Rbfox1 ~50 kDa isoform was specifically upregulated upon *miR-980* loss (Fig. 1e, Supplementary Figure 3A). At the same time, an *Rbfox1* isoform that runs at ~110 kDa was upregulated in controls and *miR-980* mutants only upon starvation, while ~60 kDa isoform levels did not show any significant fluctuations at different conditions and in the studied genotypes (Fig. 1e, Supplementary Figure 3A). Unfortunately, currently annotated transcripts do not represent all possible combinations of alternative isoforms[18–20]; therefore, it is not possible to conclusively assign the behavior of different bands on the western blots to the specific Rbfox1 isoforms.

*miR-980* can target *Rbfox1* cell-autonomously, since its clonal overexpression in the ovarian follicular epithelium resulted in the reduction of Rbfox1 protein levels compared to neighboring wild type cells (Supplementary Figure 3B,D). This miRNA-based regulation of Rbfox1 protein expression was comparable to the effect of *Rbfox1* RNAi (Supplementary Figure 3B and E).

Altogether, these data show that stress and *miR-980* affect Rbfox1 expression at the mRNA and protein levels.

In the germline and somatic cells of developing egg chambers, Rbfox1 protein has a very distinctive expression pattern that depends on the developmental stage (Fig. 1f). This is especially evident in the follicular epithelium, which undergoes major transitions in cell cycle mode during differentiation (Fig. 1a, f). In *miR-980* mutants, elevated Rbfox1 levels were detected at all developmental stages (Fig. 1g), demonstrating that *miR-980* is responsible for the control of the highly dynamic Rbfox1 expression during the entire process of oogenesis.

Since *miR-980* mutants showed abnormal response to starvation (Fig. 1a), next, we analyzed whether *miR-980* targeting of *Rbfox1* has a functional role in the dietary stress response. The delay in the termination of egg production upon starvation caused by *miR-980* deficiency can be fully rescued by the reduction of the *Rbfox1* gene by one copy (Fig. 1h, i). This demonstrates that the *miR-980* diet-sensitive ovarian phenotype is Rbfox1-dependent and that the higher levels of Rbfox1 are beneficial for oogenesis progression upon nutritional stress.

**Rbfox1 promotes cell survival and ovarian cell differentiation.** *Rbfox1* mRNA has the possibility of having alternative 3′UTRs − two short and two long 3′UTRs – and all three conserved *miR-980* target sites are found only in the alternative extended 3′UTR (Supplementary Figure 2C, D). mRNA isoforms with shorter 3′ UTRs are less available for miRNA regulation. Therefore, we wanted to address how important is the presence of alternative 3′ UTRs, thus, targeting of Rbfox1 by *miR-980*, and what effect this regulation has on cellular status. To test what happens if *Rbfox1* is not subjected to *miR-980*-based regulation, we analyzed mutants overexpressing one of the Rbfox1 isoforms lacking the extended 3′UTR (*Rbfox1-RE*, Supplementary Figure 2B).

It has been shown that upon starvation, to ensure animal survival, both the germline and somatic cells die and the entire egg chamber content is recycled[36]. Since *miR-980* deficiency results in higher Rbfox1 levels, we analyzed the consequence of *miR-980*-insensitive Rbfox1 isoform upregulation in follicle cell clones upon stress. We found that the follicle cells that had higher levels of Rbfox1 did not die upon starvation (green cells, Fig. 2a). This was in sharp contrast to the surrounding non-clonal wild type cells that show fragmented DNA (Fig. 2a, lower panel, arrows). In addition, in Rbfox1-overexpressing cells, the levels of the autophagy marker Atg8 were upregulated (Fig. 2b), while the levels of the apoptosis marker Caspase3 were downregulated (Fig. 2c). This was accompanied by slower growth, perceived by

**Fig. 2** Rbfox1 is a pro-survival factor that regulates follicular epithelium differentiation. **a** In starved females, Rbfox1-overexpressing clones (*Rbfox1 OE*, GPF-positive, *hsFlp; UAS-Rbfox1-RE/ + ; act > CD2 > Gal4 UAS-GFP*) have higher cell density and decreased cell death. Note condensed DNA (separate channel, arrows) in non-clonal control cells, which die in response to starvation. **b** Abnormal autophagy measured by the significantly increased Atg8 intensity in *Rbfox1 OE* ($n = 27$) versus neighbouring control cells within the same egg chamber ($n = 27$); **c** Decreased apoptosis in *Rbfox1 OE* cells detected by the reduced Caspase3 intensity (*Rbfox1 OE*, $n = 28$, *Control*, $n = 35$). **d** Even without starvation, Rbfox1 overexpression reduces cell growth, illustrated by the significantly decreased Myc expression in *Rbfox1 OE* clones in comparison to the equal size of the non-clonal neighboring area within the same egg chamber ($n = 17$). **e** *Rbfox1 OE* cells express higher levels of the cell adhesion proteins Cadherin (DE-Cad, $n = 10$ clones) and β-Catenin (Arm, $n = 5$ clones), and form a multilayered epithelium (arrow, **f**). **g**, **h** Temporal regulation of the Notch-dependent transcription factor Cut is abnormal, showing that activation of Notch signaling required for proper follicle cell differentiation is slowed-down upon *Rbfox1* overexpression (**g**). **i** When Rbfox1 is downregulated, Cut expression is also abnormal. **h** Bar graph and scheme show that timely repression or activation of Cut is delayed in the endocycling (st.6-10a) or amplifying (st.10b-14) mutant follicular epithelial cells, respectively. Cut expression during endocycling stages was detected in 20% and 80% of clones that have higher or lower Rbfox1 levels, respectively, in comparison to 0% in control non-clonal cells. At the same time, significantly higher percentages, 80% and 70% of clonal cells with overexpressed or downregulated Rbfox1, failed to turn on Cut expression during amplification stages ($n = 19$ and 43 clones for *Rbfox1 OE* and *Rbfox1 KD*, respectively). **j** Scheme represents *miR-980*/Rbfox1 function in response to nutritional stress. Upon stress, *miR-980* levels are reduced, causing Rbfox1 upregulation and promoting cell survival. Images in **a–g** and **i** are maximum intensity projections of multiple Z-sections. Scale bars 5 μm. Developmental stages (st.) of egg chambers are marked. Relative antibody staining intensity (AVE ± AD) is presented as relative to control (**b–e**). Student's *t*-test was applied for statistics in **b–e** and Chi square in **h**, *$p ≤ 0.05$; **$p ≤ 0.01$; ***$p ≤ 0.001$

noticeably decreased Myc levels (Fig. 2d). We also observed that the clonal cells with increased Rbfox1 express higher levels of cell adhesion proteins (Cadherin and Armadillo/, β-catenin) and delaminate from the monolayered follicular epithelium (Fig. 2e, f). Moreover, they failed to properly differentiate, as they express higher levels of the transmembrane receptor Notch (Supplementary Figure 3F, G). Notch signaling activation is essential for proper follicle cell differentiation and to switch their cell cycle mode[37]. In clonal cells overexpressing Rbfox1, Notch signaling

activation (monitored by the repression of the Notch-regulated transcription factor Cut) was delayed (Fig. 2g, h). Together, these data show that elevated Rbfox1 affects cellular status at multiple levels: it slows cell differentiation and growth, alters autophagy, and downregulates apoptosis.

We also examined follicle cell clones in which Rbfox1 was downregulated via RNAi. Rbfox1 downregulation also delayed cell differentiation (Fig. 2h, i, Supplementary Figure 3G, H). However, contrary to its upregulation, Rbfox1 downregulation in

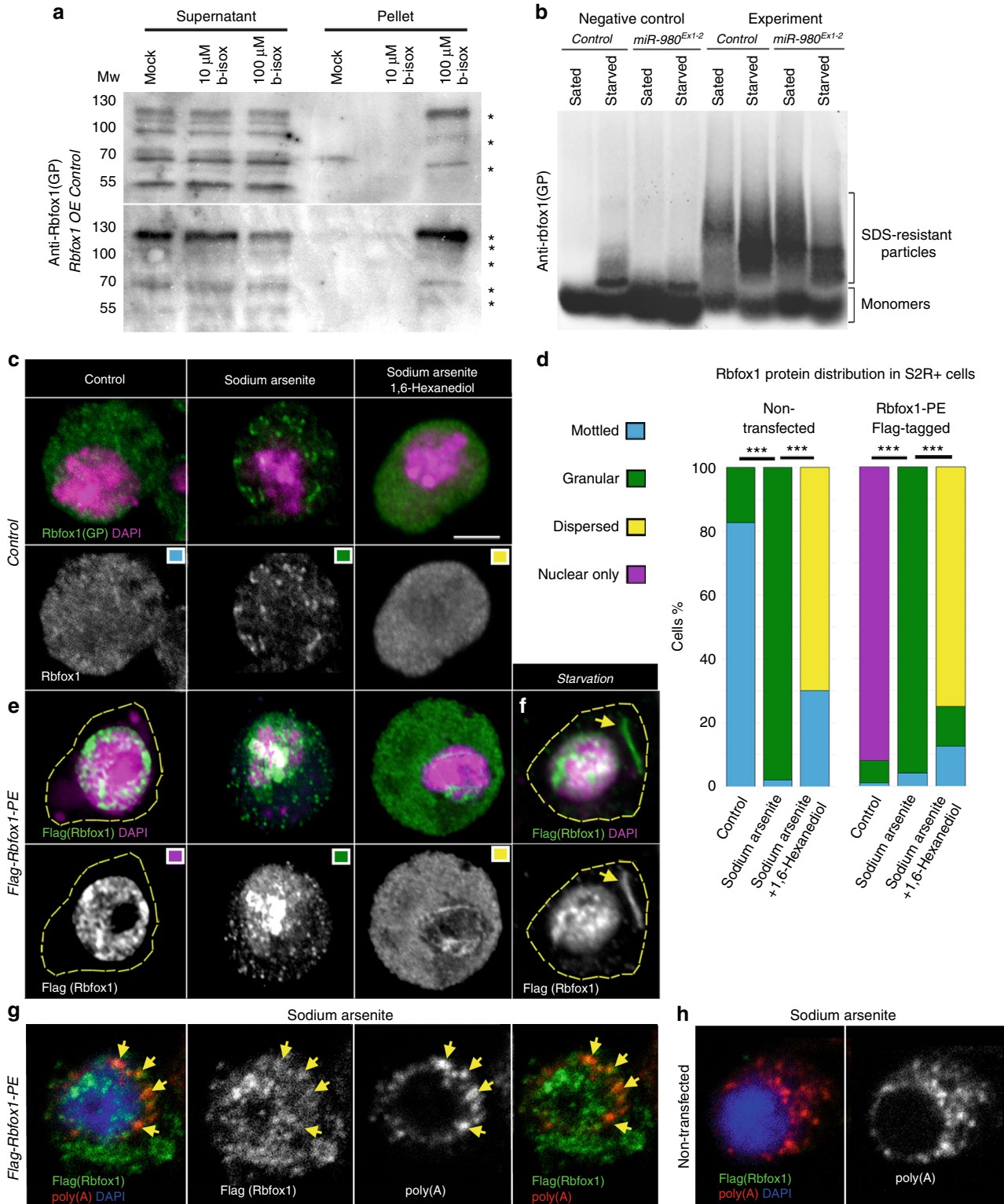

follicle cell clones increased cell death (Supplementary Figure 3H). Even more, in *Rbfox1* hypomorphic mutants, both the germline and somatic cells were completely unable to differentiate (Supplementary Figure 3I, J). Instead of forming one oocyte and 15 nurse cells with large nuclei per one egg chamber, Rbfox1 mutant egg chambers contained many small, undifferentiated germline cells (compare st.8 in Fig. 1f and Supplementary Figure 3I). In addition, they were encased by a layer of improperly differentiated epithelial cells that failed to enter the endocycle, as their nuclei were smaller and their number was higher when compared to the control egg chambers of the same size (compare Supplementary Figure 3C and 3J). This suggests that Rbfox1 is required for ovarian cell differentiation, and it would be unfavorable for all *Rbfox1* isoforms to be subjected to *miR-980* regulation. Thus, having a choice of alternative *3′UTRs* maintains the precision of Rbfox1 expression.

In summary, these data show that upon starvation, the regulation of Rbfox1 expression is partially achieved via stress-dependent *miR-980*-based buffering and that increased Rbfox1 levels modulate cell growth, differentiation, autophagy, and apoptosis. Lower Rbfox1 levels lead to cell death (Fig. 2j).

**RNA-binding protein Rbfox1 contains multiple LCDs**. Human Fox family proteins and *Drosophila* Rbfox1 have a highly evolutionarily conserved RNA recognition motif (RRM) domain (Supplementary Table 4). Our analysis also identified *Drosophila* Rbfox1 protein contains multiple low complexity sequence domains (LCDs)—stretches with little diversity in amino acids (Supplementary Figure 4A, black frames, Supplementary Table 5). These amino acids in the Rbfox1 protein are polar (Supplementary Figure 4A, green), and in general, *Drosophila* Rbfox1 contains multiple glutamine-rich (polyQ) stretches[35], a composition similar to prion-like proteins (Supplementary Figure 4A, B). The number and the length of LCDs and polyQ repeats vary in different Rbfox1 isoforms (Supplementary Figure 4A, B, Supplementary Table 5). RNA-binding proteins with LCDs are intrinsically disordered and can phase separate from the cytoplasm or nucleoplasm[38,39]. In particular, prion-like LCDs can form an array of membraneless assemblies with different material properties, ranging from liquid droplets to hydrogels to amyloid-like fibers[1,3,4,38–42].

**Rbfox1 forms liquid-like aggregates upon stress**. Since Rbfox1 has both a conserved RRM and LCDs, we hypothesized that it can aggregate into subcellular RNP granules with different physical and biochemical properties. Also, since GO term analysis identified stress-related functions for Rbfox1 (Supplementary Figure 4C) and our data show that it has a role in ovarian response to stress (Fig. 1), we presumed that Rbfox1 aggregation could be stress-dependent. The biochemical isolation of stress granules in vivo has not been developed to date; however, it has been discovered that a biotinylated version of isoxazole (b-isox) can precipitate LCD-containing RNA-binding proteins from cell lysates, mimicking stress in vitro and resulting in the cell-free formation of RNP granules[43,44].

To test whether Rbfox1 can aggregate via its LCDs, we analyzed cell lysates from control ovaries that were treated with different b-isox concentrations. This indeed resulted in Rbfox1 precipitation in a b-isox concentration-dependent manner (Fig. 3a), establishing that Rbfox1 shares chemical characteristics with multiple previously described LCD-containing proteins that form hydrogels and amyloid-like fibers[43,44]. Therefore, similar to other LCD-containing RNA-binding proteins, Rbfox1 aggregates via LCDs (Fig. 3a).

In addition, we tested if the efficiency of Rbfox1 precipitation by b-isox depends on Rbfox1 concentration by overexpressing one of the Rbfox1 isoforms, Rbfox1-PE. The advantage of experimenting with this Rbfox1 isoform, apart from the absence of the extended *3′UTR*, lies in the assortment of protein domains that are included in Rbfox1-PE (Supplementary Figure 5). First of all, it contains the RRM binding domain that has been previously shown to be important for interaction of RBFOX proteins with RNA[24,25,27,28,30,45]. Second, it has ten out of thirteen predicted LCDs, which vary in their lengths and types. In particular, the coiled-coil domain, LCD1, and LCD7 contain poly(Q), while LCD10 and LCD13 are enriched for poly(A), LCD2 for poly(N), and LCD3 and LCD6 for poly(G) stretches (Supplementary Figure 5). Thus, the Rbfox1-PE isoform has the major protein domains that we assume have important roles in Rbfox1 protein function as an RNP granule component. We used the heat-shock inducible *hsFlp; act>CD2>Gal4* system to overexpress Rbfox1-PE in follicle cell clones. Comparison of the cell lysates from control and Rbfox1-PE-overexpressing ovaries that were precipitated with different b-isox concentrations shows that upon Rbfox1 overexpression, more Rbfox1 is precipitated by b-isox, demonstrating the specificity of this interaction (Fig. 3a, lower panel). These data show that Rbfox1, similarly to other LCD-containing proteins, precipitates with b-isox in a concentration-dependent manner.

**Fig. 3** Upon stress Rbfox1 aggregates in higher-order subcellular structures in cells. **a** Western blot demonstrates the behavior of Rbfox1 protein in response to b-isox treatment. Rbfox1 precipitation is seen in both control and Rbfox1-overexpressing ovarian lysates at the 100 μM compound concentration. Note the relocation of Rbfox1 from supernatant to pellet. Asterisks show various Rbfox1 isoforms precipitated by b-isox. **b** Aggregation of Rbfox1 protein extracted from ovaries analyzed using SDD-AGE. As a negative control, protein extracts were boiled at 95 °C for 10 min. Note that in the starved control sample, some of the Rbfox1-positive aggregates remain stable at high temperatures. Rbfox1 forms SDS-resistant particles under all tested conditions. Starvation stress increases the levels of SDS-resistant particles. In *miR-980* ovaries and during starvation, larger amounts of Rbfox1-positive SDS-resistant aggregates are detected when compared to well-fed controls. **c** Under control conditions, in non-transfected S2R + cells, Rbfox1 is found in small aggregates in the nucleus and cytoplasm (mottled pattern). Treatment by sodium arsenite, a stress factor promoting RNP granule formation, resulted in Rbfox1 accumulation in more defined speckles (granular pattern). Application of 1,6-hexanediol, which triggers the dissolution of non-amyloid RNP assemblies, caused Rbfox1 granule dissociation (dispersed pattern). **d** The frequencies of the observed Rbfox1 expression patterns under different conditions ($n \geq 50$ cells for each condition, two biological replicates). Two-Way Tables and Chi-square test were applied for statistics, ***$p \leq 0.001$. **e** Upon overexpression, Flag-tagged Rbfox1-PE protein is predominantly found in the nucleus (nuclear only pattern). Upon sodium arsenite treatment, Rbfox1 cytoplasmic granules appear, which are dissolved into "dispersed" speckles upon 1,6-hexanediol treatment. Colored squares in the upper corners in **c** and **e** correspond to the quantified phenotypes in **d**. **f** In Rbfox1-overexpressing S2R + cells, starvation (1 h) can result in the formation of distinct Rbfox1-positive fibers (yellow arrow). **g, h** in situ hybridization that detects poly(A) tails of mRNAs shows colocalization of RNA-containing RNP granules with Rbfox1 (arrows). **g** stressed S2R + cells expressing Flag-Rbfox1-PE and **h** stressed non-transfected S2R + cells. DAPI (magenta or blue). Specificity of Rbfox1 aggregation is confirmed by polyclonal guinea pig anti-Rbfox1(GP) and anti-Flag antibodies (green). **c–d**, **f** maximum intensity projections of multiple Z-sections, **g, h** single Z-sections. Scale bar=5 μm

To check the solubility of Rbfox1-positive aggregates, we used semi-denaturing detergent agarose gel electrophoresis (SDD-AGE), which allows the resolution of a wide size range of SDS-resistant aggregates, including amyloids[46]. We prepared ovarian lysates from wild type and *miR-980*-deficient animals that were well-fed or starved. As a negative control, samples were incubated at high temperature, which causes restoration of most of the amyloids to monomer state[46]. Indeed, Rbfox1 formed SDS-resistant aggregates, most of which were disintegrated by boiling (Fig. 3b). The amount of Rbfox1-positive aggregates detected by the SDD-AGE was dependent on stress and *miR-980* presence,

since the increase in Rbfox1 aggregations was observed in starved control and in well-fed and starved *miR-980* mutant ovaries in comparison to well-fed controls (Fig. 3b, experiment). We noticed that under stress, some of the Rbfox1-positive aggregates remained stable even after high temperature treatment, suggesting their insoluble nature (Fig. 3b, negative control). Together, these data show that first, it is possible to reconstitute the cell-free assembly of Rbfox1-positive aggregates using b-isox crystals, which specifically bind to LCDs. Second, the amount and the solubility of SDS-resistant Rbfox1-positive aggregates were condition-dependent, suggesting that the increase in Rbfox1

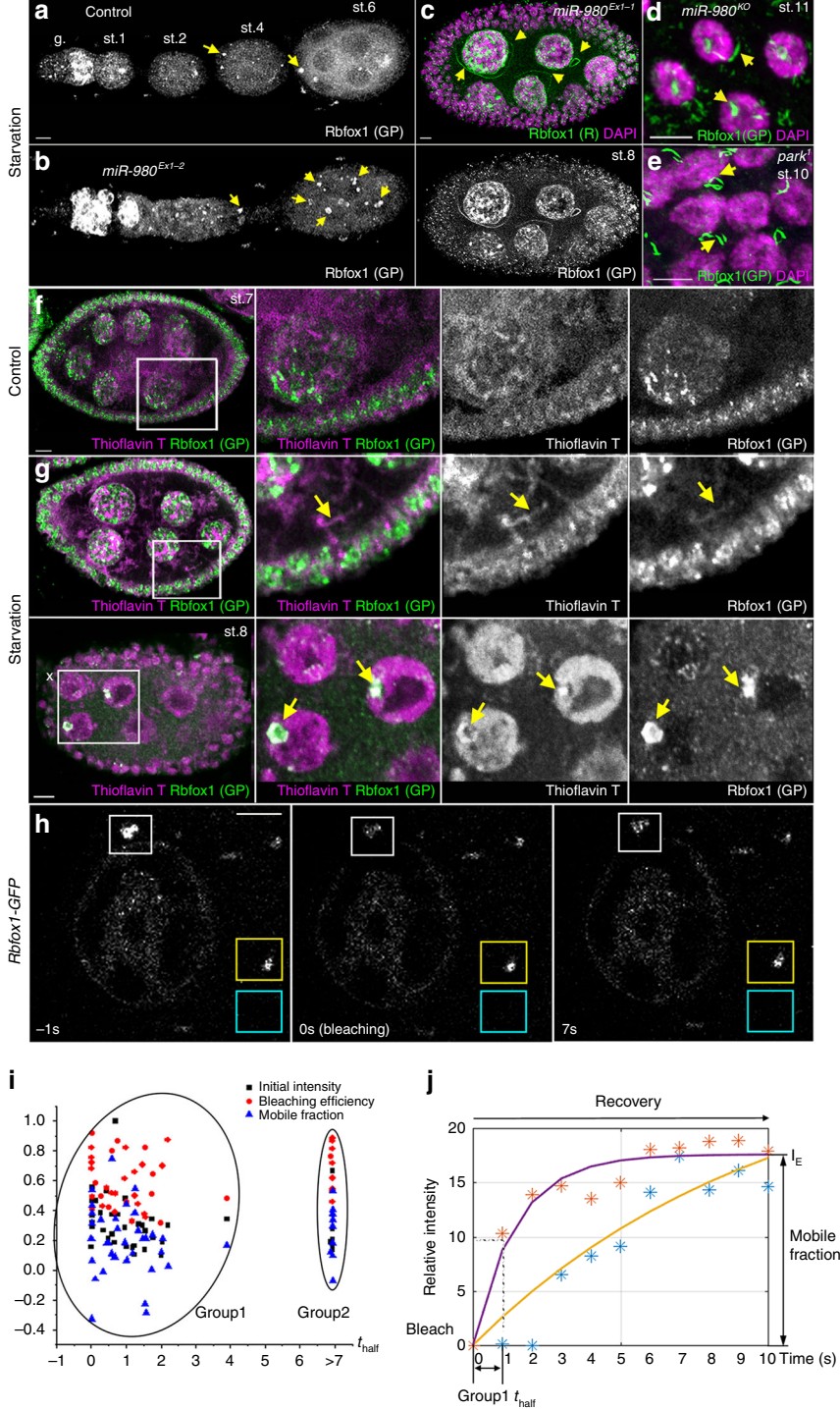

levels due to stress and *miR-980* loss promotes the formation of Rbfox1 aggregates and their transition into a less soluble, possibly amyloid-like state.

**Rbfox1 forms liquid-like compartments upon stress in cultured cells.** Next, we tested if Rbfox1-positive aggregates can be observed in vivo. First, cultured S2R+ *Drosophila* cells were treated with sodium arsenite, a commonly used inducer of stress-dependent compartments and processing bodies[47]. In non-transfected cells (Fig. 3c), low endogenous Rbfox1 protein levels were detected with anti-Rbfox1 antibodies in the cytoplasm and nucleus. Treatment with sodium arsenite stimulated Rbfox1 aggregation into cytoplasmic and nuclear granular structures (Fig. 3c, d). To investigate whether Rbfox1-positive granules are of liquid state, we applied 1,6-hexanediol to the stressed cells. 1,6-hexanediol can disintegrate associations formed through weak transient interactions and dissolve assemblies that have liquid- and hydrogel-like properties[5]. Indeed, 1,6-hexanediol addition led to the partial disintegration of sodium arsenite-induced Rbfox1 granules (Fig. 3c, e). Thus, these data suggest that in response to stress, Rbfox1 protein assembles into granules that have liquid droplet properties. These results were also confirmed using S2R+ cells expressing FLAG-tagged Rbfox1-PE isoform, which, when expressed under normal conditions predominantly localizes in the nucleus (Fig. 3d, e). Upon stress, Rbfox1-PE also assembled into nuclear and cytoplasmic granules, appearance of which could be reduced upon 1,6-hexanediol treatment (Fig. 3d, e). Sometimes, even long, filamentous Rbfox1-positive aggregates were observed in the starved, Rbfox1-expressing S2R+ cells (Fig. 3f). To test whether Rbfox1-positive granules contain RNA, we performed in situ hybridization in S2R+ cells stressed by sodium arsenite to detect *poly(A)* tails of mRNAs. We found that Rbfox1 colocalized with polyadenylated mRNAs (Fig. 3g, h), confirming that Rbfox1-positive granules contain RNA and indeed are *bona fide* RNP granules.

**Rbfox1 forms higher order structures in *Drosophila* ovaries.** Immunohistochemical analysis revealed that upon prolonged starvation stress, Rbfox1 assembled into round, granular structures in *Drosophila* ovaries (Fig. 4a, arrows). The abundance of these structures was increased in *miR-980* mutants (Fig. 4b), which, as we have shown above, express higher Rbfox1 levels. In addition, similar to the starved S2R+ cells overexpressing Rbfox1-PE (Fig. 3f), notable long filamentous structures were observed in stressed *miR-980* ovaries (Fig. 4c, d). Similar Rbfox1-positive fibers could also be detected in *parkin* mutants[48], which, due to

mitochondrial dysfunction, exhibit constant oxidative stress (Fig. 4e). These data demonstrate that Rbfox1 protein can assemble in vivo into stress-induced granules and fibers in ovarian cells. To check whether Rbfox1 fibers have amyloid characteristics, we stained ovaries with an amyloid marker, ThioflavinT[49]. We found that a portion of ThioflavinT-positive cytoplasmic and nuclear assemblies also co-stained with anti-Rbfox1 antibodies, confirming that Rbfox1 protein forms amyloid-like fibers and aggregates (Fig. 4f, g).

If Rbfox1 is capable of assembly of the liquid-like structures, then their recovery kinetics after photobleaching of the fluorescently labelled Rbfox1 protein should be fast. We performed fluorescence recovery after photobleaching (FRAP) experiments using Rbfox1::GFP, which is efficiently incorporated into granules (Fig. 4h). We found that Rbfox1-positive granules were quite heterogeneous and differed by their initial intensity, bleaching efficiency, mobile fraction, and half-time of fluorescence recovery (Fig. 4i). We found that Rbfox1-positive granules could be divided into two distinct groups based on their recovery time (Group 1: $n = 33$, $t_{half} = 0.96 \pm 0.85$ s and Group 2: $n = 10$, $t_{half} > 6.93$ s). Thus, FRAP analyses showed that Rbfox1 protein can re-assemble within RNP granules on timescales of seconds (Fig. 4j), which, based on the analysis of the previous data[50], leads us to propose that Rbfox1-positive granules belong to the category of liquid droplets showing medium recovery speed.

**Rbfox1 localizes to nuclear RNA-containing compartments.** To reveal the identity of Rbfox1-positive granules, we characterized its subcellular localization in vivo. As mentioned above, Rbfox1 expression in ovaries is extremely dynamic and depends on the developmental stage (Fig. 1f). For example, in the soma, Rbfox1 can be detected in the cytoplasm, the nucleus, or both, and it can be assembled into small or large round granules or fibril-like aggregates (Supplementary Figure 6A). In *miR-980* mutant ovaries, Rbfox1 protein levels as well as the size and number of Rbfox1-positive aggregates were apparently increased at all developmental stages in the germline and somatic cells (Figs. 1f, g, 5a, Supplementary Figure 6). To address the characteristics of Rbfox1-positive assemblies, we used multiple molecular markers. In particular, we tested for association of Rbfox1 with the subcellular liquid organelles known to be important for cellular stress response and differentiation and to be assembled via phase separation, such as the nucleolus and Cajal bodies in the nucleus.

The nucleolus is the largest membraneless organelle in the nucleus and exhibits liquid droplet characteristics[7,41]. While it is known that the nucleolar ultrastructure organization is highly

**Fig. 4** Upon stress Rbfox1 aggregates in higher-order subcellular structures in ovaries. **a, b** Upon prolonged protein starvation (1–3 weeks) in *Control* (**a**) and *miR-980* (**b**) ovaries, Rbfox1 aggregates in cytoplasmic and nuclear droplets of various size and shape. **c, d** Sometimes, Rbfox1 can form various phase-separated long fibers in both germline (**c** lower panel) and somatic follicle cells (**d**) of *miR-980* mutants. **e** Similar higher order structures can be found in starved mitochondrial stress-linked *parkin* mutant ovaries. **f, g** Rbfox1-positive fibers (**f**) and round granules (**g**) formed in starved ovaries co-stain with Thioflavin T, a benzothiazole dye that exhibits enhanced fluorescence upon binding to amyloid fibrils. **h–j** FRAP experiments performed on ovaries of flies with GFP-tagged Rbfox1 protein (Rbfox1$^{CC00511}$/TM3). **h** Images of germline cell at different time points; prebleached (−1 s), postbleached (0 s), and after recovery (7 s). White box (ROI1) outlines an Rbfox1-positive granule subjected to photobleaching, yellow box (ROI2) outlines an Rbfox1-positive granule not subjected to photobleaching, and blue box (ROI3) outlines a background area that was not subjected to photobleaching ($n > 50$ measurments for each ROI). **i** Scatter plot for correlation analysis between the initial intensity, bleaching efficiency, mobile fraction, and recovery half-time parameters. **j** Fitted plots of the FRAP recovery curves show two distinct groups of Rbfox1-positive droplets that differ in their recovery half-times. Note that for FRAP experiments, due to lethality, heterozygous animals that have only one copy of GFP-tagged Rbfox1 protein were used (Rbfox1$^{CC00511}$/TM3). This means that only half of the Rbfox1 proteins could be subjected to photobleaching and that non-labelled Rbfox1 proteins would also participate in liquid droplet recovery. Thus, the half-time of fluorescence recovery estimated here is most likely two times slower than the actual recovery half-time. DAPI (nuclei, magenta, **a–e**); Thioflavin T (amyloid, magenta, **f, g**); Rbfox1 (green, **a–g**). White rectangle in **f, g** outlines the area presented as separate panels (right), arrows point to Rbfox1-positive subcellular aggregates. **a, b**—ovarioles; **c, f, g**—egg chamber showing germline and somatic cells with large and small nuclei, respectively; **d, e**—follicular epithelial cells. Images in **a–e** are maximum intensity projections of multiple Z-sections, and images in **f–h** are single Z-sections. Scale bars 5 μm

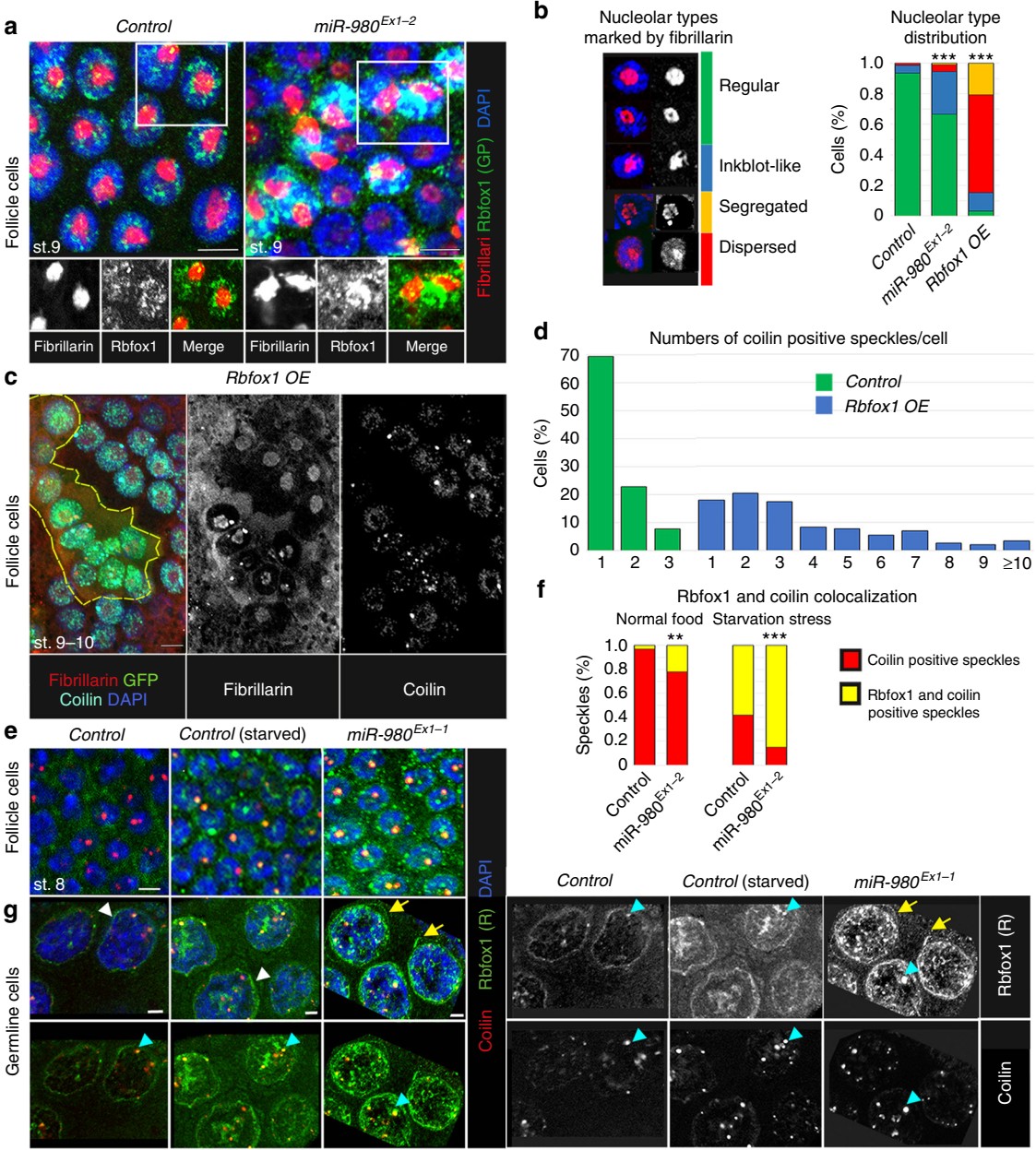

**Fig. 5** Rbfox1 colocalizes with membraneless nuclear organelles associated with RNA metabolism. **a** In follicle cell nuclei, Rbfox1 is associated with chromatin and enriched in the granular component of the nucleolus. Rbfox1 upregulation in *miR-980* mutants modifies nucleolar shape and size. **b** Quantifications of nucleolar type distribution in st.9-10 follicular epithelial cells (*n* = 273 cells for *Control*, 234 for *miR-980*[Ex1-2] and 92 for *Rbfox1 OE*). **c** Increased Rbfox1 expression modifies nucleolar shape and size and CB size and numbers. Rbfox1-overexpressing clone is outlined by yellow line (*hsFlp; UAS-Rbfox1/ + ; act > CD2 > Gal4 UAS-GFP*). **d** Most of the st.9-10 follicle cells contain one CB, while Rbfox1-overexpressing cells have 1 to > 10 smaller CBs per cell (*n* = 199 and 156 cells, respectively). **e** Rbfox1 association with CBs is enhanced by *miR-980* loss and stress. **f** Percentages of Rbfox1-positive CBs are significantly increased in *miR-980* mutants in comparison to *Control* (*n* = 82 and 256 CBs, respectively). This tendency becomes even more pronounced upon starvation stress (*n* = 598 and 262 CBs for *miR-980* and *Control*, respectively). **g** In the germline, Rbfox1 colocalizes with CBs (blue arrowheads) and this colocalization is enhanced by *miR-980* loss and stress. Rbfox1 can be also detected in the nuage of the germline nurse cells (white arrowheads). In starved *miR-980* mutants, Rbfox1-positive fibers can be observed (yellow arrows, right panel). Rbfox1 (green); DAPI (blue); Fibrillarin (nucleolus, red, **a–c**); Coilin (CB, cyan, **c**, red, **e**, **g**). Follicular epithelial cells (**a–c**, **e**), germline nurse cells (**f**). Images in **a**, **b**, **e–g** - single Z-sections, **c** - maximum intensity projection of multiple Z-sections. Scale bars 5 μm. Two-Way Tables and Chi-square test were applied for statistics in **b** and **f**, ***$p \leq 0.001$

specific for the cell type and its metabolic status, it is still difficult to correlate its appearances with cellular metabolism[51]. We observed that germline cells exhibit stage-specific variations in nucleolar appearance, and the somatic epithelial cells organize their nucleoli differently depending on whether they are endocycling or amplifying (Fig. 5a, Supplementary Figure 6A), demonstrating that nucleolar organization correlates with the

developmental status of the cell. The size and shape of the nucleolus is highly uniform in the follicular epithelial cells surrounding the same stage oocytes. For example, at st.9–10 A, when follicular epithelial cells undergo endoreplication and actively grow, most of them (95%) have a nucleolus with a regular or doughnut-shaped appearance (Supplementary Figure 6B). Co-staining of the nucleolar marker Fibrillarin with

Rbfox1 revealed that at st.9, nuclear Rbfox1 is predominantly localized around and sometimes within the nucleolus in controls (Fig. 5a, Supplementary Figure 6A). However, in *miR-980* mutants, Rbfox1 association with the nucleolus is even more evident (yellow in Fig. 5a). Moreover, *miR-980* deficiency and Rbfox1 overexpression caused changes in nucleolar morphology, as higher percentages of nucleoli appeared to have an inkblot-like shape or to be more segregated and dispersed in mutants in comparison to controls (Fig. 5a, b). Similarly, in the *miR-980*-deficient germline, the appearance of the nucleoli in the nurse cells was changed in comparison to control nurse cells of the same stage (Supplementary Figure 6C–F). Even more, over-expression of *miR-980*-insensitive Rbfox1-PE isoform in the follicle cells had an effect on the nucleolar size and shape (Fig. 5b, c). These data show that Rbfox1 can be localized in the nucleolus and that Rbfox1 levels affect nucleolar ultrastructure.

The other important stress-responsive organelle in the nucleus is the Cajal body (CB). It has been shown that Rbfox proteins are

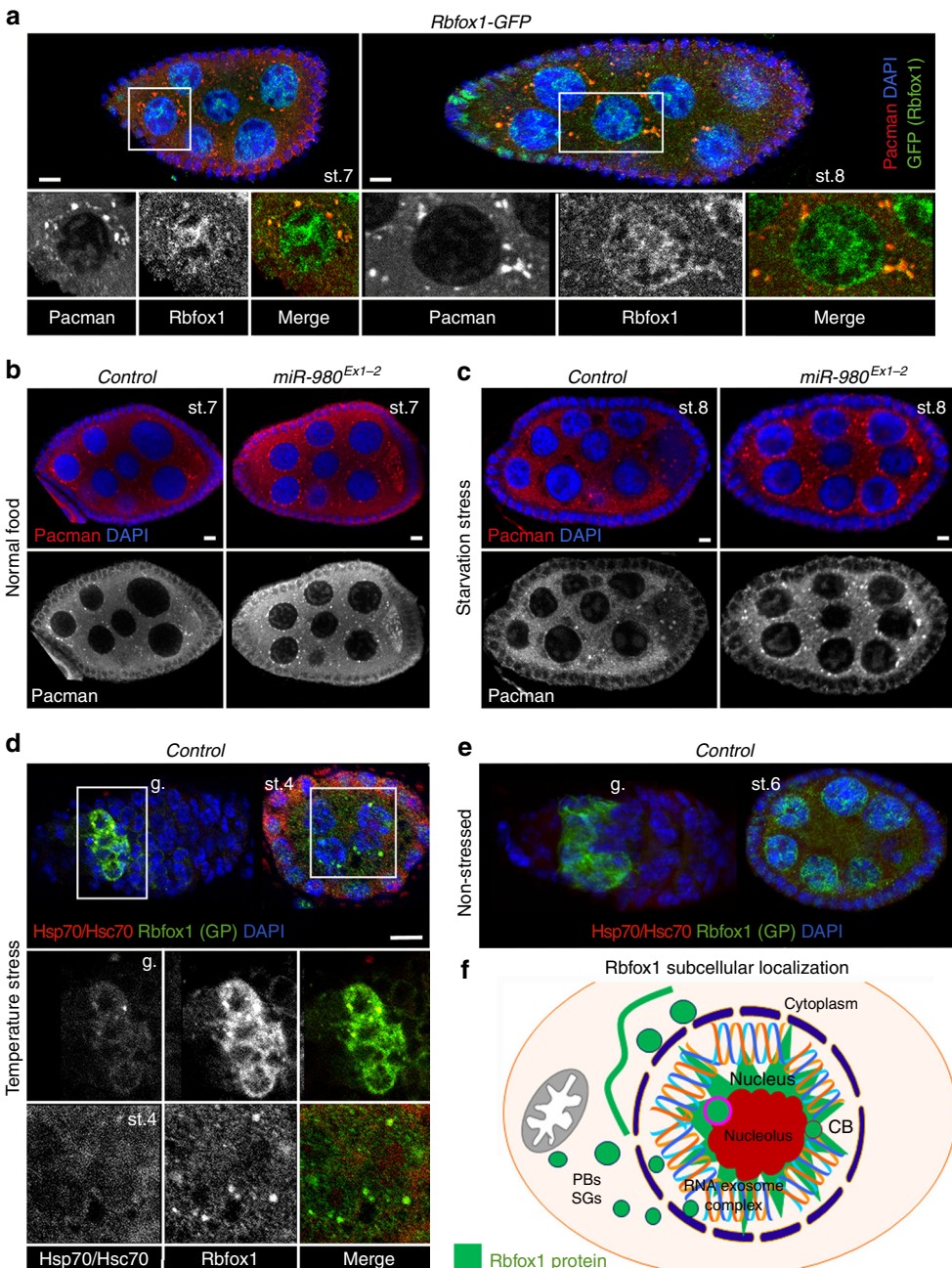

**Fig. 6** Rbfox1 colocalizes with P-bodies and stress granules in the cytoplasm. **a** In the germline, Rbfox1 co-stains with the P-body marker Pacman. Note that all Pacman-positive granules are also Rbfox1-positive, but not all Rbfox1 foci stain with Pacman. **b, c** Upon *miR-980* loss or starvation, which leads to elevated Rbfox1 levels, the appearance of P-bodies increases. **d** Upon temperature stress, Rbfox1 associates with the stress granules marked with anti-Hsp70/Hsc70 in the germarium and later stage egg chambers (g. and st.6 shown in lower panels). **e** Stress granules are undetectable in non-stressed ovaries. **f** Schematic drawing depicts a summary of subcellular localization of Rbfox1 protein. Rbfox1 is associated with the major membraneless organelles that regulate RNA metabolism, such as nucleoli and Cajal bodies in the nucleus and P-bodies, stress granules and amyloid-like fibers in the cytoplasm. Rbfox1 (green); DAPI (nuclei, blue); Pacman (P-bodies, red); Hsp70/Hsc70 (stress granules, red). Egg chambers are shown in **a–e**. Developmental stages (st.) of egg chambers are marked, germarium (g.). Images in **a–e** are single Z-sections. Scale bars 5 μm

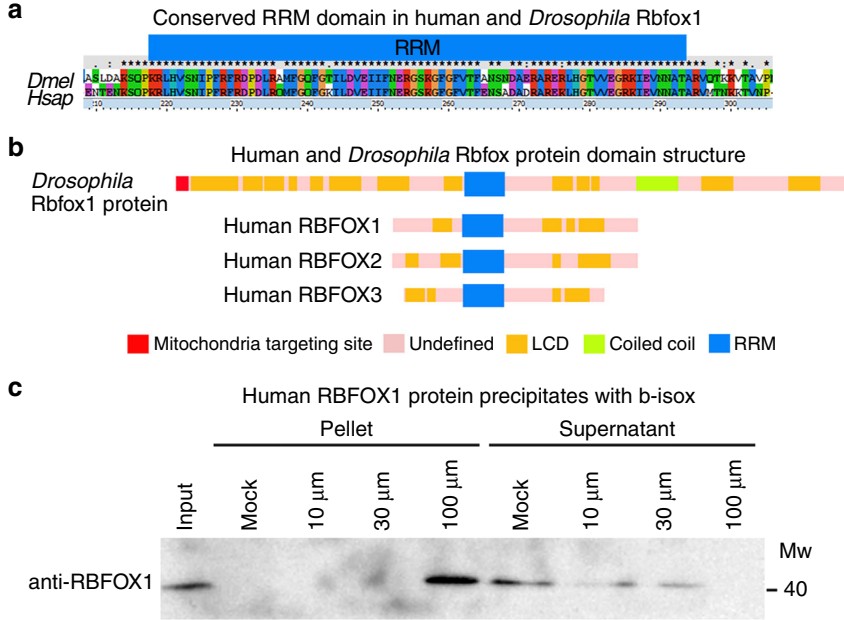

**Fig. 7** Human RBFOX proteins also contain multiple LCDs. **a** Protein sequence alignment of RNA recognition motifs (RRM) of human and *Drosophila* Rbfox1 reveals high conservation. **b** Schematic view of *Drosophila* Rbfox1 and human Fox family proteins (RBFOX1, RBFOX2, and RBFOX3). Our analysis predicts that human Fox family proteins also contain multiple LCDs; however, in comparison to human orthologs, *Drosophila* Rbfox1 has a larger number of LCDs with clearly defined poly-glutamine stretches. **c** In human fibroblasts, similarly to *Drosophila* Rbfox1, human RBFOX1 protein precipitates with b-isox in a concentration-dependent manner. Western blot with anti-human RBFOX1 antibodies shows complete precipitation of 43 kDa RBFOX1 upon treatment with 100 μM b-isox. Note the protein is appearing in the pellet, while disappearing from the supernatant fraction

involved in alternative splicing, implying that Rbfox1 could be associated with the CBs[45]. We found Rbfox1 levels have an effect on CB size and number (Fig. 5c, d). At st.10, the majority of the follicular epithelial cells (70%) contain one large CB, while upon Rbfox1 overexpression, more than 80% of cells have more than one CB per nucleus. In most of the cells (65%) with higher Rbfox1 levels, the number of CBs per cell ranged from 3 to >10 (Fig. 5c, d). In addition, we found that colocalization of Rbfox1 with the CB marker, Coilin, depends on the presence of *miR-980*. The percentage of CBs that also co-stained with Rbfox1 was increased in *miR-980* mutants that express higher Rbfox1 levels (Fig. 5e). In addition, Coilin and Rbfox1 colocalization increases upon stress. In controls, under normal conditions, less than 10% of CBs in the st.8 follicle cells were positive for Rbfox1, while upon starvation, >60% of CBs were Rbfox1-positive (Fig. 5f). This tendency was even more pronounced in *miR-980* mutants, in which more than 80% of CBs were Rbfox1-positive under stress (Fig. 5f). Similarly, in the germline, Rbfox1 colocalized with CBs (Fig. 5g, blue arrowheads) and the size and the number of CBs were increased in the starved and *miR-980*-deficient ovaries. Thus, our analyses of Rbfox1 nuclear localization demonstrate that Rbfox1 associates with the nucleolus and CBs and that Rbfox1 levels affect the organization of these nuclear, membrane-less organelles. This association with the nucleolus and CBs becomes more pronounced under conditions that lead to increased Rbfox1 expression, such as *miR-980* loss or starvation stress. Similarly, in the cytoplasm, the appearance of Rbfox1-positive granules depends on both stress and *miR-980*-based Rbfox1 regulation (Fig. 5e–g). We found that Rbfox1 assembles even longer fiber-like structures in the nuage of starved *miR-980* mutants (Fig. 5g, yellow arrow). Similar Rbfox1-positive, elongated assemblies were also seen in the somatic follicle cells at late oogenesis stages in wild type and *miR-980* deficient

animals (st.11-12, Supplementary Figure 6A). Therefore, next we analyzed the nature of Rbfox1-positive aggregates in the cytoplasm.

**Rbfox1 localizes to various cytoplasmic RNP granules.** Since in eukaryotes there are two major classes of cytoplasmic RNP granules: processing bodies (P-bodies) and stress granules (SGs), we tested whether the cytoplasmic Rbfox1-positive RNP granules belong to either of these two major classes. As a marker for P-bodies, we used antibodies against a key enzyme in RNA turn-over, Pacman, a *Drosophila* homolog of XRN1 (a 5′-3′ processive exoribonuclease that degrades decapped mRNAs and also functions in nonsense-mediated mRNA decay and degradation of mRNAs after miRNA or siRNA targeting[52,53]). We analysed co-expression of Pacman and Rbfox1 in the nurse cells of developing oocytes (Fig. 6a). Normally, Pacman is present everywhere in the cytoplasm and assembles into large granules around the nurse cell nuclei. We found that all Pacman-positive P-bodies also contain Rbfox1 protein; however, not all Rbfox1 granules contain Pacman (Fig. 6a). In general, Pacman expression levels and the assembly of Pacman-positive P-bodies were enhanced upon Rbfox1 level increase achieved by *miR-980* loss and starvation stress (Fig. 6b, c), showing that the increase in Rbfox1 expression positively affects P-body formation.

To test whether Rbfox1 is also present in stress granules, we subjected animals to temperature stress, which led to formation of Hsp70/Hsc70-positive cytoplasmic granules (Fig. 6d, e). The Hsp70/Hsc70 complex is known to act as a stress chaperone for cellular proteins[54,55]. We found that in temperature-stressed ovaries, Rbfox1 colocalizes with granules that also immunostain for Hsp70/Hsc70 (Fig. 6d). These data show that similar to Rbfox1 localization in the nucleus, in the cytoplasm, its

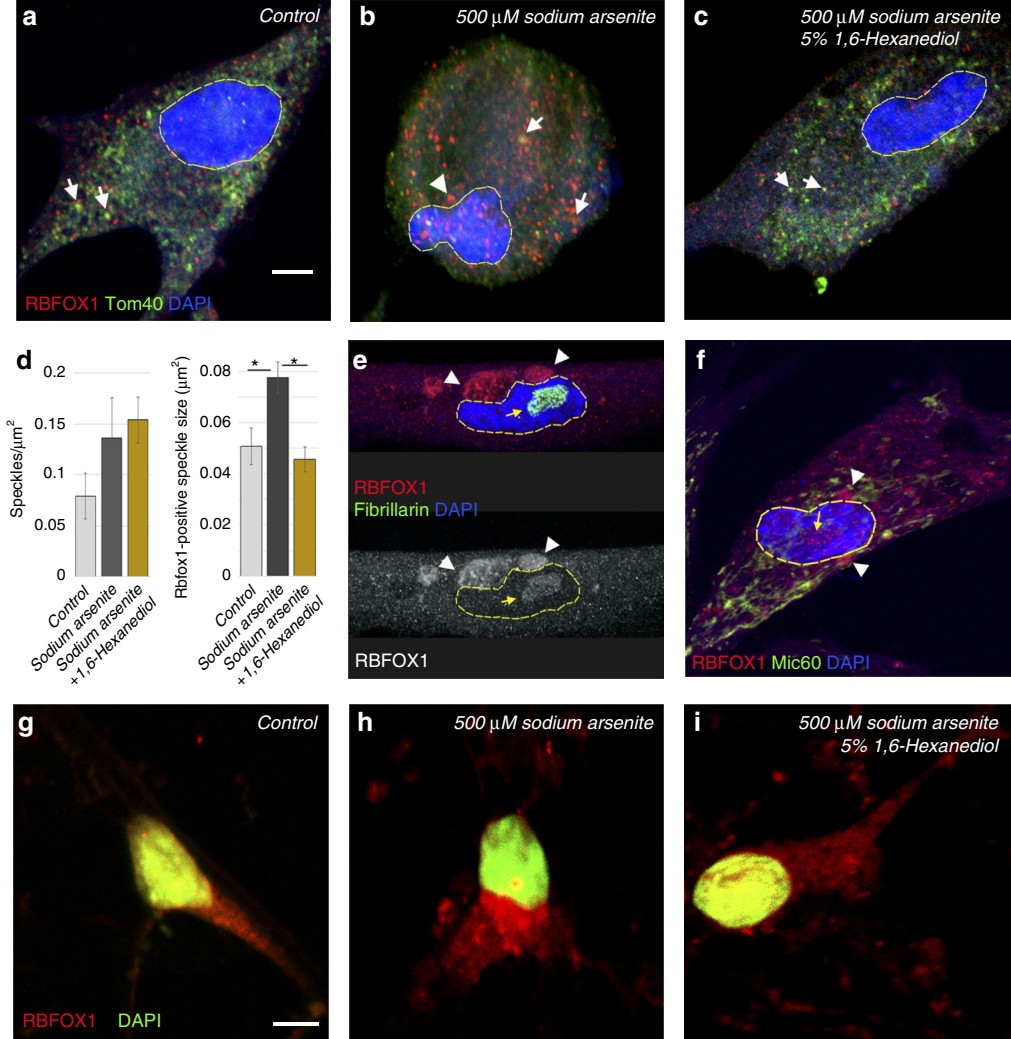

**Fig. 8** Conserved biological function of human RBFOX1 as a component of stress-induced subcellular compartments. **a** In human fibroblasts, RBFOX1 is detected in small cytoplasmic and nuclear speckles in normal conditions. **b** Treatment by the stress-promoting chemical sodium arsenite induces formation of larger RBFOX1-positive granules. **c** Moreover, application of 1,6-hexanediol, which dissociates non-amyloid RNP assemblies, causes partial disaggregation of RBFOX1-positive stress granules. **d** The number of RBFOX1-positive foci increases upon stress. The mean size of Rbfox1-positive speckles significantly increases upon sodium arsenite treatment, while additional application of 1,6-hexanediol reduces their size to control-like values. **e, f** Late passages of human fibroblast cell cultures have large aggregations of RBFOX1 protein in the cytoplasm and in the nucleus. The yellow arrow points to the nucleolus marked with Fibrillarin in **e**, white arrowheads point to mitochondria marked by Mic60 in **f**. **g** In human neurons, RBFOX1 shows similar behavior; smaller RBFOX1-positive granules are observed in control. **h** Their size is evidently increased upon stress induction; note the large nuclear and cytoplasmic RBFOX1-positive assemblies. **i**, Some of these granules show liquid state RNP granule characteristics, as they can be partially disassembled by 1,6-hexanediol. Note that RBFOX1 protein is re-localized from the nucleus to the cytoplasm upon sodium arsenite treatment (**g, h**). Quantification showed that in control (**g**), 69.5% had nuclear and cytoplasmic RBFOX1 and 30.5% expressed only cytoplasmic RBFOX1 ($n = 23$), while among sodium arsenite-treated cells (**h**), 48.6% had nuclear and cytoplasmic RBFOX1 and 51.4% cytoplasmic ($n = 37$). **a–c, e–f**, white arrows indicate associations of RBFOX1 with mitochondria, white arrowheads show RBFOX1 accumulation in the perinuclear area of the cytoplasm, yellow arrow points to the nucleolus. Nucleus is outlined. Images in **a–c, g–i** are maximum intensity projections of multiple Z-sections, and in **e, f** are single Z-sections. Scale bar 5 μm. DAPI marks nuclei. **d**, Speckle sizes are measured using the particle analyzer tool in ImageJ software. Mean speckle area and standard error of the mean are plotted in the bar graph. AVE ± SD is shown, experiments were done at least in triplicate, Student's $t$-test was applied for statistics, $*p \leq 0.05$

localization is also condition- and genotype-dependent. In response to different stresses or *miR-980* loss, more Rbfox1-positive granules could be detected in the cytoplasm (P-bodies and stress granules). However, Rbfox1 did not localize to the histone locus body in the nucleus that is responsible for processing of histone pre-mRNA[56] or stress-dependent cytoophidia[57,58] (Supplementary Figure 7), demonstrating that some degree of specificity exists for Rbfox1 inclusion into different membraneless compartments. In summary, these data demonstrate that depending on developmental stage and environmental

conditions, Rbfox1 can associate with membraneless organelles that have previously been described to have a liquid droplet character and to contain proteins regulating cytoplasmic RNA metabolism (Fig. 6f).

To understand the possible mechanism of how Rbfox1 interacts with various nuclear and cytoplasmic liquid-like organelles, we investigated the protein structure of the factors that we found in the Rbfox1-positive RNA granules. Using the SMART database (http://smart.embl-heidelberg.de/), we analyzed domain structures of the nuclear liquid compartment markers:

Fibrillarin (nucleolus), and Coilin (Cajal body) and also markers of the cytoplasmic RNA granules, Pacman (P-body), as well as Hsc70-3, Hsc70-4, Hsc70-5, and Hsc70Cb (stress granule). Notably, all these proteins are predicted to have multiple LCDs that vary in their length and composition (Supplementary Figure 8. Among these proteins, Fibrillarin, Hsc70-3, Hsc70-4, Hsc70-5, Hsc70Cb were also identified as proteins precipitated by b-isox in a liquid droplet component screen in *Drosophila* S2 cells[44]. Since LCD-containing proteins are known to promiscuously aggregate and self-aggregate via LCDs, we propose that Rbfox1 interacts with multiple LCD-containing proteins, which, in a concentration-dependent manner, promotes formation of various liquid organelles, globally regulating the cellular stress response.

**Human RBFOX1 contains LCDs and forms RNP aggregates.** We hypothesized that similarly to *Drosophila* Rbfox1, its human counterparts may also be linked to stress-induced RNP granule formation. To test this, we first analyzed human Fox family protein sequences for the presence of LCDs. Apart from the evolutionarily conserved RNA recognition motif (RMM, Fig. 7a), we identified LCDs in all human RBFOX proteins, albeit in lower numbers than in *Drosophila* Rbfox1 (Fig. 7b, Supplementary Table 5). Also, similar to *Drosophila*, various isoforms of human RBFOX1, RBFOX2, and RBFOX3, generated by alternative splicing, had different numbers of detected LCDs (Supplementary Table 5). To address their functionality, we utilized an in vitro model for RNA granule formation[44] and found that human

RBFOX1 protein from human fibroblast cell lysates precipitated with b-isox, validating that it can form RNP granules in vitro (Fig. 7c).

To test whether these granules are stress-inducible and of liquid state, we analyzed RBFOX1 subcellular distribution in human fibroblast cells, either unstressed or stressed by sodium arsenite (Fig. 8a–c). RBFOX1 was detected in small nuclear and cytoplasmic foci, the size of which was significantly increased by stress (Fig. 8d). In particular, we found that, similar to *Drosophila*, human RBFOX1 incorporates into larger protein assemblies in the nucleus and in the perinuclear area of the cytoplasm and many of these granules were mitochondria-associated (arrows, Fig. 8a–c). While the number of RBFOX1 granules did not change upon different conditions, their size increased in response to stress and was restored to normal as a result of 1,6-hexanediol treatment (Fig. 8c, d). It has been shown that stress granules contain a stable core structure surrounded by a more dynamic shell[55]. The fact that RBFOX1-positive subcellular compartments in human cells could only be partially disassembled by application of 1,6-hexanediol suggests a complex structure that possibly consists of both more stable and more liquid components (Fig. 8c). Moreover, in late passage cells, in addition to small nuclear and cytoplasmic foci, RBFOX1 accumulated in the nucleolus and formed large cytoplasmic aggregates even without additional stress induction (Fig. 8e, f).

Multiple age- and stress-related diseases, especially neurodegenerative disorders, are commonly coupled with defects in

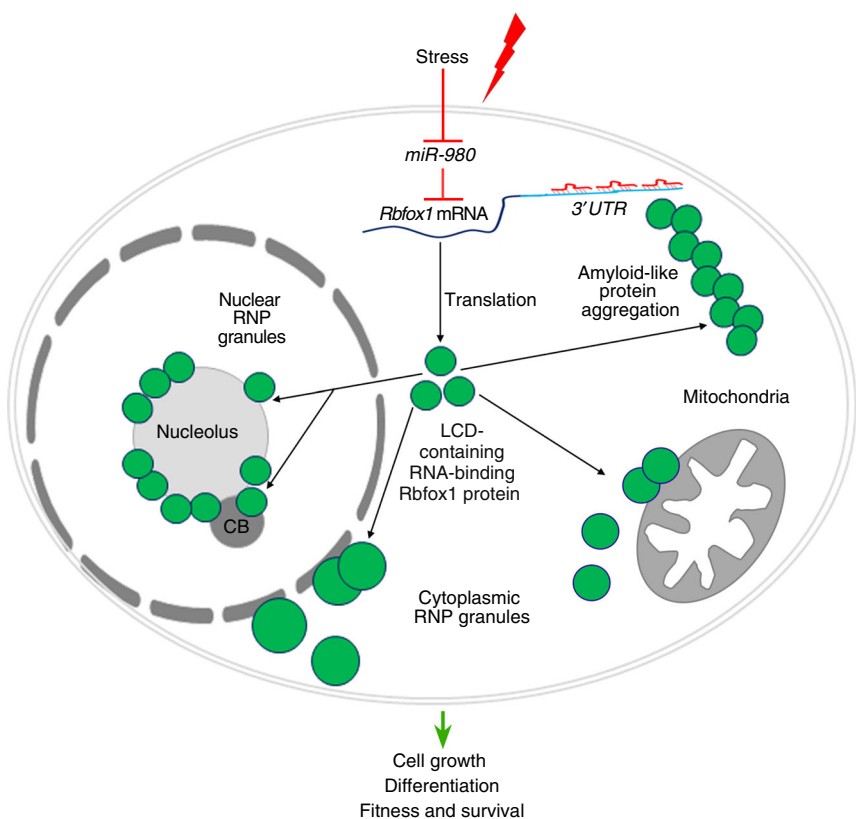

**Fig. 9** The *miR-980/Rbfox1*-mediated stress-responsive signaling cascade. Stress-dependent *miR-980* regulates Rbfox1, amending cell survival. *miR-980* can target only the portion of *Rbfox1* mRNAs with the alternative extended *3′UTRs*. This prevents the reduction of Rbfox1 levels below a certain threshold, which is detrimental for cellular homeostasis. Rbfox1 is a LCD-containing protein that phase-separates into liquid droplets and amyloid-like fibers in vivo and promiscuously joins nuclear and cytoplasmic RNP granules. Their assembly depends on Rbfox1 concentration and the presence of alternatively spliced isoforms. Since upon stress or *miR-980* loss, the levels of Rbfox1 generally increase, this, via LCD-mediated interactions, stimulates aggregation of other LCD-containing proteins that have specific functions in the nuclear and cytoplasmic liquid organelles. Since Rbfox1 levels globally influence RNA granule dynamics, we propose that the cell survival and differentiation defects observed as a result of Rbfox1 up- or down-regulation could be caused by Rbfox1 function as a liquid droplet-assembling LCD-containing protein

mRNA processing in neuronal cells[59]. Therefore, we investigated whether RBFOX1 shows a similar pattern of behavior and forms stress-inducible, soluble RNP granules in neuronal cells differentiated from human iPS cells. In human neurons, RBFOX1 was also detected in nuclear and cytoplasmic foci, assembly of which was augmented by stress and could be moderately disturbed by 1,6-hexanediol (Fig. 8g–i). These results suggest that in human neurons, RBFOX1 forms granules with differential substructures. In addition, we noted that upon stress, RBFOX1 has a tendency to be relocated from the nucleus to the cytoplasm (Fig. 8g–i). Intriguingly, bioinformatics predicts that human *Rbfox* mRNAs contain multiple miRNA binding sites (Supplementary Table 6), suggesting that they can be subjected to miRNA-based regulation.

## Discussion

Our results show that a miRNA, *miR-980*, and its target, Rbfox1, interact to have a profound effect on cell survival, growth, and differentiation, and subsequently, on organismal viability upon stress. Together they are integrated in a new stress-responsive signaling cascade (Fig. 9), in which alterations in miRNA expression in response to stress result in the upregulation of a protein with broad pro-survival properties. We found that Rbfox1 is a LCD-containing, RNA-binding protein, which associates with different nuclear, cytoplasmic and possibly mitochondrial RNP granules that have diverse functions including RNA processing, translation, splicing, storage, and decay. Our data demonstrate that Rbfox1 not only associates with these membraneless organelles, but also affects their assembly. The widespread inclusion of Rbfox1 to different subcellular organelles can be explained by the presence of multiple LCD sequences in the Rbfox1 protein. It has been shown that liquid droplet formation involves many promiscuous, weak interactions of different proteins with sticky LCDs in a concentration-dependent manner[5]. Our data show that in response to stress, alternative Rbfox1 isoforms are generated and Rbfox1 expression levels are increased. We propose that these alterations in Rbfox1 expression lead to redistribution of various Rbfox1 isoforms between the nucleus and cytoplasm and to the assembly of liquid droplet compartments, which occurs in an Rbfox1 concentration-dependent manner via Rbfox1 self-aggregation and aggregation with other LCD-containing proteins.

Furthermore, Rbfox1 phase-separation and RNP granule formation is concentration dependent, since Rbfox1 aggregates more often under conditions that increase its cellular levels, such as *miR-980* deficiency or stress, as well as Rbfox1 overexpression. This is in agreement with recent findings for the human FUS protein, where differences in protein concentration determine whether FUS forms liquid droplets, gels, or fibers[39,40,44]. In addition, Rbfox1 isoform specificity is adjusted by stress and *miR-980* deficiency. Therefore, it would be interesting to further study Rbfox1 isoform composition and functionality under stress and normal conditions. Since Rbfox1 differential expression influences the formation of different subcellular, membraneless organelles that have previously been shown to be involved in RNA processing, maintenance of its isoforms and precise cellular concentrations should be of great importance.

Human Rbfox proteins associate with a multimeric protein complex called a large assembly of splicing regulators (LASR), which contains hnRNP M, hnRNP H, Martin3, hnRNP U-like-2, hnRNP C, NF 110, NF45, and DDX5/17[45]. Half of these proteins (hnRNP M, hnRNP H, hnRNP C, DDX5/17) that complex with Rbfox were also identified as proteins precipitated by b-isox in the liquid droplet screen performed by the McKnight lab[44]. It has been demonstrated that Rbfox proteins are associated with the LASR complex in the specific nuclear fractions: particularly, in the insoluble high molecular weight fraction containing

chromatin, nuclear speckles, and unspliced RNA. Altered Rbfox expression changes activity of hnRNP M, thus affecting splicing through indirect interaction with the RNA[45]. An Rbfox splice variant lacking most of the RNA binding domain, but retaining the N- and C-terminal domains can block splicing activation, but not splicing repression, suggesting that domains other than RRM in Rbfox have important roles in regulation of RNA metabolism[60]. Recently, it has been shown that Rbfox interacts with LASR via its C-terminal domain containing repeated tyrosines that mediate higher-order assembly of Rbfox/LASR, which is essential for splicing activation by Rbfox[61]. These data imply that alternative splicing choices depend on the higher-order assembly of splicing regulators within the nucleus.

In addition to their nuclear function as splicing regulators, Rbfox proteins have cytoplasmic function. For example, a recent study showed that human Rbfox2 is included in the specific SGs that contain a subset of cell cycle-related genes[62]. Also, in the cytoplasm, Rbfox1 competes with miRNAs for the binding sites in the 3′UTRs. Thus, cytoplasmic Rbfox1 binding increases stability and translation of target mRNAs, especially mRNAs involved in cortical development and autism[31]. It is known that the components of the miRNA machinery in the cytoplasm are found in P-bodies, and our data show that Rbfox1 is involved in P-body assembly; thus, we propose that this completion between miRNAs and Rbfox protein for the binding sites in 3′UTRs occurs in these liquid organelles.

Our data also show that Rbfox1 is itself subjected to miRNA-based regulation. Importantly, cellular stress and miRNA deficiency have the same effect on Rbfox1, demonstrating that regulation by the stress-sensitive miRNA is used as a major mechanism securing proper Rbfox1 regulation upon stress. Interestingly, *miR-980* is shut down by any stress we tested: muscular dystrophy and aging[17], temperature, and starvation (this study). The resulting increase in Rbfox1 expression levels triggers RNA granule assembly, which consequently affects nuclear and cytoplasmic RNA biogenesis and promotes cell survival (Fig. 9). Recently, the *miR-980/Rbfox1* interaction in the adult *Drosophila* brain has also been shown to regulate memory via the modulation of calcium responses[20], further demonstrating the wide range of physiological processes dependent on *miR-980/Rbfox1* interaction.

Notably, we also found that only a part of *Rbfox1* mRNA could be subjected to *miR-980*-based regulation, since only a part of Rbfox1 transcripts have the extended 3′UTRs containing *miR-980* target sites. Messenger RNAs containing alternative 3′UTRs are very common and are often expressed differentially depending on the developmental stage and cell type, adding an extra level to mRNA diversification. Currently, the understanding of the mechanisms controlling the 3′UTR length in different cell types is largely missing, but recent studies suggest that 3′UTR length could have a profound effect on protein localization, levels, and functions[63,64]. Theoretically, by changing 3′UTR sequences, *Rbfox1* mRNA can be associated with different trans-regulatory factors that interact with the translation machinery, thus regulating its expression. For example, by varying the length of the 3′UTR, *Rbfox1* becomes more or less accessible for regulation by miRNAs, including *miR-980*. Now we have shown that the ability to have alternative 3′UTRs advances the mechanism of Rbfox1 regulation upon stress.

A recent model proposes that LCD-mediated protein aggregation, which results in the formation of functional, higher-order RNP assemblies that allow accumulation of RNA-binding proteins and RNA in defined space, can be beneficial for various regulatory processes[31,42]. We found that Rbfox1 protein can be promiscuously included in different types of liquid RNA granules, which normally regulate RNA metabolism. At the same time,

keeping Rbfox1 levels high for extended periods of time (for example, prolonged starvation or constant mitochondrial stress) could result in the formation of insoluble beta-amyloid-like fibers that are considered to be detrimental. Therefore, turning *miR-980* "on" in a timely fashion could potentially prevent toxic fiber formation.

Thus, our data show that the miRNA acts upstream of a crucial cellular pro-survival mechanism that allows temporal adjustment to stress. It is well established that many RNA-binding proteins linked to human diseases use protein aggregates as part of normally regulated physiological mechanisms controlling protein synthesis and RNA processing. Interestingly, *Drosophila miR-980* is related to the *miR-22* family in humans and that human RBFOX2 has a predicted binding site for *miR-22* (Supplementary Table 7). Therefore, it would be interesting to study whether regulation of any of these proteins also involves miRNAs, which could potentially allow development of new miRNA-based therapeutics for these debilitating disorders.

## Methods

**Drosophila genetics**. *Drosophila melanogaster* stocks were raised on standard cornmeal-yeast-agar medium at 25 °C. For ovary analysis, young flies (up to 1 week old) were kept on the same medium supplemented with additional yeast paste prior to dissection. For protein starvation, animals were placed on agar-apple juice plates for different periods of time as stated in the experimental procedures or figure legends. For temperature stress, animals were kept at 33 °C for one week.

The following alleles were used to deregulate *miR-980* expression: $CG3777^{NP3544}$ (DGRC, previously described[17] as a hypomorphic mutation in the *miR-980* gene and referred in the current study as *miR-980*$^{NP3544}$), *miR-980*$^{KO}$ [33], *miR-980*$^{Ex2}$, *miR-980*$^{Ex1-1}$, *miR-980*$^{Ex1-2}$, *miR-980*$^{Ex1-3}$ (identical mutations generated in current study, Supplementary Figure 1, see below), and *UAS-miR-980-dsRed*[65]. Also see the list of the used mutants in Supplementary Table 7.

The following fly strains were utilized to study Rbfox1/A2bp1 function: hypomorphic mutants caused by transposable element insertion *Rbfox1*$^{EN403}$ (a gift from M. Buszczak[66]) and *Rbfox1*$^{MI09677}$ (BDSC), and *Rbfox1*$^{RNAi}$ (VDRC, #34046) line were used for Rbfox1 downregulation, while *UAS-Rbfox1* for *Rbfox1 RE* isoform overexpression[35]. *Rbfox1 RE* isoform has a shortened 3'UTR that lacks *miR-980* binding sites. To visualize Gal4 expression patterns, a *UAS-CD8GFP UAS-nLacZ* line was used (a gift from Frank Hirth). The *park*$^1$ mutant (BDSC) was used as an in vivo system for cellular stress[48]. The wild type *Oregon R* and *w*$^{1118}$ (BDSC) strains were used as controls.

To rescue the *miR-980* loss of function phenotype by Rbfox1 downregulation, we introduced one copy of Rbfox1 mutation (*Rbfox1*$^{EN403}$/+) in the *miR-980*$^{Ex1-2}$ homozygous background.

**Generation of *Drosophila miR-980* mutants**. To generate *miR-980* loss of function mutants, the excision of the P-element *P GawB CG3777*$^{NP3544}$ (DGRC) was induced. For this, female flies of the genotype *w,P GawB CG3777*$^{NP3544}$;+;+ were crossed with transposase-expressing males *w/>;+;Sb*$^1$,*P ry[+t7.2] = Delta2–3 99B/ TM2* (DGRC). Male progeny (single flies) of the genotype *w,P GawB CG3777*$^{NP3544}$/>;+;Sb$^1$,P ry[+t7.2]=Delta2–3 99B/+were crossed to *y,w,C1(DX)/>; +;+*females. From the progeny of this cross, single males with excised P-element identified by white eye color (*w, P*$^{Ex}$/>;+;+) were crossed to females of the genotype *y,w,C1(DX)/>;+;+* and stock lines were established. Next, male flies were used to test expression levels of *miR-980* and CG3777 by qRT-PCR. Then, verified *miR-980* mutant males (*w,P*$^{Ex}$/>;+;+) with no changes in the expression of CG3777 were crossed to females carrying an FM7 balancer chromosome to obtain homozygous stocks. Selected *miR-980* loss of function mutants (*miR-980*$^{Ex2}$; *miR-980*$^{Ex1-1}$; *miR-980*$^{Ex1-2}$; *miR-980*$^{Ex1-3}$) were used for mapping of *miR-980* mutations.

**Mapping of *miR-980* mutations**. To map generated *miR-980* loss of function mutants, genomic DNA samples from males of the genotypes (1) *w*$^{1118}$, positive control, (2) *CG3777*$^{NP3544}$, negative control and (3) *miR-980* loss of function mutants were isolated using QIAGEN DNEasy Blood and Tissue Kit, following the manufacturer's protocol. Primers flanking the P-element (NP3544) insertion region were designed using NCBI/Primer-BLAST to characterize excision of the *P GawB CG3777*$^{NP3544}$. Fw1: TGTAACAAAACTTTTAGGATACCCC, Fw2: CGTAAGCCCTTCACAAGGCA, Rev: TTCCCGCGTGTGAAATACGA (see Supplementary Figure 1). DNA regions were amplified using QIAGEN HotStarTaq Master Mix Kit, following the manufacturer's protocol and using a T100 Thermal Cycler (BioRad). PCR products were separated on a 2% agarose gel. PCR amplification products were purified using the QIAquick PCR Purification Kit and sequenced by Eurofins MWG Operon using Fw: AATTTGTTCCCTGGGCTTGC and Rev: TTCCCGCGTGTGAAATACGA primers (for sequencing results see Supplementary Figure 1 legend).

**Generation of follicle cell clones**. To generate follicle cell clones expressing *Rbfox1*, *Rbfox1*$^{RNAi}$, or *miR-980*, females of the genotype *hsFlp; act>CD2>Gal4 UAS-GFP* were mated with males of genotypes *UAS-miR-980* or *UAS-Rbfox1*. Then, 1-5 days old progeny were 1 h heat shocked in a 37 °C water bath for 2 consecutive days. On the third day after the first heat shock, ovaries were dissected and immunostained for analysis. GFP-marked cells expressing the transgene of interest were compared with neighboring control cells (GFP-negative). To exclude the influence of GFP on studied phenotypes, we analyzed ovaries of heat shock-treated progeny obtained from crossing *hsFlp; act>CD2>Gal4 UAS-GFP* females and *w*$^{1118}$ males. To induce starvation stress, flies were kept on agar-apple juice plates for 2 days post heat shock treatment.

**Quantification of the ovarian protein starvation response phenotype**. To quantify the ovarian response to protein starvation, flies were deprived of a protein source for 10 days. Then, their ovaries were dissected and stained with DAPI. Ovarioles that stopped egg production prior to stage 7 were considered to have a normal response to protein starvation stress, while ovarioles containing stage 8–10 egg chambers were scored as unresponsive to starvation stress. Then, the percentage of ovarioles with no response to protein deprivation stress was calculated.

**Cell culture experiments**. *Drosophila* S2R+ cells (DGRC, stock#150) were maintained in Schneider's medium containing 10% of heat-inactivated fetal bovine serum (GE Healthcare), 100 units/ml penicillin, and 100 μg/ml streptomycin (Gibco). For immunostaining, cells were grown on PLL-coated coverslips (Neuvitro). For immunostaining and western blotting experiments, S2R+ cells were transfected with ActP-Flag-Rbfox1-PE (a gift from Michael Buszczak) by using the Effectene transfection reagent (QIAgen) and following the manufacturer's protocol. Rbfox1 protein localization under normal and stress conditions was visualized either with antibodies against Rbfox1 or with anti-Flag immunostaining.

Human material was used in accordance with ethical guidelines and the Helsinki Declaration. Subjects gave informed consent regarding generation and use of iPS cells. Human fibroblast cells and human neuronal cells differentiated from iPS cells were kindly provided by Ahmed Mansouri and Hannelore Ehrenreich, respectively.

Human fibroblasts (IMR HFB) were maintained on gelatin-coated dishes in DMEM 1 g/l glucose (1 × Glutamax) supplemented with antibiotic (1 × Antimycotic) and 0.1 mM non-essential amino acids (all from Invitrogen, Life Technologies, Carlsbad, CA, USA) and 10% fetal bovine serum (FBS; Sigma-Aldrich Co.).

Human fibroblasts were reprogrammed using a nonintegrative RNA-based virus to induce the expression of four reprogramming factors: OCT4, SOX2, KLF4 and cMYC (CytoTune-iPS 2.0 Sendai Reprogramming Kit, Life Technologies GmbH). After transduction, IPS were adapted to a feeder-free culture system (Matrigel matrix, Corning, Wiesbaden, Germany), cultured in TeSR-E8 medium (STEMCELL Technologies SARL, Cologne, Germany) and subjected to neuronal differentiation[67]. Neuronal cells were used for experiments at 65-day post-neuronal differentiation, and their neuronal characteristics were confirmed by anti-Synapsin 1 immunostaining.

Stress-induction was applied for 1 h prior to immunostaining. Starvation stress was induced by keeping cells in PBS, and redox stress was induced by adding 500 μM Sodium (meta) arsenite (Sigma Aldrich) to the medium. To disrupt non-amyloid assemblies formed by LCDs-containing Rbfox1, 1,6-hexanediol (an aliphatic alcohol that disrupts hydrophobic interactions) was added to the cell culture medium in 5% concentration. 1,6-hexanediol specifically triggers dissociation of non-amyloid assemblies while it does not affect amyloid aggregations[5].

**Luciferase reporter assay**. To generate the *Rbfox1-3'UTR* sensors, a 381 bp region (P1) and a 396 bp region (P2) containing the putative *miR-980* binding sites were amplified from genomic *Drosophila melanogaster* DNA by polymerase chain reaction (Supplementary Figure 2D). The following primers that included enzymatic digestion sites for NotI and XhoI (underlined) were used:

P1 forward – <u>TACGT</u>GCGGCCGCGGGAAAGTAACAATGCAGCC, P1 reverse – <u>CCACC</u>ATGGCTCGAGTTTGTTTGTTGCTTTGCTGC, P2 forward – <u>TACGT</u>GCGGCCGCACCCCCAGCACAGGAAAGGAAA, P2 reverse – <u>CCACC</u>ATGGCTCGAGAAAACAAAAGCTGGCGCTGGCT. Fragments were subsequently cloned into NotI and XhoI restriction sites downstream of the *Renilla* luciferase gene in the psiCHECK-2 vector (Promega). *Drosophila* S2R + cells (DGRC) were seeded in a 96-well cell culture plate one day after splitting 1:6. Cells were transfected using the Effectene transfection reagent (QIAgen) with the following amounts: 50 ng of empty psiCHECK-2 or 50 ng of *psiCHECK-2-Rbfox1-3'UTR-P1* or *psiCHECK-2-Rbfox1-3'UTR-P2*, 25 ng of *actGal4*, 50 ng of the *pUAST-miRNA* plasmid. We used plasmids expressing *miR-980*, *miR-966* and *miR-278* miRNAs. Bioinformatic predictions did not detect putative binding sites for *miR-966* and *miR-278* miRNAs in either P1 or P2 regions. Approximately 72 h after transfection, the S2R+ cells were subjected to the Dual-Glo luciferase assay (Promega) by following the manufacturer's protocol. Plates were analyzed on a Wallac 1420 luminometer. The *Renilla* luciferase activity was determined with the empty psiCHECK-2, psiCHECK-2-Rbfox1-3'UTR-P1 and psiCHECK-2-Rbfox1-3'

*UTR-P2* plasmids in the presence and absence of miRNAs. To determine the effect on miRNA presence on the reporter expression, first, the *Renilla* to *Firefly* luciferase luminescence value ratios were calculated and then, the values in the presence of miRNA were normalized to the values measured with no miRNA-expressing plasmid and to the values of plasmid without *Rbfox1-3′ UTR* regions (psiCHECK-2).

All transfections were done in triplicate to determine an average and standard deviation of the data. The Student's two-tailed t-test was used for statistical analysis.

**RNA preparation and real-time quantitative PCR.** RNAs were extracted from flies using the TRIzol reagent (Invitrogen), followed by reverse transcription using the High Capacity cDNA Reverse Transcription kit (Applied Biosystems) following the manufacturer's protocol. *miR-980* levels were tested with 2S rRNA as an endogenous control for q-PCR using TaqMan® MicroRNA Assays. *Rbfox1* and *CG3777* mRNA levels were tested with *RpL32* and/or *Actin 5c* as an endogenous control for q-PCR using FastSYBR® Green master mix on a Step One Plus 96 well system (Applied Systems). Primers were used as follows: *RpL32* forward — AAGATGACCATCCGCCCAGC; *RpL32* reverse — GTCGA-TACCCTTGGGCTTGC; *Rbfox1 long 3′UTR* forward – AAATCGAACCCAA-CAATTGCC; *Rbfox1 long 3′UTR* reverse – GGGGGTGGGCCAAAAAGTATA; *CG3777* forward – CCGAGGAGGGTATAGTGGTTGC; *CG3777* reverse – AGCTGGAGATTCCTCGCTGG. All reactions were run in triplicate with appropriate blank controls. The threshold cycle (CT) is defined as the fractional cycle number at which the fluorescence passes the fixed threshold. The $\Delta$CT value was determined by subtracting the average *RpL32* or *Act5c* CT value from the average *Rbfox1* and *CG3777* CT value or the average 2S rRNA CT value from the average *miR-980* CT value. The $\Delta\Delta$CT value was calculated by subtracting the $\Delta$CT of the control sample from the $\Delta$CT of the suspect sample. The relative amount of mRNA was then determined using the expression $2^{-\Delta\Delta CT}$ method. The two-tailed Student's t-test was used to determine statistical significance of the relative amount of miRNA or mRNA. Presented error bars represent the standard deviation from the mean.

**LNA in situ hybridization.** For the LNA in situ hybridization, miRCURY LNA probe was ordered from Exiqon (dme-miR-980 product # 21455-15). Tissues were dissected, fixed in 4% formaldehyde in PBS, dehydrated and stored at −20 °C overnight or longer. Tissues were then rehydrated, treated for 10 min with Proteinase K solution (50 µg/ml Proteinase K in 50 mM Tris–HCl pH 7.5, 50 mM EDTA), and post-fixed in 4% formaldehyde in PBT for 30 min. Tissues were prehybridized for 1 h in hybridization buffer (50% formamide, 25% 20 × SSC, 5 mg/ml Torula yeast RNA (Sigma), 0.1% Tween 20). Hybridization was carried out overnight at 60 °C with a 40 nM probe concentration in hybridization buffer. Post hybridization washing was done at 62 °C with three subsequent 20 min washes with hybridization wash solution (no yeast RNA), 50/50 v/v hybridization wash solution/PBT, PBT. Tissues were then blocked for 1 h in western Block (Sigma). Anti-DIG conjugated with Alkaline Phosphatase was diluted 1:2000 in block and incubated with tissues overnight at 4 °C. Colorimetric detection was done with 10 µl/ml NBT (Roche) in staining buffer (0.05 M Tris pH 9.5, 0.05 M MgCl₂, 0.1 M NaCl, 0.1% Tween 20) for ~30–45 min.

**In situ hybridization for total polyA mRNA localization.** Prior to the in situ hybridization procedure, S2R+ cells were plated on PLL-coated coverslips (Neuvitro) in 6-well plates. In 24 h cells were transfected with Flag-tagged Rbfox1-expressing vector coding for Rbfox1-PE isoform. In 72 h post-transfection medium was removed and cells were treated for 1 h with 500 µM Sodium arsenite (Sigma Aldrich). After treatment cell were rinsed once with PBS and fixed for 10 min in 4% formaldehyde (prepared fresh in 1× PBS, pH7.4). After formaldehyde was removed, cells were incubated in 100% cold (kept at −20 °C) methanol, for 10 min in 70% ethanol, and for 5 min with 1 M Tris pH8.0. While incubating cells with Tris, 100 µM 5′Cy3-Oligo-dT(30) probe (Microsynth) was diluted 1:1000 in the hybridization buffer containing 1 mg/ml of yeast tRNA, 0.005% BSA (AppliChem), 10% Dextran sulfate (Sigma Aldrich), 25% Formamide (VWR), and 2 × SSC buffer prepared in DEPC-treated water. After Tris was removed from plates, the hybridization buffer with the probe was added. Plate was sealed in the plastic box with wet paper towels (to maintain humidity) and incubated at 50 °C overnight. After hybridization, samples were washed once with 4×SSC and ones with 2×SSC. After this, cells were immunostained with anti-Flag antibodies to detect Rbfox1 protein. Cells were incubated in primary mouse anti-Flag antibody (diluted 1:1000 in 2xSSC containing 0.1% of Triton-X-100) for 1 h at the room temperature. Then, samples were washed 3 times with 2×SSC and incubated with secondary antibody (diluted 1:1000 in 2xSSC containing 0.1% of Triton-X-100) containing DAPI for 1 h at the room temperature. After staining, samples were washed twice with 2×SSC and mounted on a slide with a drop of 70% glycerol.

**Immunohistochemistry.** Ovaries were dissected in phosphate-buffered saline (PBS) and fixed while shaking on a nutator for 15 min in PBS containing 4% Formaldehyde. Next, they were rinsed with PBT (PBS/0.2%, Triton X-100) four times (15 min, each rinse) and blocked in PBTB (PBT, 0.2% BSA, 5% Normal Goat Serum) for 1 h at room temperature. The tissue was incubated with primary antibodies overnight at 4 °C. The next day they were rinsed with PBT four times (15 min, each rinse) and blocked in PBTB for 1 h at room temperature. The ovaries were then incubated in secondary antibodies overnight at 4 °C. The next day they were rinsed with PBT (4 times, 15 min each rinse) and stained with DAPI (1 mg/ml in PBT) for 10 min. Finally, they were washed with PBT twice (5 min, each wash) and dissected onto slides in 70% glycerol, 3% NPG, 1× PBS.

In *Drosophila* S2R+ cells, human fibroblasts, and human neuronal cells differentiated from iPS cells, immunostaining was accomplished as following: cells grown on coverslips were washed 2 times with PBS, fixed with 4% formaldehyde for 15 min, washed 3 times with PBT, and blocked with PBTB for 1 h at room temperature. Then, slides were incubated for 1 h at room temperature with the primary antibodies diluted in PBTB. Afterwards, cells were washed for three times with PBT and incubated with the diluted secondary antibodies for 1 h at room temperature. Then, samples were washed two times with PBT, incubated for 10 min with DAPI, and washed with PBT. Samples were mounted by laying the cell-containing coverslip down on the slide with a drop of 70% glycerol, 3% NPG, 1× PBS.

**Antibodies.** To detect *Drosophila* proteins the following antibodies were used: monoclonal mouse anti-Notch intracellular domain 1:20; anti-Lamin C (LamC) 1:20; anti-Lamin D (LamD) 1:20; anti-Adducin (Add) 1:20; anti-Armadillo (Arm) 1:20; anti-Cut 1:20 (DSHB); anti-ATP synthase (ATPsyn) (ATP5A) 1:500 (Abcam); monoclonal rat anti-E-Cadherin (Cad) 1:50 (DSHB); polyclonal rabbit anti-Lsm11 1:1000 (gift from Joseph Gall); anti-Myc 1:500 (gift from Peter Gallant); anti-Atg8 1:1000 (gift from Katja Köhler); anti-Fibrillarin 1:1000; anti-Caspase3 (Cas3) 1:200; anti-Tom40 1:500 (Abcam); anti-CTP Synthase (CTPsyn) 1:1000 (Santa Cruz); anti-Flag 1:160 (Sigma-Aldrich); anti-Hsc70/Hsp70 1:1000 (Enzo Life Science); anti-Mic60 1:500 (gift from Stefan Jakobs); anti-Pacman 1:1000 (gift from Sarah Newbury); polyclonal guinea pig anti-Coilin 1:1000 (gift from Joseph Gall); and polyclonal chicken anti-GFP 1:2000 (Invitrogen).

To detect *Drosophila* Rbfox1 protein, polyclonal rabbit anti-Rbfox1 (Rbfox R)[35] 1:500 and polyclonal guinea pig anti-Rbfox1 (Rbfox1 GP)[19] 1:500 were used. To detect human Rbfox1 protein the monoclonal mouse anti-RBFOX1 antibodies 1:500 (Millipore) was applied. The polyclonal guinea pig anti-Rbfox1 antibodies were generated against 84-186 residue of Rbfox1 PE isoform and therefore should recognize LCDs 2, 3, 6 and 7, while polyclonal rabbit anti-Rbfox1 antibodies were raised to full length Rbfox1 PE isoform and therefore, in addition, they recognize LCDs 8, 9,10, 13, RRM and coiled-coiled domains (Supplementary Figure 5). Based on the predicted by FlyBase Rbfox1 isoform sequences (Supplementary Figures 2 and 4), both antibodies should recognize all Rbfox1 isoforms.

Secondary antibodies used were Alexa 488, 568, or 633 goat anti-mouse, anti-rabbit, anti-guinea pig, anti-chicken (1:500, Molecular Probes); goat anti-rat Cy5 (1:250, Jackson Immunoresearch); and HRP-conjugated goat anti-guinea pig and anti-mouse 1:5000 (Sigma-Aldrich).

**Fluorescence recovery after photobleaching.** Fluorescence recovery after photobleaching (FRAP) measurements were performed on a Zeiss LSM700 confocal laser-scanning microscope. Live *Drosophila* ovaries expressing an Rbfox1::GFP fusion protein (*Rbfox1^{CC00551}/TM3* strain) were placed on a slide with a drop of Halocarbon oil 27 (Sigma Aldrich), covered with a coverslip, and immediately subjected to the experimental procedure. To photobleach Rbfox1-positive speckles, a 20-millisecond laser pulse was applied to selected regions of interest (ROI). Confocal images were obtained before and after photobleaching with 1 s intervals.

To determine the raw FRAP data, the average intensity of each bleached Rbfox1-positive speckle (ROI1) was measured for each time point by ZEN 2011 image processing software. Then, background subtraction was performed by subtracting the average intensity of non-bleached background area (ROI2) from the average intensity of ROI1 for each time point. Acquisition photobleaching was corrected for the gradual fluorescence decrease in the prebleach or postbleach images by dividing the intensity value of bleached Rbfox1-positive area by the average fluorescence of non-bleached bright control area (ROI3). After background subtraction and correction, fluorescent intensity values were normalized to one by dividing the intensity of all time points by the average prebleach intensity. Then, the mobile/immobile fractions were determined as following: $F_M = (I_E - I_0)/(I_I - I_0)$, where $F_M$—mobile fraction, $I_E$—end value of the recovered fluorescence intensity, $I_0$—first postbleach fluorescence intensity, and $I_I$—initial fluorescence intensity. The immobile fraction ($F_I$) was determined as $1 - F_M$. Next, the recovery half-life ($t_{half}$) was determined. Half-life was defined as the time from bleaching to the time point at which the fluorescence intensity reached half ($I_{1/2}$) of the final recovery intensity ($I_E$). Assuming a relaxation (exponent-governed) recovery process, we fitted our data with a model $y(t) = A(1 - \exp(-x/tau))$, where A is the final value of the recovery intensity ($I_E$), tau (the characteristic time of the process) is the fitted parameter, and $t$ is the time after the bleaching pulse. The last can be converted to the half-life time as $t1/2 = tau*\ln(2)$. The errors have been obtained from the 95%

confidence bounds returned by the fitting routine. MATLAB software was used for curve fitting. Scattered blots were build using OriginPro 9.1.

**Thioflavin T staining**. Thioflavin T was used as a histological probe to detect the formation of amyloid fibrils. Ovaries were dissected, fixed, and immunostained as described above and then incubated with a filtered solution of 0.1 N HCl and 0.05% Thioflavin T (Sigma-Aldrich) in 1xPBS for 20 min.

**Microscopy and image analysis**. Images of ovaries and S2R+ cells were obtained using a Zeiss LSM700 confocal laser-scanning microscope. Protein expression patterns and phenotypes were analyzed from confocal images taken in a z-stack (1 μM step) of ovarioles/egg chambers/follicle, S2R+, or human cells. Reproducible results are presented. For the whole fly, pupae, and larvae imaging, a SteREO Discovery V8 stereomicroscope with Zeiss AxioCam ICc1 camera was used. Images were processed with ZEN 2009 and Adobe Photoshop software.

The immunostaining intensities for anti-Atg8 and anti-Cas3 were measured by ZEN 2011 software in Rbfox1-expressing follicular epithelium cells (GFP-positive) and compared to the neighboring control cells (non-clonal, GFP-negative) in the same egg chamber (intensity per cell). Values are normalized to the control and are presented as AVE ± AD. At least 25–50 cells from 3 to 5 clonal egg chambers of comparable developmental stages were analyzed. The immunostaining intensities for anti-Myc, anti-Cad, anti-Arm and anti-Rbfox1 were measured per GFP-positive clone and compared to the same size neighboring control area (non-clonal, GFP-negative) within the same egg chamber (intensity per area). AVE ± AD values obtained from 5 to 17 clonal areas of comparable stage egg chambers are presented normalized to the control. The efficiency of Notch signaling activation was evaluated as following: first, the intensity of anti-Notch staining was measured in Rbfox1-overexpressing or Rbfox1$^{RNAi}$-expressing clones (GFP-positive), and also in two neighboring control areas of the same size within the same egg chamber. Obtained values were normalized to one of the control areas and are presented as absolute values. Measurements obtained from 11 clones for each experiment were averaged and presented as AVE ± AD. The two-tailed Student's t-test was used for statistical analysis. Dynamics of Cut expression in developing follicle cells was quantified for Rbfox1-overexpressing (n = 19) and Rbfox1$^{RNAi}$-expressing (n = 43) clones and compared to the control (neighboring non-clonal cells) within the same egg chamber. The frequency of Cut expression was evaluated in st.6-10a and in st.10b-12 clonal egg chambers. The two-tailed Student's t-test was applied for statistics.

The numbers of Coilin-positive speckles were quantified per cell from z-sections of Rbfox1-expressing clones and compared to the numbers observed in non-clonal neighboring cells within the same egg chamber. Coilin and Rbfox1 colocalization was scored from z-sections of st.7-8 egg chambers. Nucleolar types were quantified from maximum intensity projections of st.9-10 egg chambers immunostained with anti-Fibrillarin. Chi square and Two-Way Tables analysis was applied for statistics. The Rbfox1 and ATPsyn colocalization was counted from z-sections per st.9-10 follicle cell (n = 110 and 154 cells for Control and park[1] mutant, respectively). Two-tailed Student's t-test was applied for statistical analyses.

**Western blotting**. To quantitate Rbfox1 protein levels in wild type (Oregon R), miR-980 mutant, and Rbfox1-overexpressing (hsFlp; act>CD2>Gal4 UAS-GFP/ UAS-Rbfox1RE, heat shocked 1 h for two consecutive days at the pupal stage) ovaries, animals were kept either at normal (apple juice plates with yeast paste supplement) or protein starvation (apple juice only plates) food conditions for 3 days prior to dissection. Ovaries were rapidly dissected in ice-cold PBS and homogenized in lysis buffer containing 50 mM Tris (pH 7.5), 125 mM NaCl, 5% glycerol, 1% NP40, 0.5% Sodium-deoxycholate, 0.1% SDS, 1.5 mM MgCl$_2$, 1 mM DTT, 25 mM NaF, 1 mM Na$_3$VO$_4$, 2 mM EGTA, 1 mM EDTA, and 1× protease inhibitor cocktail. Protein extracts were separated by SDS–PAGE and subjected to standard western blotting for Rbfox1 detection with polyclonal guinea pig anti-Rbfox1 (1:5000) antibodies. Gels were analyzed with ImageJ. Rbfox1 isoform band intensities were measured and normalized to Actin levels and to the control at normal conditions. Data are presented as relative to control mean band intensity and standard error of mean quantified from 3 to 5 independent biological replicates. Two-tailed Student's t-test was applied for statistics. All uncropped western blots can be found in Supplementary Figure 9.

**Biotinylated isoxazole-mediated precipitation**. Biotinylated isoxazole (b-isox)-mediated precipitation of Drosophila Rbfox1 or human RBFOX1 protein was performed on cell lysates from Drosophila ovaries of control (hsFlp; act>CD2>Gal4 UAS-GFP/+) and Rbfox1-overexpressing (hsFlp; act > CD2 > Gal4 UAS-GFP/UAS-Rbfox1RE) animals and human fibroblasts and neuronal cells. In Drosophila ovaries, Rbfox1 overexpression was induced in follicle cell clones by applying 1-h heat shock for two consecutive days 3 days prior dissection. Control animals were treated analogously. Ovaries were rapidly dissected in ice-cold PBS and homogenized in lysis buffer containing 50 mM Tris (pH 7.5), 125 mM NaCl, 5% glycerol, 1% NP40, 0.5% Sodium-deoxycholate, 0.1% SDS, 1.5 mM MgCl$_2$, 1 mM DTT, 25 mM NaF, 1 mM Na$_3$VO$_4$, 2 mM EGTA, 1 mM EDTA, and 1× protease inhibitor

cocktail. Human fibroblast cells were first briefly washed with PBS and then collected in the same lysis buffer. Samples were tumbled at 4 °C for 20 min, and cell lysates were precleared with a centrifugation at 14000 × g for 15 min at 4 °C. B-isox (Sigma-Aldrich) dissolved in DMSO was added to cell lysates to 10, 30, and 100 μM final concentrations. DMSO was added to the mock control. For b-isox-mediated protein precipitation, samples were rotated gently at 4 °C for 1 h. The incubated reaction was then spun at 14000 × g for 15 min to pellet the precipitate. The pellet was washed twice in lysis buffer and resuspended in 2 × SDS loading buffer. The supernatant and precipitate were each separated by SDS–PAGE and subjected to standard western blotting to identify Rbfox1/Fox-1 among b-isox-precipitated proteins. Polyclonal guinea pig anti-Rbfox1 (1:5000) antibodies were used for Drosophila Rbfox1 detection, and monoclonal mouse anti-Fox-1 (1:1500) was used for detection of human RBFOX1 protein.

**Semidenaturing detergent-agarose gel electrophoresis**. Semidenaturing detergent-agarose gel electrophoresis (SDD-AGE) was performed as previously described[46] with minor modifications. In brief, a 1.5% agarose gel containing 0.1% SDS was cast using a standard gel casting tray for horizontal DNA electrophoresis and placed into a gel tank containing 0.1% SDS and 1 × TAE buffer.

To prepare the cell lysates for SDD-AGE, 15 female adult flies were homogenized in 70 μl of lysis buffer (50 mM Tris, pH 7.5; 125 mM NaCl; 0.5% NP-40; 0.25% sodium-deoxycholate; 25 mM NaF; 1 mM Na$_3$VO$_4$; 1 mM DTT; 2 mM EGTA at pH 8.0; 1 mM EDTA at pH 8.0; 1× protease inhibitor cocktail) and then incubated 10 min at room temperature. The lysates were cleared of cellular debris by centrifugation at 4000 rcf for 20 min. Protein concentration was measured using Bradford Reagent (Sigma-Aldrich). 300 μg of cell lysates were added to 4× sample buffer (2 × TAE; 20% glycerol; 8% SDS; 1% bromophenol) to a final 1× concentration.

In SDS-containing buffers, amyloid-like aggregates are stable at room temperature but can be disrupted by boiling. Therefore, samples were incubated for an additional 10 min either at RT or at 95 °C as negative control. Electrophoresis was performed at 4.5 V/cm-gel length for 6 h. Protein transfer was performed to a nitrocellulose membrane for 12-16 h using capillary action. The membrane was used for western blot analysis.

**Bioinformatic prediction analyses**. To identify predicted miR-980 targets in Drosophila, TargetScanFly Release 6.0 was used, and to identify miRNAs that putatively target human RBFOX family proteins, TargetScanHuman Release 7.0 was applied. Drosophila Rbfox1 protein functions were predicted by https://www.predictprotein.org. For protein sequence alignment, ClustalX 2.1 was used, and for protein sequence analysis and domain identification, Simple Modular Architecture Research Tool (SMART) and MitoFates were applied.

**Data availability**. The authors declare that all data supporting the findings of this study are available within the article and its supplementary information files or from the corresponding author upon reasonable request.

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

## Acknowledgements

We would like to thank Simon Alberti, April Marrone, Travis Carney, Yuanbin Xie, Ömer Çiçek, and all Shcherbata lab members for critical reading of the manuscript and helpful suggestions; Michael Buszczak, Lingadahalli Subrahmanya Shashidhara, Sarah Newbury, Frank Hirth, Joseph Gall, Peter Gallant, Katja Köhler, Roland Dosch, Herbert

Jäckle, and Stefan Jakobs for fly strains, vectors and antibodies; Ahmed Mansouri, Sharif Mahsur, and Hannelore Ehrenreich for human fibroblast and neuronal cells; and Anne Spang, Nicholas Proudfoot, Jeffrey A. Chao, Maria Carmo-Fonseca, and Simon Alberti for great discussions of the data; and the Max Planck Society for funding.

## Author contributions

M.M.K. and H.R.S. designed and performed experiments, analyzed data, prepared figures, and wrote the manuscript.

## Additional information

**Competing interests:** The authors declare no competing financial interests.

