## [Peer Review File · Nature Communications]

Reviewer #1 (Remarks to the Author):

This manuscript examines expression and localization of Rbfox1 as a function of stress conditions during *Drosophila* ovary development, demonstrating important new data that (1) Rbfox1 can aggregate into membraneless, subcellular RNP granules in a miR-980- and stress-dependent manner, and (2) Rbfox1 expression regulates follicle epithelial differentiation and early embryo survival. The authors also find that Rbfox1 expression is negatively regulated by miR-980, as shown earlier by others in studies of *Drosophila* brain, and show that the consequences of Rbfox1 mis-expression can be partially overcome by manipulating miR-980 expression. This data is used to propose a novel concept that miR-980 and its target Rbfox1 “interact to have a profound effect on cell survival, growth, and differentiation, and subsequently, on organismal viability upon stress.” The authors conclude that these interactions represent part of a new stress-responsive signaling cascade. Overall the data is quite interesting and provocative, especially the proposal that Rbfox1 is a pro-survival protein that contributes to slowing down metabolism under unfavorable conditions in an attempt to sustain the cells during period of temporary stress. However, there are many questions remaining to be answered.

1. Rbfox1 expression and localization: a. The authors demonstrate expression of a wide array of protein isoforms, some of which exhibit different abundances under stress. If not for earlier studies showing complex splicing patterns that can yield many proteins, this data would be quite remarkable. It is still confusing that numerous isoforms are present in the gels of Fig. 1E and 2A (supernatant), but predominantly one isoform is precipitated by b-isox (Fig. 2A, pellet).

b. In a related question, why do isoforms respond differently to changes in expression of mi-R980? Don't they all have the same miR-980 binding sites in the 3' UTR?

c. What sequence in Rbfox1 3' UTR is targeted by miR-980, and what was sequence was used in the luciferase assays? Is this different from the targets described in the recent paper by Guven-Ozkan et al , ref. 67? Does the 3'UTR differ in 50kDa isoforms, which substantially increase in the miR-908Ex12 mutants, compared with the 3'UTR i larger isoforms that don't significantly increase (Figure S2A)?

d. Which isoform of Rbfox1 is over-expressed in OE experiments?

2. Are the RBP granules in which Rbfox1 apparently resides (under certain conditions) similar to the particles reported by others (e.g. in ref. 48)? Rbfox1 does not seem to be in the list(s) of granule proteins precipitated by b-isox in mammalian cells of that study, or collected from the literature. [REDACTED]. In addition, it seems premature to conclude without direct experimentation that Rbfox1 in this system aggregates due to its LCD motifs, because these seem to differ from the [G/S]Y[G/S] motifs specifically associated with precipitation in other proteins.

3. The authors should comment on the relationship of their findings of Rbfox1- containing granules to the recent characterization of the LASR (large assembly of splicing regulators) complex containing RBFOX proteins (ref. 28). The authors do cite the LASR paper but only in the more general context of a splicing function for RBFOX proteins; the large complex should be discussed.

4. [REDACTED]

5. There are many technical issues and/or experimental details that should be clarified. These will

be discussed with the relevant figures.

Figure 3a. On p. 20 the text states "in miR- 980 mutants, Rbfox1 association with the nucleolus is even more evident (yellow in Figure 3A)". However, in these small images, it is difficult to appreciate what appear to be very tiny yellow spots.

b. Forgive my ignorance, but the upper and lower panels of Figure 3D show the same field of cells that is apparently stained with two red-fluorescing antibodies (to both coilin and fibrillarin) – how is it possible to distinguish the two?

c. If mitotic stage ends at st6 (Fig. 1), then one cannot say Fig 3A shows that "epithelial cells organize their nucleoli differently depending on whether they are mitotic, endocycling, or amplifying" because no mitotic stage is shown.

Figure 4: Differences in expression of Cad, Atg8, Cas3, Myc, etc. are clear in the Rbfox-1 OE cells. One comment- in contrast to Figure 3, it's difficult to read the green labels that indicate staining of GFP. In 4A, it is difficult to see the fragmented membranes. In 4F, it is hard to see the change in polarity due to the small images; also a few words of explanation would help general reader understand how localization of Cad staining indicates polarity. In 4J-4K, how strong is the knockdown of Rbfox1? In 4K, to the nonspecialist it is difficult to appreciate the increased cell death, at least is part because the images again are very small. How can the nonspecialist appreciate whether Notch receptor is cleaved, and what is the significance of cleavage? In 4L, why would similar phenotypes be observed when Rbfox1 levels are up- or down-regulated?

[REDACTED]

General issue: for non-specialists, it would help to give a supplementary table explaining the various mutants used in the study. The technical designations of genotype are not always easy for general readers to interpret.

6. p.20: Maybe this is just semantics, but the statement that "widespread inclusion of Rbfox1 to different subcellular organelles can be explained due to the presence of multiple LCD sequences" seems more like a hypothesis, since no experiments were done to test this. Maybe it should say "might" be explained.

Minor issues:

1. The legend for figure 6 is missing the label for section G.
2. Text refers to generation of mutants miR-980ex11, miR-980ex12 and miR-980ex13) but all of these are identical (legend to Figure S1).
3. Supplementary methods section: Citation for Bajpai et al., 2004 is not given.
4. Legend for Figure S2, part C, states "Yellow dashed lines outline contours of miR-980-overexpressing clones" but the upper panel represents a GFP over-expression control with no over-expression of miR-980.
5. p. 17 What does "sites in addition" refer to?

--

Reviewer #2 (Remarks to the Author):

Kucherenko and Shcherbata demonstrated that

- (1) Rbfox1 is a target of miR-980 and that the phenotype of miR-980 loss can be rescued by the reduction to a single copy of Rbfox1 gene.
- (2) an isoform of Rbfox1 can be precipitated by b-isox and present in SDS-resistant particles;
- (3) Rbfox1 overexpression results in granular and filamentous pattern in nucleus and cytoplasm
- (4) Rbfox1 alters apoptosis, autophagy, growth and differentiation as well as introduction of benign tumor
- (5) Human homolog also precipitates with b-isox and appears in granules.

Overall this is a detailed analysis about RBFOX1 protein, however there are quite a few conceptual questions and data needs to be clarified before further considerations.

Major points:

1. Mechanism of RBFOX1 aggregates formation: The authors showed loss of miR-980, overexpression of Rbfox1 protein or under stress Rbfox1 protein form aggregates, but they did not show any mechanistic detail or signal which causes these aggregation.
2. Notably, the most interesting piece of data is actually the loss of miR-980 by more than two fold upon starvation stress - what is the mechanism?
3. It should be noted that in Fig.1D, there are at most 2-fold of increase of Rbfox1 mRNA and protein (Fig. 1E). However, Rbfox1 overexpression model is way higher (5-fold) - would this be a good model? This is an important question because nearly most of the data presented are based on Rbfox1 overexpression model. Is this model relevant to what happens during stress.
4. It is not clear how did the authors know whether each band in Fig.1E are in fact Rbfox1 protein isoforms. Why are there only 1 Rbfox1 mRNA measured in Fig. 1D? The robust signal that was increased upon miR-980 loss in Fig.1 E is the 50kDa band; unfortunately there are no such increase in Rbfox1 overexpression model. Why?
5. It will be much clearer if the authors point out individual isoforms in their Rbfox1 diagram.
6. The authors predominantly used Rbfox1 over-expression throughout the paper, but there are no details which isoforms was in use.
7. Can the authors explain why only 1 isoform is precipitated by b-isox and which isoform it corresponds? why the other isoform with low complexity domain (LCD) were not precipitated?
8. Can the authors explain why the Rbfox1 overexpression and non-transfected has different phenotype in Fig. 2E?
9. How can the authors be sure the specificity of Thioflavin T staining in Fig. 1J. Also, they need to provide proof that the observed colocalization in green channel is not due to crosstalk from the Thioflavin T channel
10. Liquid-liquid phase separation: The authors argue that Rbfox1 behaves like liquid droplet, but they did not show any experimental evidence to support it. I suggest the authors should check liquid behavior of Rbfox1 granules using liquid droplet assay (from purified Rbfox1 protein), or live cell imaging to show fusion of Rbfox1 aggregation over-time and demonstrate with photokinetic experiments such as FRAP.
11. All over the paper, the authors argue Rbfox1 aggregates as "RNP granules" but they did not check its co-localization with mRNA. They can be mere protein aggregates?
12. Rbfox1 in pathological condition: The authors speculate Rbfox1 granules can involve in human neurodegenerative and metabolic disorders based on the evidence that they have LCD regions and aggregate into distinct granules under stress but there is no direct experiment to support their model.
13. [REDACTED]
14. So far Rbfox1 overexpression results in granule formation and other phenotypes on apoptosis,

autophagy, growth and differentiation. However, these phenotypes are only correlative but not causative. They author should demonstrate that these phenotypes are due to granule formation using mutation analyses.

Minor points:

1. Fig. 1E: the authors showed Rbfox1 isoform expressed differentially. It would be interesting if the authors can test different isoforms localize differentially into cytoplasm and nucleus.?
2. Fig. 2B: what is negative control and experiment meant?
3. Fig 3: the authors showed the aggregation of Rbfox1 into different region/organelles. Can they point out what portion of Rbfox1 localizes into different region at given time (or stress)? Also, over-expression of miR-980, equally affects the localization of Rbfox1 into these structures?
4. Fig. 3E: any change in size of coilin in control vs. Rbfox1 overexpression?
5. Fig. 3H: the data is not clear or have either control only. The expression of Hsp70 can be hardly seen. I would recommend to separate the channels for clear visualization. In general, the figure resolution is too low and should be provided with separate channel.
6. Fig. 6D-E: compared to control arsenite treated cells showed round morphology but not in arsenite with 1,6 hexanediol treatment. Is this normal for all the cells observed in this experiment?
7. In text for Fig. 6F-G, the authors suggest Rbfox1 aggregates possibly resemble core vs shell phenomenon which is interesting, however it has not been previously reported that a given protein associates in both core and shell structure. Since the large Rbfox1 structure is relatively smaller after 1,6 hexanediol treatment, what are the authors' point of view?
7. Have the authors tested about the disintegration of Rbfox1 protein aggregates upon stress removal?

--

Reviewer #3 (Remarks to the Author):

The authors demonstrated that stress regulates assembly of RNP granules in a microRNA-specific manner. There are only few reports published about stress-responsive microRNAs and their roles in regulation of gene expression thus making this work novel and original.

I recommend this work for publication. Here are my minor requests:

1. Figure 2A is of low quality. I also suggest that authors would show precipitation pattern by silver staining of sds gel.
2. Figure 1B. Control for luciferase reporter should be included. This can be done by making mutations in miR980 binding site on mRNA reporter or by using other microRNAs that are not predicted to target Rbfox1 3'UTR
3. Figure 2C-D. Authors assume that granules formed in response to sodium arsenite are stress granules. Sodium arsenite in mammalian cells also promotes formation of P-bodies. Marker for SGs an PBs as well as FISH against poly(A) mRNAs should be used.

--

Reviewer #4 (Remarks to the Author):

In the manuscript "Stress-dependent miRNA-based regulation of Rbfox1/A2bp1 promotes RNP granule formation and cell survival", the authors examine the cytoplasmic and nuclear granule-forming dynamics of the RNP Rbfox1. They show under the stressful condition of starvation, cells

increase Rbfox1 levels, and decrease the Rbfox1 negative regulator miR-980. They also claim this stressor causes Rbfox1 granules to form in the cytoplasm and nucleus. Overall, the topic of this study is generally interesting due to the potential regulation of Rbfox1 granules by miR-980 under stress. However, the data quality with respect to image quality and controls diminishes enthusiasm for their conclusions. The manuscript seems unfocused and broad to the detriment of determining how Rbfox1 granules are regulated and what the significance is of their dynamics.

Specific comments:

Image quality

The figures would be improved with controls for the antibody staining. For example, in Fig. 3 it is difficult to discern differences between Rbfox1 staining at different stages and in the absence of miR-980 (what is in the nucleus, what is mottled/granular etc). The particles often look like a haze. Given that the authors rely heavily on two Rbfox1 antibodies, the antibodies should be better described. Where did the antibodies come from? Do they recognize all the isoforms? Is the staining gone in an Rbfox1 null mutant ovary? (minor comment – in the supplementary data, the antibody descriptions are not complete with respect to provenance). In addition, the germarium Rbfox1 staining (Fig. 1F) does not look like published results. This general comment about the micrographs can also be made for Fig. 6.

In Fig 3 F, H, I it is difficult to see colocalization with Cajal bodies without single images and the mitochondrial labeling is not convincing.

Liquid droplet claims:

p8 “our analysis identified...multiple low complexity sequence domains” and p18 “we could identify LCDs in all human RBFOX proteins”. From a literature search, it seems that the fact that Rbfox has intrinsically disordered domains is already established. Relatedly, Guven-Ozkan et al (2016) showed there are miR-980 binding sites in the Rbfox1 3’UTR. The authors should make is a little clearer they are confirming previously demonstrated work.

The authors use b-isox method to precipitate Rbfox as evidence that it forms cell-free RNP granules. Since they are using ovary extract, it would be helpful to have a positive control, in addition to the negative control, for this to be convincing. The authors conclude “we propose that Rbfox1...aggregates via LCDs” (p9). To use the biochemistry to conclude this, the authors need to show deleting the LCDs causes Rbfox to lose this property and that the LCDs from Rbfox aggregate on their own. When performing SDD-AGE analysis, it is important to add a reducing agent because proteins can oxidize and crosslink, forming larger species. This could give rise to artifacts.

The authors claim the Rbfox granules are liquid droplets (example p10). This is based on the addition of 1,6-hexanediol. However, related to the comment above with respect to images, there needs to be some biochemical analysis to backup this claim, not just S2 cells staining for Rbfox1. In addition, the authors claim Rbfox1 stains “large granule structures” in Fig. 2G. It is not clear to what they are referring in these images.

Filaments

The authors show Rbfox1 also forms filaments in the cells under stress (Fig. 2). They mention in the supplemental data that they tested CTPsynthase antibody to see if the filaments are cytoophidia and that these structures did not overlap. They need to show this data at least in the supplement. Rabbit antibodies have been known to overlap with cytoophidia, and guinea pig and rabbit antibodies can cross-react. In addition, these ovary samples are from flies starved for three weeks, which is a really long time.

Thioflavin T is commonly used in vitro to identify amyloid properties and is a spectroscopic assay. There seems to be non-specific background staining in the egg chamber shown in Fig. 2J in the nucleus and follicle cells, as well as what the authors are pointing out as an amyloid fiber (yellow

arrow). Without proper controls, such as a positive control that is known to form amyloid in the germ cells, and a negative control with no starvation, the images do not make a compelling argument for Rbfox forming amyloid fibers.

Nucleolus compartmentalization

Since the images in Fig. 3 are a little difficult to discern, it would make a stronger argument to have cell fractionation and Western blot showing Rbfox1 changes compartments under different stress conditions. The follicle cells, as noted, have different cell cycle properties at different stages. There are other tissues in *Drosophila* with similar attributes, such as nurse cell nuclei and salivary glands. It would bolster the author's argument that there are changes in Rbfox1 localization to use these other tissues. Plus, they are much larger nuclei compared to the small follicle cells.

Since Cajal bodies are also used for the splicing machinery, it is perhaps not surprising that Rbfox localizes there.

Cell survival, growth, differentiation and autophagy

The authors performed overexpression clonal analysis in the follicle cells and examined markers for growth and differentiation (Fig. 4). While there is quantification, for the Atg8, Cas9 and Cadherin analysis the egg chambers have been starved and in general do not look like they are healthy since they are stage 8. This is the stage when the follicles die after starvation. For the results looking at differentiation (Cut and Notch staining), these results could be from a lack of normal splicing or mRNA stability and not Rbfox1 granule dynamics.

Point-by-point responses to reviewers' comments:

First of all, we would like to thank the reviewers for their insightful comments and great suggestions, which have now been addressed in the significantly modified manuscript. We have changed the outline of the story, added more explanations to the text, performed new experiments, modified old and added new figures, which further supported our conclusions and made our story much better.

Reviewers' comments:

Reviewer #1 (Remarks to the Author):

This manuscript examines expression and localization of Rbfox1 as a function of stress conditions during *Drosophila* ovary development, demonstrating important new data that (1) Rbfox1 can aggregate into membraneless, subcellular RNP granules in a miR-980- and stress-dependent manner, and (2) Rbfox1 expression regulates follicle epithelial differentiation and early embryo survival. The authors also find that Rbfox1 expression is negatively regulated by miR-980, as shown earlier by others in studies of *Drosophila* brain, and show that the consequences of Rbfox1 mis-expression can be partially overcome by manipulating miR-980 expression. This data is used to propose a novel concept that miR-980 and its target Rbfox1 “interact to have a profound effect on cell survival, growth, and differentiation, and subsequently, on organismal viability upon stress.” The authors conclude that these interactions represent part of a new stress-responsive signaling cascade. Overall the data is quite interesting and provocative, especially the proposal that Rbfox1 is a pro-survival protein that contributes to slowing down metabolism under unfavorable conditions in an attempt to sustain the cells during period of temporary stress. However, there are many questions remaining to be answered.

1. Rbfox1 expression and localization: a. The authors demonstrate expression of a wide array of protein isoforms, some of which exhibit different abundances under stress. If not for earlier studies showing complex splicing patterns that can yield many proteins, this data would be quite remarkable. It is still confusing that numerous isoforms are present in the gels of Fig. 1E and 2A (supernatant), but predominantly one isoform is precipitated by b-isox (Fig. 2A, pellet).

We are sorry that the western blot image in Fig. 2A was of such low quality. For some reason, all figures turned out to be of very low resolution in the final pdf. Now we have replaced image in 2A with the new rescanned blot image, which shows that several Rbfox1 isoforms (at least 3 or 4) are precipitated by b-isox.

b. In a related question, why do isoforms respond differently to changes in expression of mi-R980? Don't they all have the same miR-980 binding sites in the 3' UTR?

Thank you for mentioning such an important point, which, we fully agree, requires an additional explanation. We have made a new Supplementary Figure depicting different Rbfox1 isoforms and possible 3'UTRs with miR-980 binding sites (Fig. S2). This figure shows that Rbfox1 mRNA has a possibility of having four alternative 3'UTRs and that miR-980 can target only the isoforms with the extended 3'UTRs. We thank the reviewer for the suggestions and hope that this additional information would be helpful to readers for easier understanding of Rbfox1 miRNA-based regulation.

c. What sequence in Rbfox1 3' UTR is targeted by miR-980, and what was sequence was used in the luciferase assays? Is this different from the targets described in the recent paper by Guven-Ozkan et al , ref. 67?

As suggested by the reviewer, we now added the information about the conserved miR-980 binding sites in the extended Rbfox1 3'UTRs and marked the 3'UTR sequence that was used for the luciferase assays (Figure S2C-D). These sites are predicted by TargetScan and appear to be the same as described in Guven-Ozkan et al, where they showed the sequence and location of the predicted sites in the figure. In our work, we also performed the luciferase assay to prove that miR-980 can target Rbfox1 3'UTR in vitro.

Does the 3'UTR differ in 50kDa isoforms, which substantially increase in the miR-908Ex12 mutants, compared with the 3'UTR i larger isoforms that don't significantly increase (Figure S2A)?

It is also true that the predicted 50kDa isoform has the extended 3'UTR and could be targeted by miR-980. We now added and discussed this information in the text.

However, to make definite conclusions and surely assign which 3'UTR belongs to which isoform, more experiments need to be done. Currently, it is unknown what regulates the choice of polyadenylation sites in vivo; however, it has been shown that depending on the cell type and developmental stage, the length of 3'UTRs greatly varies. As it is currently depicted in FlyBase, different Rbfox1 isoforms have distinct 3'UTRs assigned. However, based on our research on other genes, this assignment most likely is not so simple and straightforward. We think that each isoform could have different 3'UTRs and their length is dependent on the status of cellular metabolism. Also, it has been shown in (Guven-Ozkan et al., 2016; Tastan et al., 2010) that there are many other Rbfox1 splice variants that are not represented in the FlyBase GBrowse. Therefore, we carefully put together all currently available information on Rbfox1 isoforms based on FlyBase annotations, where the shortest isoform is shown to have the extended 3'UTR and to be targeted by mR-980, which could explain the significant increase in its levels in miR-980 mutants (Figure S2).

d. Which isoform of Rbfox1 is over-expressed in OE experiments?

We agree with the reviewer that this information is very important. Previously, we've just mentioned that we have overexpressed the RE isoform in the Figure 5 legend. Now we made a new supplementary Figures S2 and S5. As mentioned above, in Figure S2 all Rbfox1 isoforms currently predicted by FlyBase are depicted. This figure also shows that Rbfox1-RE does not contain the extended 3'UTR and, thus, is not subjected to mi-980 targeting. In the new supplementary Figure S5, we show the schematic of the protein domains included in this isoform. It shows that Rbfox1-PE contains 10 out of 13 predicted for Rbfox1 LCDs. Both these facts (absence of miR-980 binding sites and presence of multiple LCDs) are critical for better understanding the obtained results and drawn conclusions. In addition, this information explains reasoning of using this isoform for overexpression experiments, which is added to the text.

2. Are the RBP granules in which Rbfox1 apparently resides (under certain conditions) similar to the particles reported by others (e.g. in ref. 48)? Rbfox1 does not seem to be in the list(s) of granule proteins precipitated by b-isox in mammalian cells of that study, or collected from the literature. [REDACTED]

[REDACTED]

[REDACTED]

In addition, it seems premature to conclude without direct experimentation that Rbfox1 in this system aggregates due to its LCD motifs, because these seem to differ from the [G/S]Y[G/S] motifs specifically associated with precipitation in other proteins.

In Kato et al., 2012, only 27 out of 106 detected proteins had [G/S]Y[G/S] motifs, which of course, is a significant enrichment, but it also tells that there are other 75% of proteins that do not have [G/S]Y[G/S] motifs but appear in the liquid droplets.

3. The authors should comment on the relationship of their findings of Rbfox1- containing granules to the recent characterization of the LASR (large assembly of splicing regulators) complex containing RBFOX proteins (ref. 28). The authors do cite the LASR paper but only in the more general context of a splicing function for RBFOX proteins; the large complex should be discussed.

Thank you for this suggestion. Now we added the following discussion to the Discussion section: "Human Rbfox proteins associate with a multimeric complex of proteins called a large assembly of splicing regulators (LASR), which contains hnRNP M, hnRNP H, Martin3, hnRNP U-like-2, hnRNP C, NF 110, NF45, and DDX5/17 (Damianov et al., 2016). Importantly, a half of these proteins (hnRNP M, hnRNP H, hnRNP C, DDX5/17) that complex with Rbfox were also identified as proteins precipitated by b-isoX in the liquid droplet screen performed by the McKnight lab (Kato et al., 2012). Importantly, it has been shown that Rbfox proteins are associated with the LARS complex in the specific nuclear fractions; particularly, in the insoluble high molecular weight fraction containing chromatin, nuclear speckles, and unspliced RNA. Altered expression of Rbfox changes activity of hnRNP M, thus affecting splicing through indirect interaction with the RNA (Damianov et al., 2016). An Rbfox splice variant lacking most of the RNA binding domain, but retaining the N- and C-terminal domains can block splicing activation, but not splicing repression, suggesting that domains other than RRM in Rbfox have important roles in regulation of RNA metabolism (Damianov and Black, 2010). It has been also shown that in addition to its nuclear function as a splicing regulator, in the cytoplasm, Rbfox1 competes with miRNAs for the binding sites in the 3'UTRs. Thus, cytoplasmic Rbfox1 binding increases stability and translation of target mRNAs, especially mRNAs involved in cortical development and autism (Lee et al., 2016). It is known that the components of the miRNA machinery in the cytoplasm are found in P-bodies and our data show that Rbfox1 is involved in assembly of P-bodies, and we propose that this competition between miRNAs and Rbfox protein for the binding sites in 3'UTRs occurs in these liquid organelles".

4. [REDACTED]

[REDACTED]

5. There are many technical issues and/or experimental details that should be clarified. These will be discussed with the relevant figures.

Figure 3a. On p. 20 the text states “in miR- 980 mutants, Rbfox1 association with the nucleolus is even more evident (yellow in Figure 3A)”. However, in these small images, it is difficult to appreciate what appear to be very tiny yellow spots.

As suggested by the reviewer, we now show the enlarged images and new images of the nuclear Rbfox1 localization in the significantly larger nurse cells. (Fig 5A, new supplementary Figure S6).

b. Forgive my ignorance, but the upper and lower panels of Figure 3D show the same field of cells that is apparently stained with two red-fluorescing antibodies (to both coilin and fibrillarin) –how is it possible to distinguish the two?

We have stained the sample simultaneously with GFP (Alexa 488), Fibrillarin (Alexa 568), Coilin (Alexa 633) and DAPI (405) and then combined only 3 channels for each image in D. But we agree, it is confusing. We now put Coilin in Cyan (new Figure 5C).

c. If mitotic stage ends at st6 (Fig. 1), then one cannot say Fig 3A shows that “epithelial cells organize their nucleoli differently depending on whether they are mitotic, endocycling, or amplifying” because no mitotic stage is shown.

True, deleted.

Figure 4: Differences in expression of Cad, Atg8, Cas3, Myc, etc. are clear in the Rbfox-1 OE cells. One comment- in contrast to Figure 3, it’s difficult to read the green labels that indicate staining of GFP. In 4A, it is difficult to see the fragmented membranes. In 4F, it is hard to see the change in polarity due to the small images; also a few words of explanation would help general reader understand how localization of Cad staining indicates polarity.

We have fixed the colour for GFP text and deleted the statement for the fragmented membranes. As correctly mentioned, Cadherin the way we show it in 2E is not a clear cell polarity marker (the cell polar basolateral localization of Cadherin is better seen in 2F). In 2E, we used it to mark the cell membranes. What we meant to show was that the monolayer of the epithelial follicle cells is

perturbed. We clarified the description of the observed phenotype in the text and showed the enlarged image for this epithelial polarity defect (new Figure 2F).

In 4J-4K, how strong is the knockdown of Rbfox1? In 4K, to the nonspecialist it is difficult to appreciate the increased cell death, at least in part because the images again are very small. How can the nonspecialist appreciate whether Notch receptor is cleaved, and what is the significance of cleavage? In 4L, why would similar phenotypes be observed when Rbfox1 levels are up- or down-regulated?

To evaluate the strength of Rbfox1 RNAi, we measured Rbfox1 protein levels in follicle cell clones. These data are now included in the supplementary Figure S3B and S3E. The efficiency of RNAi is similar to miR-980 overexpression.

When the Notch receptor is cleaved, Notch signaling is activated and the transcription factor Cut is suppressed. Now we explained in the text that Cut expression was used to measure Notch signaling activation. Notch staining has been moved to supplementary S3F and in the main Figure 2G-I, we show only the Notch signaling reporter, Cut.

Up- and down- regulation of Rbfox1 delays follicle cell differentiation; however, it has an opposite effect on cell survival: upregulation promotes, while downregulation reduces cell survival. These data suggest that the maintenance of proper Rbfox1 levels is important for cell differentiation and that Rbfox1 expression levels positively correlate with cell survival. We clarified this in the text.

[REDACTED]

[REDACTED]

General issue: for non-specialists, it would help to give a supplementary table explaining the various mutants used in the study. The technical designations of genotype are not always easy for general readers to interpret.

We thank for the suggestion, the table with genotypes has been added to the Supplementary section (Table S7).

6. p.20: Maybe this is just semantics, but the statement that “widespread inclusion of Rbfox1 to different subcellular organelles can be explained due to the presence of multiple LCD sequences” seems more like a hypothesis, since no experiments were done to test this. Maybe it should say “might” be explained.

Agree, Corrected.

Minor issues:

1. The legend for figure 6 is missing the label for section G.

The legend has been added.

2. Text refers to generation of mutants miR-980ex11, miR-980ex12 and miR-980ex13) but all of these are identical (legend to Figure S1).

Right, they all turned out to be the same. We used multiple mutants to reduce the probability of having any additional mutations, deletions, suppressions that could affect the phenotype. We now renamed the mutants as miR-980^{ex1-1}, miR-980^{ex1-2} and miR-980^{ex1-3} and mentioned this in the Supplementary Figure 1 legend and Table S7.

3. Supplementary methods section: Citation for Bajpai et al., 2004 is not given.

Thank you, corrected.

4. Legend for Figure S2, part C, states “Yellow dashed lines outline contours of miR-980-overexpressing clones” but the upper panel represents a GFP over-expression control with no over-expression of miR-980.

Thank you, corrected. Now it is in Figure S3C.

5. p. 17 What does “sites in addition” refer to?

Corrected.

--

Reviewer #2 (Remarks to the Author):

Kucherenko and Shcherbata demonstrated that

(1) Rbfox1 is a target of miR-980 and that the phenotype of miR-980 loss can be rescued by the reduction to a single copy of Rbfox1 gene.

(2) an isoform of Rbfox1 can be precipitation by b-isox and present in SDS-resistant particles;

(3) Rbfox1 overexpression results in granular and filamentous pattern in nucleus and cytoplasm

(4) Rbfox1 alters apoptosis, autophagy, growth and differentiation as well as introduction of benign tumor

(5) Human homolog also precipitates with b-isox and appears in granules.

Overall this is a detailed analyses about RBFOX1 protein, however there are quite a few conceptual questions and data needs to be clarified before further considerations.

Major points:

1. Mechanism of RBFOX1 aggregates formation: The authors showed loss of miR-980,

overexpression of Rbfox1 protein or under stress Rbfox1 protein form aggregates, but they did not show any mechanistic detail or signal which causes these aggregation.

We agree with the reviewer that this is a very interesting question. Currently it is believed that the LCD-containing proteins aggregate and self-aggregate depending on their concentration. It has been shown that the presence of LCDs is both necessary and sufficient to form these hydrogels. In addition, the presence of RNAs significantly promotes formation of these aggregates. Our data show that any condition that results in the increased Rbfox1 levels causes assembly of Rbfox1 positive granules, suggesting that Rbfox1 aggregation is concentration dependent. We make sure that we now clearly explained this in the text. Of course, it would be interesting in the future to study whether there is a threshold for Rbfox1 concentration after which it starts to aggregate in vivo, what other LCD-containing proteins or RNAs can accelerate this process.

2. Notably, the most interesting piece of data is actually the loss of miR-980 by more than two fold upon starvation stress - what is the mechanism?

Again, it is a very interesting question and we do not have any answer to it yet, but agree it would be interesting to know the answer, especially when taken into account that miR-980 is downregulated by any stress we tested.

3. It should be noted that in Fig.1D, there are at most 2-fold of increase of Rbfox1 mRNA and protein (Fig. 1E). However, Rbfox1 overexpression model is way higher (5-fold) - would this be a good model? This is an important question because nearly most of the data presented are based on Rbfox1 overexpression model. Is this model relevant to what happens during stress.

It is true that Rbfox1 overexpression using UAS/GAL4 system is more efficient than the increase in Rbfox1 levels in miR-980 mutants and under stress. However, we disagree that most of the data presented are based on Rbfox1 overexpression model: in vivo Rbfox1 overexpression results in ovaries are shown only in one panel of Figure 2 and in Figure 5 and only for the clonal analysis. In S2 cells, we always studied Rbfox1 function in control and Rbfox1 transfected cells in parallel, which gave similar results. In human cells, no overexpression was used at all. Since all these analyses resulted in similar findings, we are quite confident that the overexpression data nicely complement stress and miR-980 loss of function analyses and can be used as a relevant model.

4. It is not clear how did the authors know whether each band in Fig.1E are in fact Rbfox1 protein isoforms.

The antibodies we used were generated in other labs (Shashidhara and Buszczak) and have been intensively tested for specificity. Of course, nothing is absolute and these antibodies could be somewhat unspecific and recognize additional bands. To additionally confirm the antibody specificity, we stained ovaries of hypomorphic mutant (Figure S3I) and added images where Rbfox1 RNAi clones were stained with Rbfox1 antibodies, which results in decreased Rbfox1 staining intensity in the clone (Figure S3E). Due to the complexity of Rbfox1 alternative splicing, it is really difficult to assign bands on westerns to protein isoforms. It has been already attempted by the Michael Buszczak and Ron Davis labs and proven to be difficult. Our expression patterns are very similar to the previously published patterns (Carreira-Rosario et al., 2016; Guven-Ozkan et al., 2016; Tastan et al., 2010; Usha and Shashidhara, 2010). Also, in our hands Rbfox1 antibody staining intensity is increased upon Rbfox1 overexpression and decreased upon Rbfox1 downregulation; therefore, we do not doubt that it is quite specific and can be used to address our questions. Now to additionally clarify what antibodies were used, we generated a new Sup Figure S5, where we show Rbfox1 sequences used to generate antibodies.

Why are there only 1 Rbfox1 mRNA measured in Fig. 1D?

In Figure 1D, the primers were generated to the extended 3'UTR of *Rbfox1*, which recognises all different *Rbfox1* isoforms with the extended 3'UTRs. Now in addition to primer sequences given in the Mat&methods, we show primers position in the 3'UTR in the new Sup. Figure S2D (green arrows).

The robust signal that was increased upon miR-980 loss in Fig.1 E is the 50kDa band; unfortunately there are no such increase in *Rbfox1* overexpression model. Why?

We agree with the reviewer that this observation has to be clarified. In the *Rbfox1* overexpression, we specifically overexpressed RE isoform that is predicted to produce *Rbfox1*-PE protein, while in miR-980 mutants, only the isoforms that do not contain the extended 3'UTRs with miR-980 binding sites can be upregulated. These include *Rbfox1*-RM that is predicted to produce *Rbfox1*-PM (~50kDa) protein. We have added more info about *Rbfox1* isoforms and 3'UTRs in the text and in new supplementary figures S2 and S5. Please see also response to reviewer 1, point 1.b, c, d.

5. It will be much clearer if the authors point out individual isoforms in their *Rbfox1* diagram.

We agree that it would be helpful, but based on the previously published extensive studies about *Rbfox1* isoforms, we feel that we still cannot confidently point out which isoform corresponds to which band. This question will require more detailed further studies. Please see also our response to reviewer 2, point 1b. Now added the following statement to the text: "currently annotated transcripts do not represent all possible combinations of alternative isoforms (Carreira-Rosario et al., 2016; Guven-Ozkan et al., 2016; Tastan et al., 2010); therefore, it was not possible to conclusively assign the behaviour of different bands on the western blots to the specific *Rbfox1* isoforms".

6. The authors predominantly used *Rbfox1* over-expression throughout the paper, but there are no details which isoforms was in use.

We thank the reviewer for pointing that the information we have provided about *Rbfox1* isoform was insufficient. It was provided in the Materials and Methods and in the old Figure 5 legend, but we agree that it should have been introduced in the text earlier. Now we show schematics of *Rbfox1*- RE mRNA isoform and *Rbfox1*-PE protein isoform with all predicted domains that was used for overexpression (Figures S5 and S2) and carefully explained the reasons for using this isoform in the text.

7. Can the authors explain why only 1 isoform is precipitated by b-isox and which isoform it corresponds? why the other isoform with low complexity domain (LCD) were not precipitated?

We thank the reviewer for noticing this discrepancy. We are sorry that the western blot picture in Fig. 2A was of such low quality. For some reason, all figures in our pdf file turned out to be of very low resolution. Now we have replaced it with the new rescanned image that clearly shows that several *Rbfox1* isoforms (at least 3 or 4) are precipitated by b-isox (now Figure 3A).

8. Can the authors explain why the *Rbfox1* overexpression and non-transfected has different phenotype in Fig. 2E?

We are not sure what the reviewer is referring to. Sodium arsenite treatment and sodium arsenite plus hexanediol treatments elicited exactly the same response for cytoplasmic *Rbfox1* localization. Under normal conditions, patterns are different because upon transfection, Flag-*Rbfox1* is present in high concentration and tends to accumulate in the nucleus. We called this expression pattern as "nuclear only", and this phenotype was not quantified for speckles assembly/disassembly. We are sorry if we have not explained this clearly. Now we added squares of different colours that correspond to the quantified phenotypes in the upper corners of all S2 cell images and made sure that all this info is described in the text.

9. How can the authors be sure the specificity of Thioflavin T staining in Fig. 1J. Also, they need to provide proof that the observed colocalization in green channel is not due to crosstalk from the Thioflavin T channel

Thioflavin T staining is quite difficult, since it requires HCl treatment before Ab staining. We do not think that there is a bleed-through between Thioflavin T and Rbfox1 channels because they show quite distinct patterns when shown in separate channels. However, we performed this experiment 2 times more. Now we have additional images of different Rbfox1 granules induced by stress that are positive for Thioflavin T. These new images are now included in new Fig. 4G.

10. Liquid-liquid phase separation: The authors argue that Rbfox1 behaves like liquid droplet, but they did not show any experimental evidence to support it. I suggest the authors should check liquid behavior of Rbfox1 granules using liquid droplet assay (from purified Rbfox1 protein), or live cell imaging to show fusion of Rbfox1 aggregation over-time and demonstrate with photokinetic experiments such as FRAP.

We thank the reviewer for this great suggestion. We now have performed FRAP experiments in the live oocytes to address the kinetics of Rbfox1-positive droplet behaviour. The new experimental data supporting the idea that Rbfox1 aggregates behave like liquid droplets are now shown in the main Figure 4H-J.

11. All over the paper, the authors argue Rbfox1 aggregates as “RNP granules” but they did not check its co-localization with mRNA. They can be mere protein aggregates?

We thank the reviewer for mentioning this lack of evidence. We thought that if it colocalizes with other known RNA-containing subcellular organelles, they should be RNP granules. However, recent studies show that there are so many types of RNA granules and that LCD proteins can aggregate even without RNA. Therefore, we performed additional experiments supporting our findings that Rbfox1 positive assemblies contain RNA. First, we performed in situ hybridization with the RNA probe that detects poly(A) tails of cytoplasmic mRNAs. The results are shown in Fig. 3G. In addition, we co-stained ovaries with Rbfox1 and additional known RNP granule proteins, such as Dis3 and Pacman (both are exoribonucleases) that detect different types of RNP granules, the nuclear RNA exosome complex and cytoplasmic P-bodies, respectively. The new data are included in the new Figure 5H-K and 6A-B and described in the text.

12. Rbfox1 in pathological condition: The authors speculate Rbfox1 granules can involve in human neurodegenerative and metabolic disorders based on the evidence that they have LCD regions and aggregate into distinct granules under stress but there is no direct experiment to support their model.

This is correct, we agree with the reviewer that this hypothesis must be further studied in vertebrate model organisms and humans and we made sure that we stated this in the text.

13. [REDACTED]

[REDACTED]

[REDACTED]

[REDACTED]

14. So far Rbfox1 overexpression results in granule formation and other phenotypes on apoptosis, autophagy, growth and differentiation. However, these phenotypes are only correlative but not causative. They author should demonstrate that these phenotypes are due to granule formation using mutation analyses.

We do not have any data that show that Rbfox1 overexpression results in granule formation. Rbfox1-positive granules are formed in response to stress. To be clearer about it, we now moved the overexpression data forward (Figure 2). We do not think that Rbfox1 acts directly on the components of the Notch, autophagy and apoptosis signaling pathways, but rather affects them indirectly via regulation of RNA metabolism in general. Now we have discussed this in the manuscript.

We also are not sure what other mutation analyses the reviewer is suggesting. We analysed miR-980 loss-of-function and overexpression, Rbfox1 up- and downregulation under normal and stress conditions. We quantified the appearance and the number of different granules, measured the levels of different markers for cellular metabolism. If the reviewer is implying that we should perform additional studies and generate new *Drosophila* mutants that have Rbfox1 isoforms unable to assemble into speckles, it will require a lot of additional work and time and cannot be included into this manuscript. Please see also our reply to the question above.

Minor points:

1. Fig. 1E: the authors showed Rbfox1 isoform expressed differentially. It would be interesting if the authors can test different isoforms localize differentially into cytoplasm and nucleus.?

We agree with the reviewer that since Rbfox1 contains isoforms that have very different isoform composition, it would be interesting to address this question in the future. For now, we gave a more detailed information about the LCD composition in Rbfox1 1-PE isoform.

2. Fig. 2B: what is negative control and experiment meant?

The negative control means that the sample were boiled, which should disintegrate aggregates, experiment means no boiling. We make sure that we clearly stated this in the text.

3. Fig 3: the authors showed the aggregation of Rbfox1 into different region/organelles. Can they point out what portion of Rbfox1 localizes into different region at given time (or stress)? Also, over-expression of miR-980, equally affects the localization of Rbfox1 into these structures?

We have not noticed any noticeable changes in re-distribution of Rbfox1-positive granules into different regions, it seems that there just more granules assembled in response to stress or miR-980 loss. Upon miR-980 overexpression, levels of Rbfox1 in general go down and Rbfox1 granules are not formed.

4. Fig. 3E: any change in size of coilin in control vs. Rbfox1 overexpression?

Yes, we see that the size of coilin-positive granules upon Rbfox1 overexpression are somewhat changed; however, due to the fact that the appearance of these granules is stage-dependend and Rbfox1 overexpressing clones are delayed in differentiation, we are hesitant to make any stronger statements.

5. Fig. 3H: the data is not clear or have either control only. The expression of Hsp70 can be hardly seen. I would recommend to separate the channels for clear visualization. In general, the figure resolution is too low and should be provided with separate channel.

We thank the reviewer for mentioning this. We now included a control picture (no heat shock) and the experiment data where we have separate channels for Hsp70 and Rbfox1 staining for the germarium and developing oocyte. The new image with better resolution are shown in Figure 6D-E. We also discussed stress granules more in the text and explained that they appear only upon stress and cannot be seen under normal conditions.

6. Fig. 6D-E: compared to control arsenite treated cells showed round morphology but not in arsenite with 1,6 hexanediol treatment. Is this normal for all the cells observed in this experiment?

No, the cell morphology was not apparently changed due to hexanediol or arsenite treatments, in all our experiments, cells were round- or more star-like shaped, which possibly depends on the efficiency of heir adhesion to the slide.

7. In text for Fig. 6F-G, the authors suggest Rbfox1 aggregates possibly resemble core vs shell phenomenon which is interesting, however it has not been previously reported that a given protein associates in both core and shell structure. Since the large Rbfox1 structure is relatively smaller after 1,6 hexanediol treatment, what are the authors' point of view?

We agree with the reviewer that it is an interesting question. Since our data show that Rbfox1 can aggregate into assemblies with different properties (small or larger droplets and even fibers), we think it can exist in the states necessary to fulfil the requirements for the core and shell proteins. The statement about core vs. shell is quite new, we believe that soon there would be more examples published adding to our understanding of this phenomenon.

7. Have the authors tested about the disintegration of Rbfox1 protein aggregates upon stress removal?

We have not tested this, but we think it is a great idea. It must be done in vivo, when analysing the same cell under different conditions, unfortunately, we do not have this method established yet.

--

Reviewer #3 (Remarks to the Author):

The authors demonstrated that stress regulates assembly of RNP granules in a microRNA-specific manner. There are only few reports published about stress-responsive microRNAs and their roles in regulation of gene expression thus making this work novel and original.

I recommend this work for publication. Here are my minor requests:

1. Figure 2A is of low quality. I also suggest that authors would show precipitation pattern by silver staining of sds gel.

We are sorry that the western blot image in Fig. 2A was of such low quality. For some reason, all figures in our pdf file turned out to be of very low resolution in the final pdf. Now we have replaced image in 2A with the new rescanned blot image, which shows that several Rbfox1 isoforms (at least 3 or 4) are precipitated by b-isox.

2. Figure 1B. Control for luciferase reporter should be included. This can be done by making mutations in miR980 binding site on mRNA reporter or by using other microRNAs that are not predicted to target Rbfox1 3'UTR

Thank you, we repeated the luciferase experiment using, in addition to miR-980, two other miRNAs that are not predicted to target Rbfox1. The new data are now included in Figure 1B and Table S7.

3. Figure 2C-D. Authors assume that granules formed in response to sodium arsenite are stress granules. Sodium arsenite in mammalian cells also promotes formation of P-bodies. Marker for SGs an PBs as well as FISH against poly(A) mRNAs should be used.

We thank the reviewer for suggesting these experiments. We stained ovaries with Drosophila P-body marker, Pacman and it appeared that all P-bodies contain Rbfox1, but not all Rbfox1 granules are P-bodies. We added the info to the text about sodium arsenite being able to induce P-bodies in addition to stress granules. The new data are in Figure 6A-B. We also performed FISH against poly(A) mRNAs and found that some of the Rbfox1 granules contain mRNAs. The new data are in Figure 3G-H.

--

Reviewer #4 (Remarks to the Author):

In the manuscript "Stress-dependent miRNA-based regulation of Rbfox1/A2bp1 promotes RNP granule formation and cell survival", the authors examine the cytoplasmic and nuclear granule-forming dynamics of the RNP Rbfox1. They show under the stressful condition of starvation, cells increase Rbfox1 levels, and decrease the Rbfox1 negative regulator miR-980. They also claim this stressor causes Rbfox1 granules to form in the cytoplasm and nucleus. Overall, the topic of this study is generally interesting due to the potential regulation of Rbfox1 granules by miR-980 under stress. However, the data quality with respect to image quality and controls diminishes enthusiasm for their conclusions. The manuscript seems unfocused and broad to the detriment of determining how Rbfox1 granules are regulated and what the significance is of their dynamics.

Specific comments:

Image quality

The figures would be improved with controls for the antibody staining. For example, in Fig. 3 it is difficult to discern differences between Rbfox1 staining at different stages and in the absence of miR-980 (what is in the nucleus, what is mottled/granular etc). The particles often look like a haze.

Given that the authors rely heavily on two Rbfox1 antibodies, the antibodies should be better described. Where did the antibodies come from? Do they recognize all the isoforms? Is the staining gone in an Rbfox1 null mutant ovary? (minor comment – in the supplementary data, the antibody descriptions are not complete with respect to provenance). In addition, the germarium Rbfox1 staining (Fig. 1F) does not look like published results. This general comment about the micrographs can also be made for Fig. 6.

We would like to apologize for the quality of the images in the pdf file that the reviewers received. Now we have uploaded the full resolution images and we hope that the reviewers could appreciate them better. In Fig. 3C-D, we have added a small coloured box in the upper corner of the images of the cells, colour of which corresponds to the described and quantified patterns in the bargraph in Fig. 3E. We enlarged and moved images of the follicle epithelium cells at different stages from old Fig. 3A and 3B into new supplementary Figure S6. Now, in Figure 5A, only st.9 follicle cells with the separate channels and overlay are shown.

We also agree that we should have added more information on the Rbfox1 antibodies. We now added a new supplementary Figure S5, where we describe in greater detail which LCDs these antibodies can potentially recognize. In short, polyclonal guinea pig anti-Rbfox1 antibodies (Tastan et al., 2010) recognize – LCD 2, 3, 6 and 7, while polyclonal rabbit anti-Rbfox1 antibodies (Usha and Shashidhara, 2010) in addition recognize LCD 8, 9,10, 13, RRM and coiled-coiled domains. Based on the predicted by FlyBase Rbfox1 isoform sequences (Figure S2 and S4), both antibodies should recognize all Rbfox1 isoforms.

Rbfox1 mutants are lethal; therefore, we could analyze only Rbfox1 RNAi clones and Rbfox1 hypomorphic mutants. In both cases, Rbfox1 levels were downregulated suggesting that the antibodies are specific. These data are now shown in Figure S3B, E, I. We also added the antibody description to the Materials and Methods section.

In Fig. 1F, a maximum projection of multiple Z-sections of the ovariole was shown for both controls and miR-980 mutants. It was necessary to use a maximum projection to show Rbfox1 staining at different stages of oogenesis from the germarium to st. 8. Now we replaced 1F image with a new control that is an overlay of fewer sections. In addition, we also show a single section of control and mutant germarium in the new Figure S6A-A', where Rbfox1 staining in the germarium looks more distinct. We double checked and made sure that the information whether the image is a projection of Z-stacks or a single plane image is always included in the figure legends.

In Fig 3 F, H, I it is difficult to see colocalization with Cajal bodies without single images and the mitochondrial labeling is not convincing.

We are thankful for the reviewer's comment. We enlarged old and added new images with Rbfox1 and Cajal body marker colocalization in the nurse cells (Figure 5F). We also added new images in Figure 5, 6 and S6 to make sure that the colocalization of Rbfox1 with different other markers of the liquid organelles is better shown. We also used new markers for P-body and the exosome complex to additionally support our conclusions about Rbfox1 subcellular localization (5H-I, 6F-G).

Liquid droplet claims:

p8 “our analysis identified...multiple low complexity sequence domains” and p18 “we could identify LCDs in all human RBFOX proteins”. From a literature search, it seems that the fact that Rbfox has intrinsically disordered domains is already established. Relatedly, Guven-Ozkan et al (2016) showed there are miR-980 binding sites in the Rbfox1 3'UTR. The authors should make it a little clearer they are confirming previously demonstrated work.

To the best of our knowledge and based on our literature search the only info about Rbfox being a LCD-containing protein comes from our conference abstracts that appear online. If there is any other publication about this that we could not find, please send us the info and we would be happy to incorporate it into the manuscript. Usha and Shadshidhara, 2010 noted long poly(Q) tracts in *Drosophila* Rbfox1 (Supp. Figure 1) and we now referred to this work. It is also true, in Guven-Ozkan et al (2016) it has been nicely shown that Rbfox1 levels depend on miR-980 and we have cited this work and discussed in Results and Discussion. Now we also added this info at the very

beginning of the Results section. However, in the mentioned paper, the authors only found based on *in silico* analysis that there are miR-980 binding sites in the Rbfox1 3'UTR, and no further testing to prove that miR-980 can target Rbfox1 via its 3'UTR was done.

The authors use b-isox method to precipitate Rbfox as evidence that it forms cell-free RNP granules. Since they are using ovary extract, it would be helpful to have a positive control, in addition to the negative control, for this to be convincing. The authors conclude “we propose that Rbfox1...aggregates via LCDs” (p9). To use the biochemistry to conclude this, the authors need to show deleting the LCDs causes Rbfox to lose this property and that the LCDs from Rbfox aggregate on their own.

To precipitate Rbfox1, we used a commonly accepted protocol, where Rbfox1 overexpression was used as an additional positive control settings. We found that Rbfox1 association with b-isox is concentration-dependent. If it would depend on other unspecific proteins, we would not expect to observe such an increase in the precipitated Rbfox1 levels due to Rbfox1 overexpression as we see in the lower panel of Figure 3A. Similar, in the SDD-AGE analysis, we used non-starved animals as controls. The gel is showing that the amount of SDS-resistant Rbfox1-positive particles increases in response to starvation stress. As an additional control, all samples were boiled, which normally results in aggregates disintegration.

To prove that the ability of Rbfox1 depends on LCDs, we analyzed Rbfox1 isoforms in S2 cells that contain different LCDs. As can be seen from the schematics of Rbfox1 protein structure, it contains multiple extended LCDs, and deleting all of them is not feasible. At the same time, it is known that these domains interact with very low specificity; therefore, deleting one or another most likely would not have an effect on Rbfox1 functions. When we expressed Rbfox1 isoforms that differed by their LCD content, the isoform with 4, 5, 9, coiled-coil,10 and no 11 and 12 (Rbfox1 PE) and isoform that had no 4, 5, 9,10 and coiled-coil domain, but with 11 and 12 (Rbfox1 PI), we found that their subcellular localization and the ability to assemble granules upon stress were not significantly different. However, expression of the isoform with all of the mentioned LCDs significantly promoted granule formation, suggesting that Rbfox1 forms granules depending on the presence of LCDs, but it is not clear which LCD influences each specific localization. In addition, now we completed generation of few other deletion constructs that lack RRM, mitochondria localization signal and coiled-coil domain. The data will be included in the next paper. Please see also response to reviewer 2, point 13.

When performing SDD-AGE analysis, it is important to add a reducing agent because proteins can oxidize and crosslink, forming larger species. This could give rise to artifacts.

We thank the reviewer for noticing this mistake, we did use the reducing agent, 1mM DTT in all our experiments; now this typo is corrected.

The authors claim the Rbfox granules are liquid droplets (example p10). This is based on the addition of 1,6-hexanediol. However, related to the comment above with respect to images, there needs to be some biochemical analysis to backup this claim, not just S2 cells staining for Rbfox1. In addition, the authors claim Rbfox1 stains “large granule structures” in Fig. 2G. It is not clear to what they are referring in these images.

To additionally support our claim, we performed FRAP analysis and found that the recovery time of Rbfox1-positive granules is similar to other published results that described liquid droplets. The new data are now in Figure 4H-J and discussed in the text. We also added arrows in panel Figure 4A that point to Rbfox1 positive granules.

Filaments

The authors show Rbfox1 also forms filaments in the cells under stress (Fig. 2). They mention in the supplemental data that they tested CTPsynthase antibody to see if the filaments are cytophidia and that these structures did not overlap. They need to show this data at least in the

supplement. Rabbit antibodies have been known to overlap with cytoophidia, and guinea pig and rabbit antibodies can cross-react. In addition, these ovary samples are from flies starved for three weeks, which is a really long time.

As suggested by the reviewer, we have added a new Figure S7, showing that Rbfox1 antibodies do not mark cytoophidia and histone locus bodies (new Figure S7). Unlike Rbfox1 positive fibres, cytoophidia can be found in most cells under normal and stressed conditions (Figure S7B). Rbfox1 and CTpsyn co-staining shows no overlap. Moreover, we can detect fibres not only with rabbit Rbfox1 (Figure 4C, 5F), but also with guinea pig Rbfox1 (Figure 4D-F) and anti-Flag (Flag-tagged Rbfox1 in S2 cells, Figure 3F) antibodies.

As stress, we used protein starvation only. Under such conditions, flies seem to be OK; however, the oogenesis is completely disrupted. We made sure that the info about stress conditions is clear. In addition, we analysed not only 3 week-starved ovaries, but also ovaries after one week of starvation. Now this info is included into figure legends and text.

Thioflavin T is commonly used in vitro to identify amyloid properties and is a spectroscopic assay. There seems to be non-specific background staining in the egg chamber shown in Fig. 2J in the nucleus and follicle cells, as well as what the authors are pointing out as an amyloid fiber (yellow arrow). Without proper controls, such as a positive control that is known to form amyloid in the germ cells, and a negative control with no starvation, the images do not make a compelling argument for Rbfox forming amyloid fibers.

As correctly mentioned by the reviewer, Thioflavin staining is complicated, the required treatment does not always allow for the following antibody staining. We have repeated this experiment and added an additional panel to Figure 4, showing Thioflavin staining to be colocalized with Rbfox1 in a large nuclear droplet (Figure 4E). Since Thioflavin does not overlap in most cases with Rbfox1 signal, just in some fibres and large granules, we think that both stainings are specific. Unfortunately, we are not aware of any protein that would make amyloid fibers in the germline and could be used as a positive control. Therefore, we make sure that we always call Rbfox1 fibres as “amyloid-like”.

Nucleolus compartmentalization

Since the images in Fig. 3 are a little difficult to discern, it would make a stronger argument to have cell fractionation and Western blot showing Rbfox1 changes compartments under different stress conditions. The follicle cells, as noted, have different cell cycle properties at different stages. There are other tissues in *Drosophila* with similar attributes, such as nurse cell nuclei and salivary glands. It would bolster the author's argument that there are changes in Rbfox1 localization to use these other tissues. Plus, they are much larger nuclei compared to the small follicle cells.

We thank the reviewer for the suggestion. We now included images of Rbfox1 association with the nucleolus in the nurse cells. Now most of the images from Figure 3 are moved to new supplementary Figure S6 and new images with better resolution and also showing large nurse cells are used in new Figure 5 and 6 to depict Rbfox1 subcellular localization.

Since Cajal bodies are also used for the splicing machinery, it is perhaps not surprising that Rbfox localizes there.

We agree, it has been previously shown that Rbfox proteins act as components of the splicing machinery and therefore, could be localized to Cajal bodies. Now we have mentioned this in the text.

Cell survival, growth, differentiation and autophagy

The authors performed overexpression clonal analysis in the follicle cells and examined markers for growth and differentiation (Fig. 4). While there is quantification, for the Atg8, Cas9 and Cadherin analysis the egg chambers have been starved and in general do not look like they are healthy since

they are stage 8. This is the stage when the follicles die after starvation. For the results looking at differentiation (Cut and Notch staining), these results could be from a lack of normal splicing or mRNA stability and not Rbfox1 granule dynamics.

The reviewer correctly noticed that the Atg8, Cas9 and Cadherin analysis was done in the starved egg chambers. It was done specifically to show that in starved ovaries, control follicle cells are dying, while clonal cell with higher Rbfox1 levels survive better. It is also plausible that for the results looking at differentiation, the observed defects could be associated with abnormal splicing or mRNA stability, the processes that are happening in RNA granules. Since Rbfox1 levels influence RNA granule dynamics, we propose that the differentiation defects observed as a result of Rbfox1 up- or down-regulation could be caused by Rbfox1 function in these granules. Now we modified the text and added the reviewer suggestion to explain the differentiation and cell death phenotypes better.

- Brix, J., Dietmeier, K., and Pfanner, N. (1997). Differential recognition of preproteins by the purified cytosolic domains of the mitochondrial import receptors Tom20, Tom22, and Tom70. *The Journal of biological chemistry* *272*, 20730-20735.
- Carreira-Rosario, A., Bhargava, V., Hillebrand, J., Kollipara, R.K., Ramaswami, M., and Buszczak, M. (2016). Repression of Pumilio Protein Expression by Rbfox1 Promotes Germ Cell Differentiation. *Developmental cell* *36*, 562-571.
- Damianov, A., and Black, D.L. (2010). Autoregulation of Fox protein expression to produce dominant negative splicing factors. *RNA* *16*, 405-416.
- Damianov, A., Ying, Y., Lin, C.H., Lee, J.A., Tran, D., Vashisht, A.A., Bahrami-Samani, E., Xing, Y., Martin, K.C., Wohlschlegel, J.A., *et al.* (2016). Rbfox Proteins Regulate Splicing as Part of a Large Multiprotein Complex LASR. *Cell* *165*, 606-619.
- Fukasawa, Y., Tsuji, J., Fu, S.C., Tomii, K., Horton, P., and Imai, K. (2015). MitoFates: improved prediction of mitochondrial targeting sequences and their cleavage sites. *Mol Cell Proteomics* *14*, 1113-1126.
- Guyen-Ozkan, T., Busto, G.U., Schutte, S.S., Cervantes-Sandoval, I., O'Dowd, D.K., and Davis, R.L. (2016). MiR-980 Is a Memory Suppressor MicroRNA that Regulates the Autism-Susceptibility Gene A2bp1. *Cell reports* *14*, 1698-1709.
- Kato, M., Han, T.W., Xie, S., Shi, K., Du, X., Wu, L.C., Mirzaei, H., Goldsmith, E.J., Longgood, J., Pei, J., *et al.* (2012). Cell-free formation of RNA granules: low complexity sequence domains form dynamic fibers within hydrogels. *Cell* *149*, 753-767.
- Kotzamani, D., and Plaitakis, A. (2012). Alpha helical structures in the leader sequence of human GLUD2 glutamate dehydrogenase responsible for mitochondrial import. *Neurochem Int* *61*, 463-469.
- Lee, J.A., Damianov, A., Lin, C.H., Fontes, M., Parikshak, N.N., Anderson, E.S., Geschwind, D.H., Black, D.L., and Martin, K.C. (2016). Cytoplasmic Rbfox1 Regulates the Expression of Synaptic and Autism-Related Genes. *Neuron* *89*, 113-128.
- Lemire, B.D., Fankhauser, C., Baker, A., and Schatz, G. (1989). The mitochondrial targeting function of randomly generated peptide sequences correlates with predicted helical amphiphilicity. *The Journal of biological chemistry* *264*, 20206-20215.
- Tastan, O.Y., Maines, J.Z., Li, Y., McKearin, D.M., and Buszczak, M. (2010). Drosophila ataxin 2-binding protein 1 marks an intermediate step in the molecular differentiation of female germline cysts. *Development* *137*, 3167-3176.
- Usha, N., and Shashidhara, L.S. (2010). Interaction between Ataxin-2 Binding Protein 1 and Cubitus-interruptus during wing development in Drosophila. *Developmental biology* *341*, 389-399.

Reviewer #1 (Remarks to the Author):

This revised manuscript contains substantial immunofluorescence data to support the hypothesis that Rbfox1, in cells subjected to stress, promotes formation of various RNA granules in both nucleus and cytoplasm that regulate RNA metabolism to help modulate a survival response. Included are several new experiments showing that Rbfox1 can co-localize with the P body marker Pacman, and with a marker for the nuclear RNA exosome complex, Dis3. Moreover, additional analysis indicates that several of these marker proteins contain LCD domains that might facilitate interaction with Rbfox1 to form these liquid granule compartments. There is also new co-localization data with the mitochondrial marker ATP synthase, as well as identification of a putative mitochondrial localization signal near the N-terminus of Rbfox1, needed for transport across the mitochondrial membranes. The revised manuscript is improved but there are still some questions to be addressed.

1. Some of the new immunofluorescence data, while intriguing, is not entirely convincing. For example, Figure 5I says that "both Rbfox1 and Dis3 assembled into the nuclear assemblies in response to stress" but there is only one nucleus showing this colocalization at st.7, while the other three nuclei in the field do not show this. Can this be quantitated in some way so as to exclude the possibility that the apparent co-localization is actually due to occasional overlap between Rbfox1-containing Cajal bodies? Until this data is stronger, it seems quite speculative to propose that Rbfox1 promotes assembly of granules containing RNA exosome complex components to modulate RNA turnover as part of a stress response.

2. Top of p.13 : It seems highly unlikely that the half-life of Group 2 droplets could have such a narrow time of recovery in the FRAP experiment ($t_{half}=6.93\pm 0.01s$), especially if evaluating recovery intensity is somewhat subjective.

3. [REDACTED]

4. Authors should cite and discuss the very recent paper from Doug Black's lab on hydrogel behavior of mammalian RBFOX proteins, published in July 2017 (just before resubmission of this manuscript).

Minor issues:

1. On p. 17, the data related to Pacman expression levels and the assembly of Pacman-positive P-bodies upon increased Rbfox1 levels is shown in Figure 6B-C (not Figure 5B-C).

--

Reviewer #2 (Remarks to the Author):

The revised manuscript is much improved in terms of its clarity and image qualities. The authors clearly demonstrated that (1) Rbfox1 level is regulated by starvation-dependent level of miR-980; (2) Rbfox1/miR-980 level is correlated with the cell survival; (3) Rbfox1/miR-980 level is correlated with the formation of RNA granules in different part of the cells. Though the mechanisms on how RNA granule formation contribute to cell survival and how LCD contribute to its "promiscuous" association with various cellular structure are not explored in this manuscript, the data presented are interesting and solid. Therefore, the manuscript should be considered for publication with minor adjustments into their claims:

(1) Abstract: "Reduced miR-980 expression during stress leads to increased Rbfox1 levels, followed by widespread formation of RNP granules, promoting cell survival." The latter part of the sentence implies causality between widespread formation of RNP granules and cell survival, where correlation is only shown.

(2) Introduction: "Via its multiple LCDs, Rbfox1 promiscuously associates with the nucleolus and Cajal bodies in the nucleus as well as stress granules and processing bodies in the cytoplasm. " It is not shown (though a reasonable hypothesis) in the manuscript that Rbfox1 promiscuously associates with different RNP granules in cytoplasm and nucleus due to its association with multiple LCDs.

--

Reviewer #3 (Remarks to the Author):

All my concerns were addressed adequately. I recommend the manuscript for publication.

--

Reviewer #4 (Remarks to the Author):

Specific comments:

I stand by my first criticism that the paper is too unfocused to the detriment of its conclusions: miR-980 knockdown has interesting phenotypes, Rbfox is very complicated with many isoforms, Rbfox may change subcellular localization, there is the idea of liquid-droplet-like properties to Rbfox, etc. Any of these subjects by themselves is very complex. What is the main point of the paper?

In addition, while I appreciate the authors better describing the Rbfox antibodies, I remain unconvinced whether we are looking at the right things or not. The manuscript relies very heavily on immunofluorescence. It is exceedingly difficult to discern different staining patterns of Rbfox in the ovary of any of the figures.

Previous comment: Given that the authors rely heavily on two Rbfox1 antibodies, the antibodies should be better described. Where did the antibodies come from? Do they recognize all the isoforms? Is the staining gone in an Rbfox1 null mutant ovary?

Author's response:

We also agree that we should have added more information on the Rbfox1 antibodies. We now added a new supplementary Figure S5, where we describe in greater detail which LCDs these antibodies can potentially recognize. In short, – LCD 2, 3, 6 and 7, while polyclonal guinea pig anti-Rbfox1 antibodies (Tastan et al., 2010) recognize polyclonal rabbit anti-Rbfox1 antibodies (Usha and Shashidhara, 2010) in addition recognize LCD 8, 9,10, 13, RRM and coiled-coiled domains. Based on the predicted by FlyBase Rbfox1 isoform sequences (Figure S2 and S4), both antibodies should recognize all Rbfox1 isoforms. Rbfox1 mutants are lethal; therefore, we could analyze only Rbfox1 RNAi clones and Rbfox1 hypomorphic mutants. In both cases, Rbfox1 levels were downregulated suggesting that the antibodies are specific. These data are now shown in Figure S3B, E, I.

Reviewer's Response: The representative micrograph shown in Fig. S3E (Rbfox1 RNAi) is not convincing with respect to reduced Rbfox protein levels. As described in the rebuttal above, this antibody should recognize all isoforms and the RNAi knockdown construct should recognize all isoforms as well.

Previous comment: In Fig 3 F, H, I it is difficult to see colocalization with Cajal bodies without single images and the mitochondrial labeling is not convincing.

Author's response: We are thankful for the reviewer's comment. We enlarged old and added new images with Rbfox1 and Cajal body marker colocalization in the nurse cells (Figure 5F).

Reviewer's response: It is still hard to see without the single-labeled images.

Author's response: We also added new images in Figure 5, 6 and S6 to make sure that the colocalization of Rbfox1 with different other markers of the liquid organelles is better shown. We also used new markers for P-body and the exosome complex to additionally support our conclusions about Rbfox1 subcellular localization (5H-I, 6F-G).

Reviewer's response: [REDACTED]

Previous comment: The authors use b-isox method to precipitate Rbfox as evidence that it forms cell-free RNP granules. Since they are using ovary extract, it would be helpful to have a positive control, in addition to a negative control, for this to be convincing.

Author's Response:

To precipitate Rbfox1, we used a commonly accepted protocol, where Rbfox1 overexpression was used as an additional positive control settings. We found that Rbfox1 association with b-isox is concentration-dependent. If it would depend on other unspecific proteins, we would not expect to observe such an increase in the precipitated Rbfox1 levels due to Rbfox1 overexpression as we see in the lower panel of Figure 3A.

Reviewer's response: A positive control would be probing the blot for a protein that is known to precipitate in response to b-isox.

Previous comment: The authors show Rbfox1 also forms filaments in the cells under stress (Fig. 2). They mention in the supplemental data that they tested CTPsynthase antibody to see if the filaments are cytophidia and that these structures did not overlap. They need to show this data at least in the supplement. Rabbit antibodies have been known to overlap with cytophidia, and guinea pig and rabbit antibodies can cross-react.

Author's response:

As suggested by the reviewer, we have added a new Figure S7, showing that Rbfox1 antibodies do

not mark cytophidia and histone locus bodies (new Figure S7).... Rbfox1 and CTPsyn co-staining shows no overlap.

Reviewer's response: Fig. S7 does not show Rbfox1-positive filaments in the same cells (miR-980/park mutant) as CTPsyn-positive filaments. In addition, the "filaments" in Fig. 4C,C' look like they could be nuclear envelope and do not look the same as Fig. 4E.

Previous comment: Thioflavin T is commonly used in vitro to identify amyloid properties and is a spectroscopic assay. There seems to be non-specific background staining in the egg chamber shown in Fig. 2J in the nucleus and follicle cells, as well as what the authors are pointing out as an amyloid fiber (yellow arrow). Without proper controls, such as a positive control that is known to form amyloid in the germ cells, and a negative control with no starvation, the images do not make a compelling argument for Rbfox forming amyloid fibers.

Author's response:

Thioflavin staining is complicated, the required treatment does not always allow for the following antibody staining. We have repeated this experiment and added an additional panel to Figure 4, showing Thioflavin staining to be colocalized with Rbfox1 in a large nuclear droplet (Figure 4E)....

Unfortunately, we are not aware of any protein that would make amyloid fibers in the germline and could be used as a positive control. Therefore, we make sure that we always call Rbfox1 fibres as “amyloid-like”.

Reviewer’s response: I appreciate that the authors now say “amyloid-like”, but the staining is so weak for the Rbfox1 co-localization (Fig. 4F, Rbfox1(GP)), and unconvincing for the nuclear stain (Fig. 4G (not E)). In addition, there is still no negative control of well-fed flies.

Point-by-point responses to reviewers' comments:

We would like to thank the reviewers again for the careful reading of our manuscript and helpful comments, which have now been addressed in the modified manuscript.

Reviewers' comments:

Reviewer #1 (Remarks to the Author):

This revised manuscript contains substantial immunofluorescence data to support the hypothesis that Rbfox1, in cells subjected to stress, promotes formation of various RNA granules in both nucleus and cytoplasm that regulate RNA metabolism to help modulate a survival response. Included are several new experiments showing that Rbfox1 can co-localize with the P body marker Pacman, and with a marker for the nuclear RNA exosome complex, Dis3. Moreover, additional analysis indicates that several of these marker proteins contain LCD domains that might facilitate interaction with Rbfox1 to form these liquid granule compartments. There is also new co-localization data with the mitochondrial marker ATP synthase, as well as identification of a putative mitochondrial localization signal near the N-terminus of Rbfox1, needed for transport across the mitochondrial membranes. The revised manuscript is improved but there are still some questions to be addressed.

1. Some of the new immunofluorescence data, while intriguing, is not entirely convincing. For example, Figure 5I says that “both Rbfox1 and Dis3 assembled into the nuclear assemblies in response to stress” but there is only one nucleus showing this colocalization at st.7, while the other three nuclei in the field do not show this. Can this be quantitated in some way so as to exclude the possibility that the apparent co-localization is actually due to occasional overlap between Rbfox1-containing Cajal bodies? Until this data is stronger, it seems quite speculative to propose that Rbfox1 promotes assembly of granules containing RNA exosome complex components to modulate RNA turnover as part of a stress response.

We thank the reviewer for this question, we quantified this experiment and actually found that only 27.86% of nurse cell nuclei contained Dis3 speckles (n=61 nuclei) The data is included in Figure Legend. In addition, depending on their size, Dis3 granules have either high (large speckles, yellow arrow) or low Rbfox1 levels (small speckles, blue arrow). It would be really interesting to study this phenomenon further. Now we added the following statement to the text: “Notably, all large Dis3-positive stress-induced speckles ($1.17 \pm 0.39 \mu\text{m}$, n=17 speckles from 9 egg chambers) also had high Rbfox1 levels, while smaller speckles ($0.52 \pm 0.15 \mu\text{m}$, n=19 speckles from 6 egg chambers) had low Rbfox1, suggesting that Dis3-positive aggregates may have distinct composition (Figure 5I, yellow and blue arrows, respectively).”

2. Top of p.13 : It seems highly unlikely that the half-life of Group 2 droplets could have such a narrow time of recovery in the FRAP experiment ($t_{\text{half}}=6.93 \pm 0.01\text{s}$), especially if evaluating recovery intensity is somewhat subjective.

We thank the reviewer for this comment, we agree that we should not show t_{half} value as $t_{\text{half}}=6.93 \pm 0.01\text{s}$, but as $t_{\text{half}} > 6.93\text{s}$. In the FRAP experiment, we measured the time of recovery only for 10s, and Rbfox1 granules in Group 2 have not reached the plateau of their recovery by 10s. Now we corrected this in the figure and in the text. We could not do longer measurements because of the sample heating that would cause the tissue to move.

3. [REDACTED]

4. Authors should cite and discuss the very recent paper from Doug Black's lab on hydrogel behavior of mammalian RBFOX proteins, published in July 2017 (just before resubmission of this manuscript).

We now discussed the paper by the Black's lab about hydrogel behavior of mammalian RBFOX proteins.

Minor issues:

1. On p. 17, the data related to Pacman expression levels and the assembly of Pacman-positive P-bodies upon increased Rbfox1 levels is shown in Figure 6B-C (not Figure 5B-C).

Thank you, the typo is corrected.

--

Reviewer #2 (Remarks to the Author):

The revised manuscript is much improved in terms of its clarity and image qualities. The authors clearly demonstrated that (1) Rbfox1 level is regulated by starvation-dependent level of miR-980; (2) Rbfox1/miR-980 level is correlated with the cell survival; (3) Rbfox1/miR-980 level is correlated with the formation of RNA granules in different part of the cells. Though the mechanisms on how RNA granule formation contribute to cell survival and how LCD contribute to its "promiscuous" association with various cellular structure are not explored in this manuscript, the data presented are interesting and solid. Therefore, the manuscript should be considered for publication with minor adjustments into their claims:

(1) Abstract: "Reduced miR-980 expression during stress leads to increased Rbfox1 levels, followed by widespread formation of RNP granules, promoting cell survival." The latter part of the sentence implies causality between widespread formation of RNP granules and cell survival, where correlation is only shown.

Agree, now we say "Reduced miR-980 expression during stress leads to increased Rbfox1 levels, followed by widespread formation of RNP granules and promotes cell survival"

(2) Introduction: "Via its multiple LCDs, Rbfox1 promiscuously associates with the nucleolus and Cajal bodies in the nucleus as well as stress granules and processing bodies in the cytoplasm." It is not shown (though a reasonable hypothesis) in the manuscript that Rbfox1 promiscuously associates with different RNP granules in cytoplasm and nucleus due to its association with multiple LCDs.

True, now we say "Rbfox1 promiscuously associates with the nucleolus and Cajal bodies in the nucleus as well as stress granules and processing bodies in the cytoplasm, possibly via its multiple LCDs".

--

Reviewer #3 (Remarks to the Author):

All my concerns were addressed adequately. I recommend the manuscript for publication.

Thank you very much for positive evaluation of our work.

--

Reviewer #4 (Remarks to the Author):

Specific comments:

I stand by my first criticism that the paper is too unfocused to the detriment of its conclusions: miR-980 knockdown has interesting phenotypes, Rbfox is very complicated with many isoforms, Rbfox may change subcellular localization, there is the idea of liquid-droplet-like properties to Rbfox, etc. Any of these subjects by themselves is very complex. What is the main point of the paper?

In addition, while I appreciate the authors better describing the Rbfox antibodies, I remain unconvinced whether we are looking at the right things or not. The manuscript relies very heavily on immunofluorescence. It is exceedingly difficult to discern different staining patterns of Rbfox in the ovary of any of the figures.

Previous comment: Given that the authors rely heavily on two Rbfox1 antibodies, the antibodies should be better described. Where did the antibodies come from? Do they recognize all the

isoforms? Is the staining gone in an Rbfox1 null mutant ovary?

Author's response:

We also agree that we should have added more information on the Rbfox1 antibodies. We now added a new supplementary Figure S5, where we describe in greater detail which LCDs these antibodies can potentially recognize. In short, – LCD 2, 3, 6 and 7, while polyclonal guinea pig anti-Rbfox1 antibodies (Tastan et al., 2010) recognize polyclonal rabbit anti-Rbfox1 antibodies (Usha and Shashidhara, 2010) in addition recognize LCD 8, 9,10, 13, RRM and coiled-coiled domains. Based on the predicted by FlyBase Rbfox1 isoform sequences (Figure S2 and S4), both antibodies should recognize all Rbfox1 isoforms. Rbfox1 mutants are lethal; therefore, we could analyze only Rbfox1 RNAi clones and Rbfox1 hypomorphic mutants. In both cases, Rbfox1 levels were downregulated suggesting that the antibodies are specific. These data are now shown in Figure S3B, E, I.

Reviewer's Response: The representative micrograph shown in Fig. S3E (Rbfox1 RNAi) is not convincing with respect to reduced Rbfox protein levels. As described in the rebuttal above, this antibody should recognize all isoforms and the RNAi knockdown construct should recognize all isoforms as well.

It is true that the RNAi knockdown construct can recognize all isoforms, however, RNAi knockdown per se is not very efficient. We know this because the stronger downregulation of Rbfox1, for example in hypomorphic mutants (Fig. S3I) leads to severe follicle epithelium and germline differentiation defects, which is not observed by weaker RNAi knockdown. The severity of Rbfox1 downregulation correlates with Rbfox1 protein expression levels, please compare Fig. S3E (RNAi clone) and S3I (hypomorphic mutant).

Previous comment: In Fig 3 F, H, I it is difficult to see colocalization with Cajal bodies without single images and the mitochondrial labeling is not convincing.

Author's response: We are thankful for the reviewer's comment. We enlarged old and added new images with Rbfox1 and Cajal body marker colocalization in the nurse cells (Figure 5F).

Reviewer's response: It is still hard to see without the single-labeled images.

As suggested by the reviewer, we now show staining for Coilin and Rbfox1 as separate single-channel images (Figure 5F').

Author's response: We also added new images in Figure 5, 6 and S6 to make sure that the colocalization of Rbfox1 with different other markers of the liquid organelles is better shown. We also used new markers for P-body and the exosome complex to additionally support our conclusions about Rbfox1 subcellular localization (5H-I, 6F-G).

Reviewer's response: [REDACTED]

[REDACTED]

Previous comment: The authors use b-isox method to precipitate Rbfox as evidence that it forms cell-free RNP granules. Since they are using ovary extract, it would be helpful to have a positive control, in addition to a negative control, for this to be convincing.

Author's Response:

To precipitate Rbfox1, we used a commonly accepted protocol, where Rbfox1 overexpression was used as an additional positive control settings. We found that Rbfox1 association with b-isox is concentration-dependent. If it would depend on other unspecific proteins, we would not expect

to observe such an increase in the precipitated Rbfox1 levels due to Rbfox1 overexpression as we see in the lower panel of Figure 3A.

Reviewer's response: A positive control would be probing the blot for a protein that is known to precipitate in response to b-isox.

The point of this experiment was to show that Rbfox1 can be precipitated by b-isox, not to show that b-isox can precipitate any other protein. There are multiple studies that already used and published this method, which show that b-isox precipitates LCD-containing proteins. We just wanted to confirm that Rbfox1 behaves like other LCD-containing proteins; therefore, we are not certain why the reviewer wants us to prove that b-isox can precipitate a protein that is known to precipitate with b-isox. In addition, we used ovarian lysates and, as we mentioned before, we are not aware of any *Drosophila* ovarian proteins that precipitate with b-isox. We also should have mentioned in the previous response that there is an additional control that shows specificity of Rbfox1 precipitation by b-isox in our blots. Similar to Kato et al., 2012, we used different b-isox concentrations, and it is clear from the blot that Rbfox1 is not efficiently precipitated by 10 μ M of b-isox, but by 100 μ M, showing that Rbfox1 precipitation depends on b-isox concentrations. In addition, in the Rbfox1 overexpression experiment we clearly see that Rbfox1 levels, in particular 120 kD band, is reduced in supernatant, but increased in pellet upon 100 μ M of b-isox treatment, implying the b-isox-dependant relocation of Rbfox1 from supernatant to pellet.

Previous comment: The authors show Rbfox1 also forms filaments in the cells under stress (Fig. 2). They mention in the supplemental data that they tested CTPsynthase antibody to see if the filaments are cytoophidia and that these structures did not overlap. They need to show this data at least in the supplement. Rabbit antibodies have been known to overlap with cytoophidia, and guinea pig and rabbit antibodies can cross-react.

Author's response:

As suggested by the reviewer, we have added a new Figure S7, showing that Rbfox1 antibodies do not mark cytoophidia and histone locus bodies (new Figure S7).... Rbfox1 and CTPsyn co-staining shows no overlap.

Reviewer's response: Fig. S7 does not show Rbfox1-positive filaments in the same cells (miR-980/ park mutant) as CTPsyn-positive filaments. In addition, the "filaments" in Fig. 4C,C' look like they could be nuclear envelope and do not look the same as Fig. 4E.

Multiple cytoophidia fibres are present in all oocytes, at all stages and conditions. Rbfox1 fibres are very rare. The experiment we performed meant to test whether Rbfox1 antibodies would non-specifically bind to cytoophidia and they are not. Rbfox1 staining does not overlap with CTPsyn at all. Also, fibres in 4C are not nuclear envelope, we studied them in 3D when taking multiple confocal pictures. Fibres in 4C and 4E are different because they are of different magnification (note the scale bars) and in different cell types, the germline and follicle epithelium cells. Additional fibres are shown in Figure 3F and 5F, 4D.

Previous comment: Thioflavin T is commonly used in vitro to identify amyloid properties and is a spectroscopic assay. There seems to be non-specific background staining in the egg chamber shown in Fig. 2J in the nucleus and follicle cells, as well as what the authors are pointing out as an amyloid fiber (yellow arrow). Without proper controls, such as a positive control that is known to form amyloid in the germ cells, and a negative control with no starvation, the images do not make a compelling argument for Rbfox forming amyloid fibers.

Author's response:

Thioflavin staining is complicated, the required treatment does not always allow for the following antibody staining. We have repeated this experiment and added an additional panel to Figure 4, showing Thioflavin staining to be colocalized with Rbfox1 in a large nuclear droplet (Figure 4E).... Unfortunately, we are not aware of any protein that would make amyloid fibers in the germline and could be used as a positive control. Therefore, we make sure that we always call Rbfox1 fibres as "amyloid-like".

Reviewer's response: I appreciate that the authors now say "amyloid-like", but the staining is so

weak for the Rbfox1 co-localization (Fig. 4F, Rbfox1(GP)), and unconvincing for the nuclear stain (Fig. 4G (not E)). In addition, there is still no negative control of well-fed flies.

As suggested by the reviewer, we now added a negative control with no starvation (Figure 4F, Control).

Reviewer #1 (Remarks to the Author):

The authors addressed most of my major comments.

The issue raised in point #3, regarding statements that Rbfox1 assembles the granules, was addressed in the section heading as requested. However, similar over-statements are present in other places and should be modified.

Line 283-4: "we presumed that Rbfox1-mediated RNP aggregation could be stress-dependent".

And line 294: "To test whether Rbfox1 can assemble RNP granules via its LCDs". In these examples it could be a matter of semantics, but to me the wording in both cases implies as a matter of fact that Rbfox1 does assemble RNP granules, and the question is only whether this process is stress-dependent or mediated via LCDs. Moreover, at line 284, it hasn't been shown yet that Rbfox1 has any role in RNP aggregation.

2. There is another new paper published that should be cited: "Stress Granules Contain Rbfox2 with Cell Cycle-related mRNAs" [PMID:28894257]

Other small issues that were missed earlier (#1 and 2) or are new (#3):

1. In line 323: regarding the statement "Rbfox1 formed SDS-resistant aggregates in vivo": I suspect this should say in vitro, since the summary of this paragraph refers to in vitro results and the next paragraph focuses on in vivo data.

2. Line 457: "Rbfox proteins act as the splicing machinery components". This sentence is quite awkward. Perhaps the authors would like to refer to a recent review of Rbfox proteins (PMID:27748060).

3. The response to Reviewer 2's point #1 is the following awkward sentence, which needs editing: "Reduced miR-980 expression during stress leads to increased Rbfox1 levels, followed by widespread formation of RNP granules and promotes cell survival", making sure not to imply causality.

Comments on the response to Referee #4

Reviewer 4 requested what is the main point of the paper – as far as I can tell the authors never rebutted this point by giving a concise statement of the major finding.

1. Regarding the thread about antibody information and whether they are looking at the right things: There is more detail given in Figure S5, with a diagram illustrating proteins regions used in raising the antibodies. However, there is no mention in the text to say that the domains used for raising antibodies are shown in S5.

2. [REDACTED]

3. Regarding the thread about the need for positive and negative controls for the B-isox precipitation method: having a positive control is standard good practice to ensure that the conditions used are appropriate. Admittedly the demonstration that B-isox precipitates Rbfox1 with a similar concentration-dependence as described for other LCD proteins is reassuring, but it still would be nice to include a positive control if possible. Among the numerous reported LCD-containing proteins that precipitate with this protocol, aren't any of them expressed in Drosophila ovary?

Point-by-point responses to the reviewer's comments:

First of all we would like to thank the reviewer again for careful reading of our manuscript and helpful suggestions.

Reviewer #1 (Remarks to the Author):

The authors addressed most of my major comments.

The issue raised in point #3, regarding statements that Rbfox1 assembles the granules, was addressed in the section heading as requested. However, similar over-statements are present in other places and should be modified.

Line 283-4: "we presumed that Rbfox1-mediated RNP aggregation could be stress-dependent".

And line 294: "To test whether Rbfox1 can assemble RNP granules via its LCDs". In these examples it could be a matter of semantics, but to me the wording in both cases implies as a matter of fact that Rbfox1 does assemble RNP granules, and the question is only whether this process is stress-dependent or mediated via LCDs. Moreover, at line 284, it hasn't been shown yet that Rbfox1 has any role in RNP aggregation.

We agree, these statements are deleted, now we say:

"Rbfox1 aggregation could be stress-dependent" instead of "Rbfox1-mediated RNP aggregation could be stress-dependent".

and also:

"To test whether Rbfox1 can aggregate via its LCDs" instead of
"To test whether Rbfox1 can assemble RNP granules via its LCDs"

2. There is another new paper published that should be cited: "Stress Granules Contain Rbfox2 with Cell Cycle-related mRNAs" [PMID:28894257]

Thank you, we added this reference to the Discussion section.

Other small issues that were missed earlier (#1 and 2) or are new (#3):

1. In line 323: regarding the statement "Rbfox1 formed SDS-resistant aggregates in vivo": I suspect this should say in vitro, since the summary of this paragraph refers to in vitro results and the next paragraph focuses on in vivo data.

We meant to say not in lysates from cultured S2 cells, but from ovaries; but we agree, it is confusing and "in vivo" is deleted.

2. Line 457: "Rbfox proteins act as the splicing machinery components". This sentence is quite awkward. Perhaps the authors would like to refer to a recent review of Rbfox proteins (PMID:27748060).

Now we say "Rbfox proteins are involved in alternative splicing" and refer also to the recent review.

3. The response to Reviewer 2's point #1 is the following awkward sentence, which needs editing: "Reduced miR-980 expression during stress leads to increased Rbfox1 levels, followed by widespread formation of RNP granules and promotes cell survival", making sure not to imply causality.

Thank you, to avoid causality, we divided this sentence in two separate sentences: "Reduced miR-980 expression during stress leads to increased Rbfox1 levels and widespread formation of Rbfox1-positive RNP granules. Importantly, Rbfox1 upregulation in response to stress promotes cell survival".

Comments on the response to Referee #4

Reviewer 4 requested what is the main point of the paper – as far as I can tell the authors never rebutted this point by giving a concise statement of the major finding.

True, we never rebutted this point because we thought that it is clear that a concise statement of the major findings is in the summary of our paper.

In summary, we found that Rbfox1 is a pro-survival RNA-binding protein with low complexity sequence domains (LCDs). Our *in vivo* studies using *Drosophila* oogenesis as a model system for stress response demonstrate that depending on concentration and isoform composition, it can form various nuclear and cytoplasmic RNP granules, ranging from liquid droplets to hydrogels to beta-amyloid-like fibers. We uncovered an elegant mechanism by which Rbfox1 levels are adjusted by a stress-dependent miRNA. *miR-980* acts to buffer Rbfox1 levels, since it can target only the portion of *Rbfox1* transcripts that contain extended 3'UTRs. This prevents the reduction of Rbfox1 levels below a certain threshold, which is also detrimental for cellular homeostasis. Reduced *miR-980* expression during stress leads to increased Rbfox1 levels, followed by widespread formation of RNP granules. We found that human RBFOX proteins also contain multiple LCDs and form membraneless compartments, suggesting that the RNP-granule-linked control of cellular adaptive responses may contribute to a wide range of RBFOX-associated pathologies in humans.

1. Regarding the thread about antibody information and whether they are looking at the right things: There is more detail given in Figure S5, with a diagram illustrating proteins regions used in raising the antibodies. However, there is no mention in the text to say that the domains used for raising antibodies are shown in S5.

Thank you, now we mentioned that we used two different antibodies and gave the references to the original papers where they were produced already in the first section of Results. Also, we refer the reader to the detailed description of antibodies in the Materials and Methods section that has references to Figure S5 (protein domains) and also Figures S2 and S4 that show Rbfox mRNA isoforms and protein sequences, respectively. We could not refer to Figure S5 in the first section, because we would have to renumber all figures, which would affect current logic of the paper.

2. [REDACTED]

[REDACTED]

[REDACTED]

[REDACTED]

3. Regarding the thread about the need for positive and negative controls for the B-isox precipitation method: having a positive control is standard good practice to ensure that the conditions used are appropriate. Admittedly the demonstration that B-isox precipitates Rbfox1 with a similar concentration-dependence as described for other LCD proteins is reassuring, but it still would be nice to include a positive control if possible. Among the numerous reported LCD-containing proteins that precipitate with this protocol, aren't any of them expressed in *Drosophila* ovary?

We thank the reviewer for saying “Admittedly the demonstration that B-isox precipitates Rbfox1 with a similar concentration-dependence as described for other LCD proteins is reassuring, but it still would be nice to include a positive control if possible”. In Kato et al., 2012, b-isox precipitation was performed only in S2 cells and to the best of our knowledge, b-isox precipitation was never performed using ovaries. As we mentioned before, we are not aware of any proteins that were

precipitated with b-isox using ovarian lysates. Since the germline is known to have quite unique protein expression profile, finding this “positive control” could require a lot of new experiments that are not relevant to this research.

In addition, to precipitate Rbfox1, we used a commonly accepted protocol and there are multiple studies that already used and published this method, which show that b-isox precipitates LCD-containing proteins. Since we directly followed the protocol, we think that it is somewhat unreasonable to ask for an additional control for this experiment. In addition, the “positive control” would be required to show that the method works if Rbfox1 would not be precipitated with b-isox, while in our hands, Rbfox1 does precipitate with b-isox in the concentration-dependent manner, similar to the previously described.

However, we agree with the reviewer that it is a good practice to have multiple controls and ensure that the conditions used are appropriate, in particular for this experiment we used the following controls:

First, Rbfox1 overexpression was used as a positive control settings. We found that Rbfox1 association with b-isox is concentration-dependent. If it would depend on other unspecific proteins, we would not expect to observe such an increase in the precipitated Rbfox1 levels due to Rbfox1 overexpression as we see in the lower panel of Figure 3A.

Second, in the Rbfox1 overexpression experiment we also clearly see that Rbfox1 levels, in particular 120 kD band, is reduced in supernatant, but increased in pellet upon 100 μ M of b-isox treatment, implying the b-isox-dependent relocation of Rbfox1 from supernatant to pellet.

Third, we used different b-isox concentrations, and it is clear from the blot that Rbfox1 is not efficiently precipitated by 10 μ M of b-isox, but by 100 μ M, showing that Rbfox1 precipitation depends on b-isox concentrations.

Therefore, we are quite confident that *Drosophila* Rbfox1 from ovaries precipitates with b-isox.